# KRAS-dependent glycolytic reprogramming of endothelial cells in sporadic arteriovenous malformations

Ruilin Wu[1,2], Negar Khosraviani [1,2], Ann Mansur[1,3], Emilie Boudreau [2], Gabrielle E Largoza [4,5], Suejean Park[1,2], Dakota Gustafson [1,2,6], Sneha Raju [7,8], Crizza Ching[2,7], Amira Klip[9,10,11], Thomas Wälchli [3,12,13], Kathryn L Howe [1,2,7,8,14], Ivan Radovanovic[1,3,15], Joshua D Wythe [4,5] & Jason E Fish [1,2,7,14 ✉]

## Abstract

Somatic activating *KRAS* mutations in endothelial cells are the predominant cause of sporadic brain arteriovenous malformations (bAVMs) and also occur in sporadic extracranial AVMs. We found that KRAS[G12V] expression in the endothelium increased angiogenesis, which was accompanied by enhanced glucose uptake and glycolytic flux. Mechanistically, this increase in glycolysis was facilitated by enhanced membrane localization of glucose transporters (e.g., GLUT1) and induction of hexokinase-2 (HK2) expression. Importantly, RNA-sequencing and proteomics revealed that HK2 appeared to be the only glycolytic component elevated. Analysis of single-cell RNA-sequencing data and immunofluorescence staining confirmed that HK2 was elevated in mouse and human bAVMs. Critically, either pharmacologic inhibition of glycolytic flux or knockdown of *HK2* suppressed sprouting angiogenesis in cultured KRAS[G12V] endothelial cells. Glycolysis inhibition also reversed arteriovenous shunts and potentiated the effect of MEK inhibition in a KRAS-mutant zebrafish model. Finally, combined glycolysis and MEK inhibition suppressed angiogenesis in patient-derived bAVM primary endothelial cells. Together, our findings show that KRAS-driven reprogramming of endothelial metabolism represents a potential therapeutic vulnerability for sporadic AVMs.

**Keywords** Endothelial Glycolysis; Arteriovenous Malformations; Metabolic Regulation; KRAS Signaling; Targeted Therapeutics
**Subject Categories** Metabolism; Vascular Biology & Angiogenesis

See also: M Rudnicki and TL Haas

## Introduction

Arteriovenous malformations (AVMs) are vascular anomalies characterized by the presence of a tangled knot of dilated and fragile vessels (also referred to as a nidus), resulting from the direct connection between arteries and veins that bypasses the normal intervening capillary vasculature (Al-Shahi et al, 2002; Wälchli et al, 2023). The AVM nidus exhibits a tortuous vascular morphology and altered hemodynamics with high-pressure, fast-flowing arterial blood continuously shunting into the venous system (Solomon and Connolly 2017; Wälchli et al, 2023). Consequently, the vessels of the nidus are highly susceptible to rupture, as evidenced by the elevated risk of spontaneous intracranial hemorrhage and stroke in brain AVM (bAVM) patients (Solomon and Connolly 2017; Wälchli et al, 2023). Current methods of intervention for AVMs rely on invasive surgical procedures (i.e., microsurgery, radiosurgery, and endovascular embolization), which are not suitable for all patients, and no approved pharmacological treatment options are currently available (Derdeyn et al, 2017; Sugiyama et al, 2022). This highlights two critical needs in AVM research: (1) detailed investigation of the molecular mechanisms underlying the initiation and maintenance of AVMs; and (2) discovery of novel therapeutic targets and non-invasive strategies to treat this disease.

Our group identified somatic activating *KRAS* mutations, and others have subsequently confirmed the presence of p.G12V, G12D, G12C, and Q61H *KRAS* mutations in clinical sporadic bAVM and extracranial AVM tissues (Al-Olabi et al, 2018; Bameri et al, 2021; El Sissy et al, 2022; Hong et al, 2019; Li et al, 2021; Nikolaev et al, 2018; Priemer et al, 2019; Schmidt et al, 2024). Importantly, we found that *KRAS* mutations were exclusively localized to CD31-expressing endothelial cells (ECs) (Nikolaev et al, 2018).

[1]Department of Laboratory Medicine and Pathobiology, Temerty Faculty of Medicine, University of Toronto, Toronto, ON, Canada. [2]Toronto General Hospital Research Institute, University Health Network, Toronto, ON, Canada. [3]Division of Neurosurgery, Department of Surgery, Toronto Western Hospital, University Health Network, Toronto, ON, Canada. [4]Department of Cell Biology, University of Virginia School of Medicine, Charlottesville, VA, USA. [5]Robert M. Berne Cardiovascular Research Center, University of Virginia School of Medicine, Charlottesville, VA, USA. [6]Department of Public Health Sciences, Queen's University, Kingston, ON, Canada. [7]Institute of Medical Science, Temerty Faculty of Medicine, University of Toronto, Toronto, ON, Canada. [8]Division of Vascular Surgery, Department of Surgery, Toronto General Hospital, Toronto, ON, Canada. [9]Cell Biology Program, The Hospital for Sick Children, Toronto, ON, Canada. [10]Department of Biochemistry, Temerty Faculty of Medicine, University of Toronto, Toronto, ON, Canada. [11]Department of Physiology, Temerty Faculty of Medicine, University of Toronto, Toronto, ON, Canada. [12]Victor Horsley Department of Neurosurgery, National Hospital for Neurology and Neurosurgery, University College London Hospitals, London, UK. [13]Division of Neurosurgery, University Hospital Zurich, Zurich, Switzerland. [14]Peter Munk Cardiac Centre, University Health Network, Toronto, ON, Canada. [15]Krembil Brain Institute, University Health Network, Toronto, ON, Canada. ✉E-mail: jason.fish@utoronto.ca

Subsequent studies in mouse and zebrafish models showed that EC-specific expression of mutant KRAS was sufficient to drive the development of AVM phenotypes (Fish et al, 2020; Nguyen et al, 2023; Park et al, 2021). This demonstrated the critical role of somatic activating KRAS variants within the endothelium in sporadic AVM pathogenesis. Mechanistically, multiple studies showed that mutant KRAS preferentially elevated MAPK/ERK signaling over other downstream effector pathways (i.e., PI3K/AKT and p38). Upregulated MEK/ERK signaling resulted in: (1) impaired barrier integrity with decreased junctional vascular endothelial (VE)-cadherin; (2) enlarged cell size in conjunction with expanded vessel lumen diameter in mice and zebrafish; and (3) enhanced cell motility and ectopic angiogenic sprouting, resulting in abnormal connections between arteries and veins (Fish et al, 2020; Nikolaev et al, 2018; Soon et al, 2022). Others have observed degradation of the basement membrane matrix, as well as altered mural cell interactions, including a reduction in pericyte coverage on blood vessels both in vitro and in clinical bAVM tissue (Sun et al, 2022; Winkler et al, 2018). Recent single-cell (and bulk) RNA-sequencing studies revealed an inflammatory signature in sporadic bAVM with changes to the overall immune landscape driven by both ECs and immune cells (Wälchli et al, 2024; Winkler et al, 2022). These transcriptional profiling studies also highlighted alterations to arteriovenous specification, reactivation of fetal programming, and loss of cerebrovascular identity of brain ECs in bAVM (Wälchli et al, 2024; Winkler et al, 2022). Relatedly, BRAF, a RAF family kinase that functions downstream of KRAS to activate MEK, can also be mutated in sporadic bAVMs, albeit much less frequently, highlighting the key role of KRAS/BRAF/MEK/ERK signaling (Al-Olabi et al, 2018; Hong et al, 2019). Importantly, activating KRAS mutations are also found in extracranial AVMs (Al-Olabi et al, 2018; El Sissy et al, 2022; Schmidt et al, 2024), demonstrating the ability of these mutations to drive vascular morphological abnormalities in vascular beds outside of the central nervous system.

Despite emerging studies on the pathobiology of sporadic AVMs, how mutant KRAS signaling impacts endothelial metabolism remains unexplored. ECs lining the interface between circulating blood and perivascular tissue, where gas and nutrient exchange occur, exhibit unique metabolic characteristics (Falkenberg et al, 2019). These cells are inherently more glycolytic than other cell types, even with sufficient access to oxygen, and can reprogram their metabolism to meet changing metabolic demands (Falkenberg et al, 2019). When ECs are quiescent, they upregulate fatty acid oxidation to maintain redox homeostasis (Kalucka et al, 2018). During sprouting angiogenesis, however, tip and stalk ECs increase their glycolytic flux to generate the energy and biomaterials required to fuel vascular growth (Du et al, 2021). Endothelial glycolysis has been directly linked to active cytoskeletal remodeling and regulation of cell motility (Wu et al, 2021). Suppressing glycolysis also lowers VE-cadherin endocytosis, resulting in a tightened EC barrier (Cantelmo et al, 2016). Recent studies emphasize the importance of glycolytic by-products (e.g., lactate) in modulating the extracellular microenvironment and endothelial-mural cell interactions (Li et al, 2022). Under pathological conditions, metabolic dysregulation is often closely tied to disease progression, as occurs during the rewiring of cancer cell metabolism, resulting in a glycolytic shift, known as the Warburg effect, with increased glucose uptake and lactate secretion (Warburg 1925). This metabolic reprogramming also occurs in tumor ECs to drive tumor perfusion that supplies nutrients to the growing malignant cells (Du et al, 2021).

KRAS is an oncogene, and the same mutations detected in sporadic AVMs are also known to drive cellular changes, including metabolic rewiring, in various cancers (Huang et al, 2021). As such, current therapeutic development efforts for KRAS-dependent AVMs are focusing on repurposing cancer drugs that target major signaling pathways acting downstream or in parallel to KRAS, such as trametinib (MEK), rapamycin (mTOR), and bevacizumab (VEGF) (Mansur and Radovanovic 2023; Mansur and Radovanovic 2024). By inhibiting these signaling effectors, pathological changes that are "addicted" to continuous KRAS signaling activities can potentially be reversed, leading to therapeutic recovery. However, while conventional cancer treatments aim to target cell proliferation and induce cancer cell death to eliminate tumor growth (Debela et al, 2021), sporadic AVM lesions contain a heterogenous population of wild-type and mutant ECs within a high-flow cerebral environment (Nikolaev et al, 2018). Thus, therapies targeting cell viability might do more harm than good in AVM management. It remains unclear if knowledge from the oncology field, including insight into cancer metabolism, can be directly translated to the management of sporadic AVMs.

Herein, we demonstrate that expression of mutant KRAS reprograms EC metabolism towards glycolysis. We show that this metabolic shift is primarily driven by an increase in the membrane localization of the glucose uniporter Glucose Transporter 1 (GLUT1), which leads to elevated glucose uptake, along with upregulated transcription of hexokinase 2 (HK2), which increases glycolytic flux. Interrogation of transcriptomic and proteomic profiles implicates HK2 as a specific mediator of this metabolic change, which we have validated using single-cell RNA-sequencing datasets and immunofluorescence in mouse and human patient bAVMs. Functionally, we show that disrupting glycolysis via small molecule inhibition or via knockdown of HK2 reverses pathological changes in mutant KRAS ECs, and glycolysis inhibitors provide combinatorial benefits when used in conjunction with MEK inhibitors in ECs isolated from patient bAVMs and in zebrafish AVM models. Collectively, these studies provide novel insights into the molecular mechanisms underlying sporadic AVM pathogenesis and identify a promising therapeutic strategy for future treatment of these lesions by leveraging their dependence on glycolysis.

## Results

### KRAS^G12V-expressing endothelial cells display altered barrier, angiogenic, and metabolic phenotypes

To investigate the molecular and cellular effects of mutant KRAS expression on endothelial cell (EC) biology, we utilized our established and validated telomerase-immortalized human umbilical vein endothelial cell (IM-HUVEC) lines (Hayer et al, 2016; Soon et al, 2022) with stable incorporation of either doxycycline (Dox)-inducible mScarlet-tagged wild-type (WT) or mutant G12V KRAS – one of the predominant mutations identified in patients with sporadic bAVM and also identified in sporadic extracranial AVMs (Al-Olabi et al, 2018; Nikolaev et al, 2018; Alharbi et al, 2025). We previously demonstrated that these cells can be used to generate an

AVM-on-a-chip model (Soon et al, 2022). We confirmed the endothelial identity of these cell lines by examining the expression of endothelial and mesenchymal marker genes in the presence and absence of Dox in comparison to non-transfected IM-HUVECs (Fig. EV1A–C). Of note, we also attempted to generate a stable Dox-inducible line using immortalized brain ECs, but these cells lost their endothelial character during selection. Using western blotting, we confirmed that exogenous KRAS proteins (mScarlet-KRAS$^{WT}$ or mScarlet-KRAS$^{G12V}$) were only expressed at modest levels exclusively upon Dox induction of IM-HUVECs (Fig. EV1D–F). Our group previously demonstrated that ECs isolated from clinical KRAS-mutant bAVM samples, as well as ECs transiently electroporated with mutant KRAS constructs, preferentially activated the MAPK/ERK pathway (Fish et al, 2020; Nikolaev et al, 2018). We confirmed that the stable KRAS$^{G12V}$ cell line also showed elevated MAPK/ERK pathway activity, as early as 6 h after Dox induction, with no change in PI3K/AKT signaling (Fig. EV1E,F). Upregulation of SPRY4 and EGR1, two established MAPK/ERK target genes, occurred only in the IM-HUVEC KRAS$^{G12V}$ cell line in the presence of Dox treatment (Fig. EV1B,C), further confirming MAPK/ERK pathway activation in mutant KRAS ECs. Finally, we showed that inhibition of MAPK/ERK signaling, using a MEK inhibitor, SL327, did not impact endogenous or exogenous KRAS protein expression (Fig. EV1D).

Using this validated in vitro cell line model, we performed a series of phenotypic and functional assays to gain a holistic understanding of the cellular changes driven by mutant KRAS signaling in ECs. Consistent with previous findings (Fish et al, 2020; Nikolaev et al, 2018; Soon et al, 2022), KRAS$^{G12V}$ expression in ECs led to compromised barrier integrity as shown by the diminished localization of VE-cadherin to adherens junctions and increased leak of a 70 kDa fluorescent tracer molecule in a transwell leak assay (Fig. 1A,B). Interestingly, neither CDH5 transcript nor VE-cadherin protein levels were altered, suggesting that the observed junctional disruption was due to altered subcellular localization, rather than decreased protein expression (Figs. 1C and EV1B). In line with this, barrier disruption was more severe when KRAS$^{G12V}$ expression was induced prior to the establishment of a confluent monolayer compared to after (Appendix Fig. S1A). Notably, phosphorylation of VE-cadherin at Tyr658, which regulates p120-catenin interaction and modulates barrier integrity (Potter et al, 2005), was not significantly altered (Appendix Fig. S1B).

In addition to barrier defects, KRAS$^{G12V}$ ECs also demonstrated enhanced angiogenic sprouting (Fig. 1D) with increased secretion of several angiogenic factors, such as ANGPT2, CXCL8, PlGF, HB-EGF, PDGF-AA and others (Fig. 1E; Appendix Fig. S1C–E). Moreover, KRAS$^{G12V}$ ECs were more migratory (Fig. 1F) with significant extracellular matrix (ECM) remodeling. These ECM changes included reduced levels of hyaluronic acid, a component of the apical glycocalyx, and fragmented fibronectin, typically found in the basolateral basement membrane (Fig. 1G,H; Appendix Fig. S1F). ECs are known to exhibit unique metabolic adaptations – being relatively more glycolytic than other cell types (Falkenberg et al, 2019). Quantification of the extracellular acidification rate (ECAR) showed that KRAS$^{G12V}$ ECs featured a notable elevation of glycolysis in comparison with KRAS$^{WT}$ ECs, without changes in mitochondrial respiration as measured by the oxygen consumption rate (OCR) (Fig. 1I; Appendix Fig. S1G). This shift in metabolism

was dependent on MAPK/ERK signaling as treatment with the MEK inhibitor, SL327, negated the increase in glycolysis (Fig. 1I; Appendix Fig. S1G).

Clinically, sporadic AVM lesions are comprised of a mixture of KRAS$^{WT}$ and KRAS$^{mutant}$ ECs (Nikolaev et al, 2018). To explore the cell autonomous versus non-autonomous effects of mutant KRAS signaling, we co-cultured Dox-inducible KRAS$^{WT}$ or KRAS$^{G12V}$ ECs with another IM-HUVEC line (F-Tractin-mCherry, which fluorescently labels filamentous actin) at a 1:1 ratio (Fig. EV2A). Immunofluorescence (IF) staining showed that activation of ERK (as revealed by staining for phosphorylated ERK [pERK]) and VE-cadherin disruptions were confined to Dox-induced mutant KRAS ECs, even though F-Tractin-mCherry$^{+}$ ECs were in direct contact with mutant cells (Fig. EV2B,C). While decreased fibronectin abundance was also predominantly observed under mutant KRAS ECs, some loss was evident at boundary regions between mutant KRAS and F-Tractin-mCherry ECs (Arrows; Fig. EV2D). Tracing the cell boundaries showed that KRAS$^{G12V}$ ECs displayed increased cell size and decreased circularity (i.e., a more elongated cell morphology) compared to KRAS$^{WT}$ cells (Fig. EV2I,J). BrdU incorporation and Ki67 IF did not reveal any differences in proliferation, and TUNEL staining did not reveal any changes in DNA damage/cell death between KRAS$^{WT}$ and KRAS$^{G12V}$ ECs (Fig. EV2E–G,K,L). However, analysis of each EC population revealed a significant reduction in the area occupied by IM-HUVEC F-Tractin cells when co-cultured with KRAS$^{G12V}$-expressing ECs but not KRAS$^{WT}$-expressing ECs (Fig. EV2H,M). This suggests that mutant KRAS ECs have a competitive advantage over WT ECs, potentially independent of mechanisms involving increased proliferation or reduced apoptosis.

## Establishing the presence of the Warburg Effect in KRAS$^{G12V}$ ECs

KRAS is a prominent oncogene, and one distinguishing feature of cancer cells is a pronounced shift in their metabolism from oxidative phosphorylation to glycolysis, also known as the Warburg Effect (Warburg 1925). In ECs, glycolysis is crucial for tip cell differentiation (Yetkin-Arik et al, 2019) and vessel sprouting (De Bock et al, 2013), establishing a potential connection between metabolic alterations and the formation of excessive, tangled vascular networks present in sporadic AVMs. Metabolic alterations in mutant KRAS ECs and their contribution to AVM pathogenesis remain unknown.

Accordingly, we first examined the expression of mRNAs encoding several key glycolytic enzymes using qPCR. Among the genes tested, HK1 and HK2 were upregulated in ECs expressing KRAS$^{G12V}$ (Fig. 2A). We then measured the protein levels of HK1 and HK2, of which only HK2 showed a significant increase in KRAS$^{G12V}$-expressing ECs (Fig. 2B). The Warburg Effect is defined by both an increase in glucose uptake and lactate secretion (Warburg 1925). We performed two glucose uptake assays: imaging of the uptake of a fluorescent glucose analog 2-(N-(7-Nitrobenz-2-oxa-1,3-diazol-4-yl)amino)-6-Deoxyglucose (2-NBDG) and a luminescent assay that measures 2-deoxy-d-glucose (2-DG) uptake and conversion to 2-deoxyglucose-6-phosphate. Both assays showed enhanced glucose uptake in KRAS$^{G12V}$-expressing ECs (Fig. 2C,D). BAY876, a potent and selective inhibitor of the glucose transporter, GLUT1, significantly suppressed glucose uptake. Treatment with

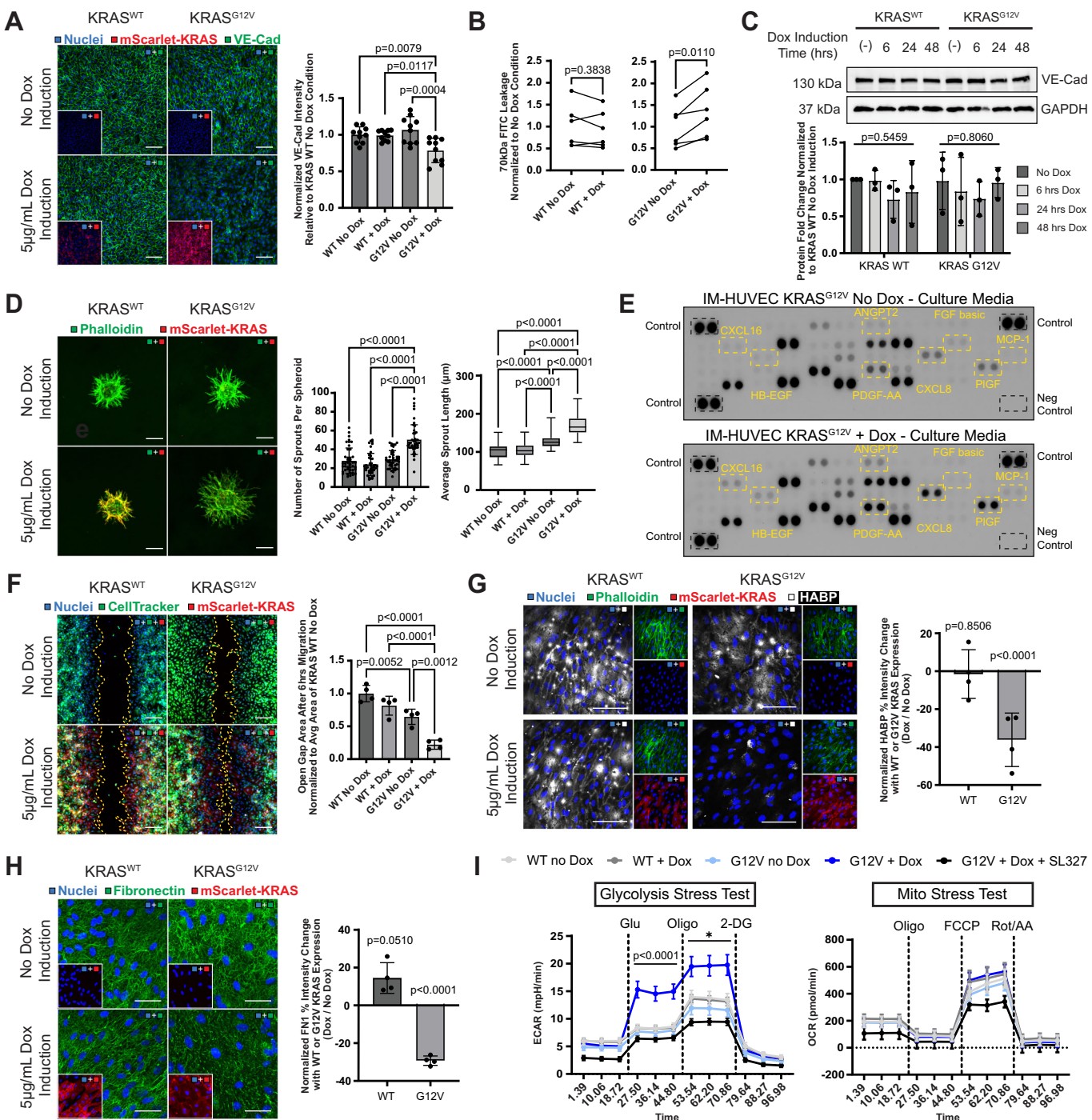

the MEK inhibitor SL327 also lowered glucose uptake in cells expressing KRAS[G12V] (Fig. 2D). Levels of *GLUT1* and *GLUT3* mRNA, which encode the major glucose transporters in ECs, were unaffected (Fig. 2E). However, membrane localization of these transporters is known to affect their function (Cura and Carruthers 2010). Thus, we analyzed GLUT1 subcellular localization via cell fractionation and western blotting, which revealed an increase in GLUT1 plasma membrane localization in KRAS[G12V] ECs (Fig. 2F, G). GLUT1 levels were unchanged in the total membrane fraction, indicating that these changes were not due to differences in GLUT1

expression between KRAS[WT] and KRAS[G12V] ECs (Appendix Fig. S2A). There was also a trend toward an increase in GLUT3 membrane localization, but this did not reach statistical significance (Appendix Fig. S2B). Downstream of the glycolysis pathway, there was an increase in lactate secretion in KRAS[G12V] ECs. 2-DG (an inhibitor of hexokinase activity) and BAY876 (glucose transporter inhibitor) treatments both suppressed lactate secretion to a similar extent, while SL327 (MEK inhibitor) did not affect lactate secretion (Fig. 2H). In parallel to lactate secretion, we also observed a significant increase in proton secretion as evidenced by altered

**Figure 1. Endothelial cells expressing mutant KRAS display altered barrier, angiogenic, and metabolic phenotypes.**

Experiments were conducted with IM-HUVEC KRAS$^{WT}$ and KRAS$^{G12V}$ cells (−/+ doxycycline [Dox] induction). (**A**) Representative immunofluorescence (IF) staining and quantification of VE-cadherin (Green). Mean ± standard deviations (SD). One-way ANOVA with Tukey's post hoc tests. n (independent experiments) = 3 (3–4 fields of view per replicate). Scale bars = 100 μm. (**B**) Fluorescence measurements of 70 kDa FITC transwell leak assay. Two-tailed paired t test. n = 6. (**C**) Representative western blots and quantification of VE-cadherin. Mean ± SD. One-way ANOVA. n = 3. (**D**) Representative spheroid sprouting assay (24 h). Bar graph: mean ± SD. Box and whiskers: min to max. Kruskal–Wallis test with Dunn's multiple comparisons. n = 5 (5–12 spheroids per condition per replicate). Scale bars = 200 μm. (**E**) Representative Angiogenesis Dot Blot profiling of cell culture media. Significantly changed proteins are highlighted using yellow dotted boxes. n = 3. (**F**) Representative image of wound migration assay (6 h). Yellow dotted lines highlight cell boundaries after migration. Mean ± SD. One-way ANOVA with Tukey's post hoc tests. n = 4. Scale bars = 200 μm. (**G**) Representative IF staining and intensity quantification of hyaluronic acid binding protein (HABP). Mean ± SD. Two-tailed unpaired t tests (comparing Dox vs No Dox). n = 4. Scale bars = 100 μm. (**H**) Representative IF staining and intensity quantification of fibronectin (FN1). Mean ± SD. Two-tailed unpaired t tests (comparing Dox vs No Dox). n = 4. Scale bars = 50 μm. (**I**) Glycolysis stress test (left) and mitochondrial (mito) stress test (right). SL327 treatment = 2 μM. Mean ± standard error of means (SEM). Selected statistics shown represent significant changes between KRAS$^{G12V}$ +Dox condition compared to all other conditions. One-way ANOVA with Tukey's post hoc tests. n = 6 (glycolysis test), n = 5 (mito test) and n = 4 for SL327 treatments. In this figure, *P value < 0.02 with exact P values shown in Appendix Table S2. Source data are available online for this figure.

media pH after long-term culture (Fig. 2I; Appendix Fig. S2C). In addition to the increased extracellular acidification indicated by the Seahorse assay (Fig. 1I), alteration in the pH of cell culture media demonstrated that the degree of proton secretion by KRAS$^{G12V}$ ECs was sufficient to impact the extracellular environment, even in the presence of a pH buffer system in the media. Inhibition of hexokinase activity, glucose transport, or MEK could rescue this effect (Fig. 2I). It is worth highlighting that the extracellular pH of KRAS$^{G12V}$ ECs (~6.9) is comparable to the reported extracellular pH of cancer cells (~6.7–7.1) (Webb et al, 2011).

## Transcriptomic analyses implicate HK2 as a mediator of metabolic reprogramming in KRAS$^{G12V}$ ECs

To determine whether additional glycolytic components are altered in mutant KRAS ECs, we performed bulk RNA-sequencing and total lysate tandem mass-tag (TMT) proteomics to profile global RNA and protein changes, respectively (Fig. 3A). In the RNA-sequencing study, we included several conditions where cells were treated with different glycolysis inhibitors to gain a deeper mechanistic understanding of the impact of glycolytic suppression in KRAS$^{G12V}$ ECs. In addition to the previously mentioned glycolysis inhibitors (i.e., 2-DG and BAY876), we added another glycolysis inhibitor, 3-(3-Pyridinyl)-1-(4-pyridinyl)-2-propen-1-one (3-PO), which was proposed to inhibit PFKFB3 (Cantelmo et al, 2016). The MEK inhibitor, SL327, was also included in both RNA-sequencing and proteomics experiments to examine the MEK-dependent effects of KRAS$^{G12V}$ signaling in ECs.

RNA-sequencing revealed 445 upregulated and 417 down-regulated genes (P value < 0.05) between KRAS$^{G12V}$ ECs with and without Dox induction (Fig. 3B). In agreement with our previous studies (Fish et al, 2020; Nikolaev et al, 2018), gene ontology analysis of differentially expressed genes (DEGs) identified enrichment for pathways related to cell motility, angiogenesis, cell and matrix adhesion, and cytoskeleton organization (Fig. 3C). Notable pathways that were represented in upregulated genes were Ras/MAPK signaling, DNA replication, nucleotide biosynthesis and ribosome biogenesis, whereas downregulated genes revealed pathways related to stress response, smooth muscle cell interactions, and apoptosis (Appendix Fig. S3A,B). Cellular Component analysis showed distinct enrichment of the nuclear compartment

amongst upregulated genes, and overrepresentation of the junctional and ECM compartments amongst downregulated genes (Appendix Fig. S3C,D). These results matched our phenotypic characterizations (Fig. 1). Pathway analysis of unique DEGs with 2-DG treatment highlighted disruptions of protein folding and processing, likely due to 2-DG's suppressive effect on protein glycosylation (Kurtoglu et al, 2007) (Fig. EV3A,B). Ribosome biogenesis was uniquely affected by SL327 (MEKi), while DEGs with BAY876 treatment impacted peptide translation (Fig. EV3C,D). Unique DEGs with 3-PO treatment did not reveal any statistically significant pathway enrichment. In fact, 3-PO treatment did not show any functional effects in follow-up studies, and hence, was excluded from later experiments. Conversely, shared DEGs with 2-DG and BAY876 treatments of KRAS$^{G12V}$ ECs showed enrichment of glycolysis, validating the functionality of these inhibitors (Fig. EV3E). The top pathway affected by both glycolytic and MEK inhibition was cell cycle regulation (Fig. EV3F,G).

Focusing on glycolysis-related genes, the only upregulated gene in KRAS$^{G12V}$ ECs was *HK2* (Fig. 3D). Furthermore, we did not observe significant changes in genes involved in other metabolic pathways, such as the citrate cycle, pentose phosphate pathway, pyruvate, fructose, and mannose metabolism. SL327 treatment downregulated the expression levels of both *HK1* and *HK2*, while glycolytic inhibition had no direct effect on *HK2* transcript levels (Appendix Fig. S4). This observation suggested that *HK2* upregulation could be due to MAPK/ERK signaling activity. To test our hypothesis, we first confirmed that HK2 protein levels were downregulated following SL327 treatment (Appendix Fig. S5A). We then stimulated MAPK/ERK signaling in both KRAS$^{WT}$ and KRAS$^{G12V}$ ECs, with and without Dox induction, using either VEGF, thrombin, or PMA (Phorbol 12-myristate 13-acetate), which are all known activators of MAPK/ERK signaling. Despite high levels of ERK activation (as revealed by pERK immunoblotting and assessment of ERK-target gene expression), all three stimulated conditions failed to alter HK2 expression (Appendix Fig. S5B–H). These results indicate that while the MAPK/ERK pathway is a necessary and central regulator of cellular metabolism, activation of this effector cascade alone is insufficient to drive metabolic reprogramming in ECs through HK2 induction.

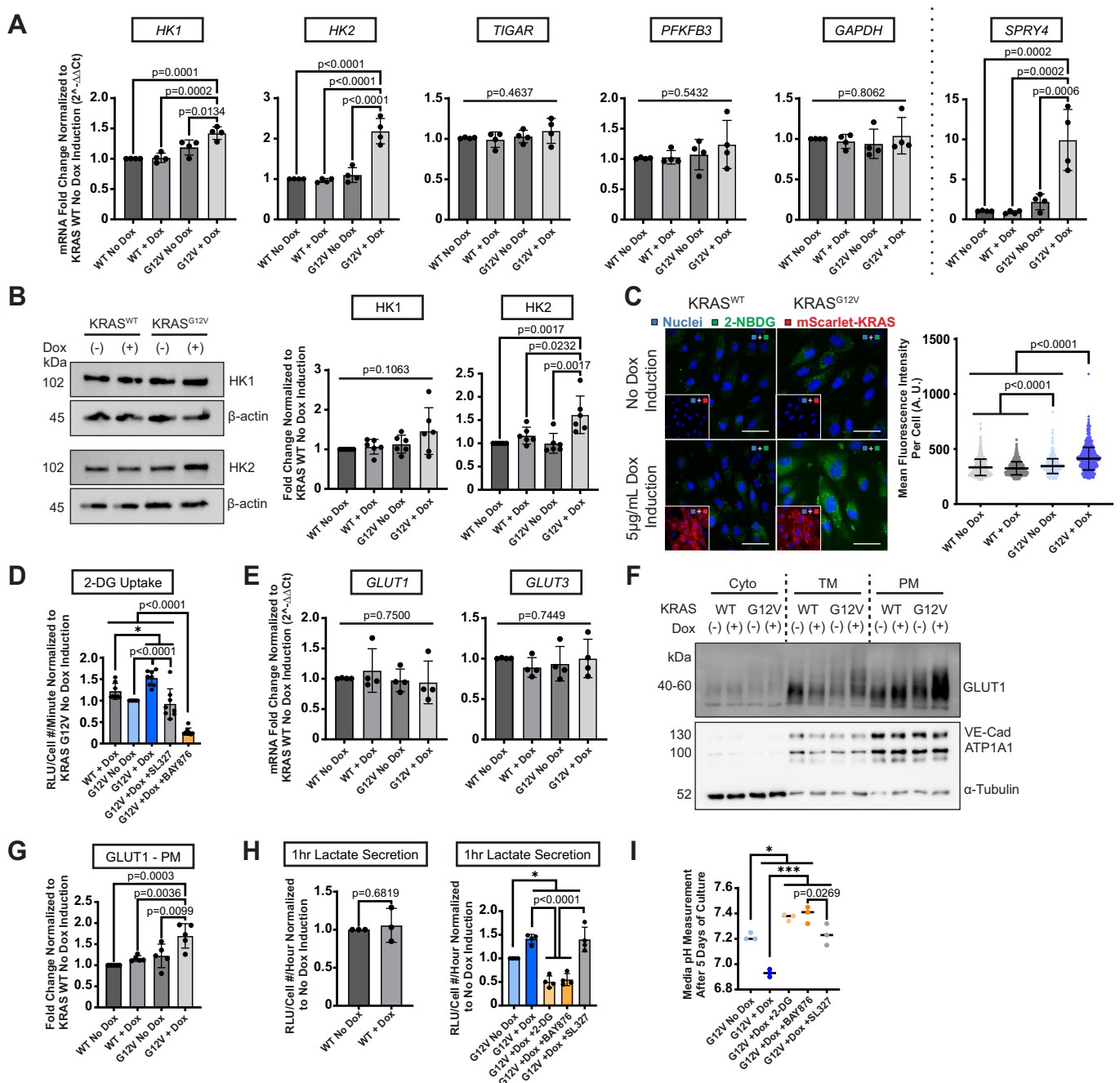

**Figure 2. Mutant KRAS ECs exhibit the Warburg effect.**

Experiments were conducted with IM-HUVEC KRAS cell lines (−/+ Dox induction). (A) qPCR of glycolysis-related genes. Housekeeping: *TBP* and *GAPDH*. MAPK/ERK pathway activation: *SPRY4*. Mean ± SD. One-way ANOVA with Tukey's post hoc test. $n = 4$. (B) Representative western blots and quantifications of HK1 and HK2. Mean ± SD. One-way ANOVA with Tukey's post hoc test. $n = 6$. (C) 2-NBDG (150 μM) uptake assay with live cell imaging (representative images shown). Mean ± SD. Kruskal–Wallis test with Dunn's multiple comparisons. $n = 5$ (887–1005 cells per condition). Scale bars = 50 μm. (D) Luminescent 2-DG uptake assay. Inhibitor treatments: SL327 (10 μM) and BAY876 (5 μM). Mean ± SD. One-way ANOVA with Tukey's post hoc test. $n = 8$. (E) qPCR of *GLUT1* and *GLUT3* genes. Controls and statistics were the same as indicated in (A). $n = 4$. (F) Representative membrane fractionation western blot of GLUT1. Cyto, TM, and PM represent cytoplasmic, total membrane, and plasma membrane fractions, respectively. $n = 5$. (G) Quantification of GLUT1 in the plasma membrane (PM) fractions of the membrane fractionation blot in (F). Mean ± SD. One-way ANOVA with Tukey's post hoc test. $n = 5$. (H) Lactate measurement in cell culture media (1-h incubation). Inhibitor treatments: 2-DG (2 mM), BAY876 (5 μM), SL327 (10 μM). Mean ± SD. KRAS^WT: Two-tailed unpaired $t$ tests; KRAS^G12V: one-way ANOVA with Tukey's post hoc tests. $n = 3$ for WT and $n = 4$ for G12V cell lines. (I) pH measurements of cell culture media after 5 days of culturing IM-HUVEC KRAS^G12V cells (−/+Dox). Inhibitor treatments: 2-DG (2 mM), BAY876 (5 μM), SL327 (10 μM). Scatter dot plot showing median pH value. One-way ANOVA with Tukey's post hoc test. $n = 3$. In this figure, *$P$ value < 0.05, ***$P$ value ≤ 0.0003 with exact $P$ values shown in Appendix Table S2. Source data are available online for this figure.

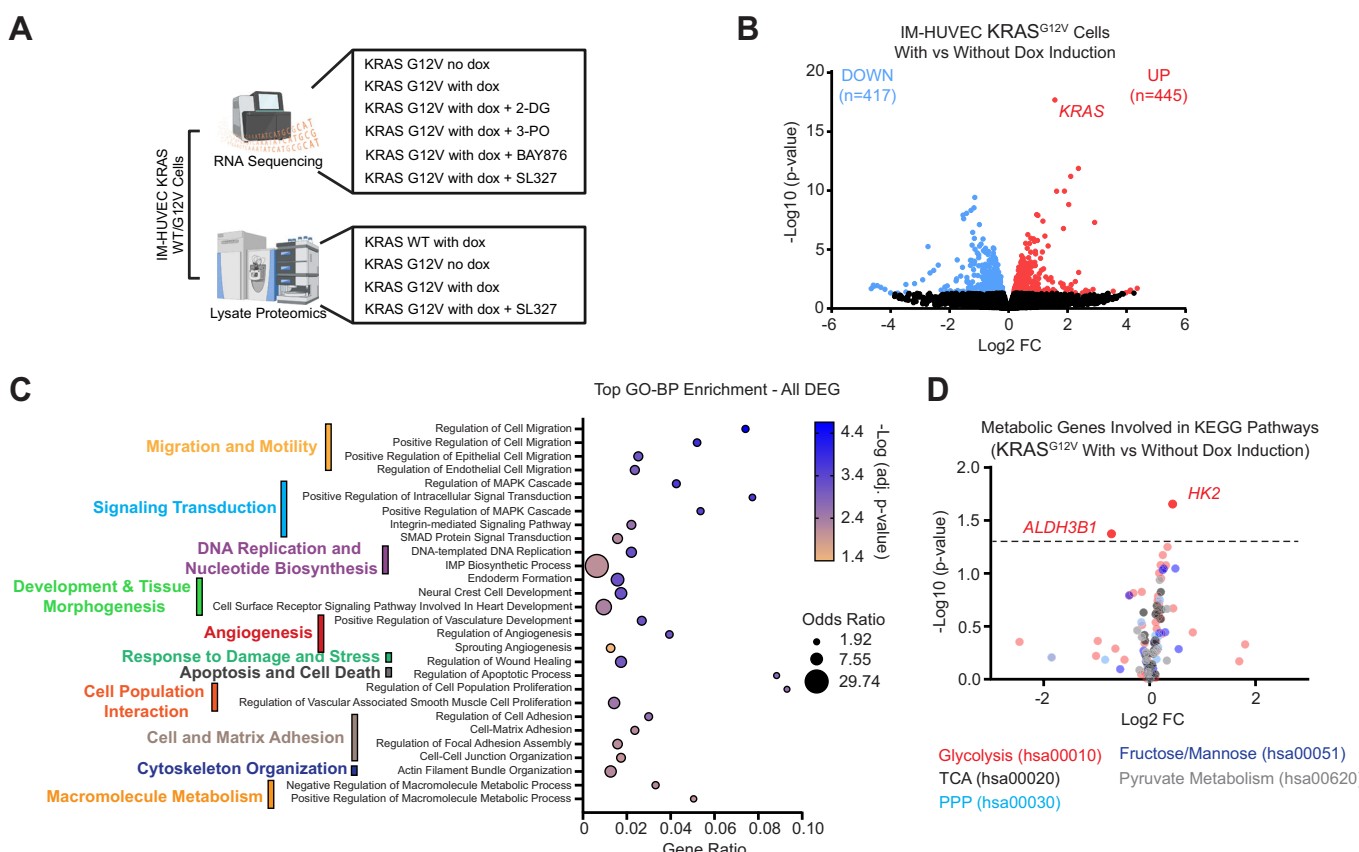

**Figure 3. Transcriptional profiling implicates HK2 as a mediator of glycolytic changes in mutant KRAS ECs.**

(A) Schematics of RNA-sequencing (seq) and proteomics experimental conditions. RNA-seq: $n = 3$. Proteomics: $n = 4$. (B) Volcano plots of differentially expressed genes (DEGs; $P$ value < 0.05, as calculated by DESeq2) between IM-HUVEC KRAS$^{G12V}$ cells with and without Dox induction of KRAS ($-/+$ Dox). $n = 3$. (C) Gene Ontology Biological Process (GO-BP) pathway enrichment using all DEGs between IM-HUVEC KRAS$^{G12V}$ cells ($-/+$ Dox). Enrichment was done using Enrichr, which computes Fisher's exact tests with Benjamini–Hochberg corrections for statistics. Adjusted $P$ value cutoff of <0.05 is considered significant. Color of the circle presents $-$log10 of adjusted $P$ value for pathway enrichment. The size of the circle represents the odds ratio. Gene ratio was calculated as the number of genes used to enrich for each GO term divided by the total number of genes used to enrich for all GO terms. (D) Volcano plot of DEGs ($P$ value < 0.05, as calculated by DESeq2) involved in five metabolic KEGG pathways between IM-HUVEC KRAS$^{G12V}$ cells ($-/+$ Dox). Each color represents genes in a specific metabolic KEGG pathway as labeled in the graph. $n = 3$. Source data are available online for this figure.

## Proteomics validates HK2 as a specific mediator of metabolic reprogramming in KRAS$^{G12V}$ ECs

Proteomics analyses corroborated the RNA-sequencing results. We captured a total of 6400 proteins with 5492 proteins identified from ≥2 unique peptides (Fig. 4A). Proteomics confirmed the upregulation of KRAS protein upon Dox induction (Appendix Fig. S6A). KRAS$^{G12V}$ expression ( + Dox) in ECs affected 650 and 719 proteins in comparison to KRAS$^{G12V}$ (no Dox) and KRAS$^{WT}$ ( + Dox) cell lines, respectively, while MEK inhibition (SL327) altered the expression of 1342 proteins (Fig. 4B). Major Gene Ontology enrichment themes from upregulated proteins included transcription, translation, ribosome biogenesis, RNA processing, DNA metabolism and chromosome organization. Conversely, KRAS$^{G12V}$ expression downregulated proteins related to cell junctions and adhesion, oxidative phosphorylation, plasma membrane organization, lipid metabolism, cytoskeleton, and vesicle-related terms. Many of the identified themes were inversely enriched with SL327 treatment, suggesting that these processes are MEK-dependent

(Fig. 4C; Appendix Fig. S6B). Detailed Gene Ontology enrichment terms are shown in Figs. S7 and S8.

Importantly, we captured 45/67 glycolysis proteins (listed in the KEGG pathway hsa00010) in our proteomics data (Fig. 4D,E). Among these, HK2 was again the only glycolysis protein upregulated with KRAS$^{G12V}$ expression (Dox vs. no Dox). Consistent with our transcriptome results, HK2 protein levels were decreased with SL327 treatment (Appendix Fig. S9A). We were also intrigued to find oxidative phosphorylation to be one of the top enriched terms among downregulated proteins between KRAS$^{G12V}$ ECs with and without Dox induction (Fig. 4C). Protein-protein interaction network analysis (STRING) showed that a cluster of tightly interacting proteins involved in oxidative phosphorylation was downregulated (Appendix Fig. S9B). One of the node proteins, PDHX, is a structural component of the pyruvate dehydrogenase (PDH) complex—the gatekeeper between glycolysis and the tricarboxylic acid cycle (Inoue et al, 2021). Western blot showed a decrease of PDHX levels in KRAS$^{G12V}$ ECs, confirming the mass spectrometry results (Appendix Fig. S9C). Downregulation of PDHX, along with other proteins involved in

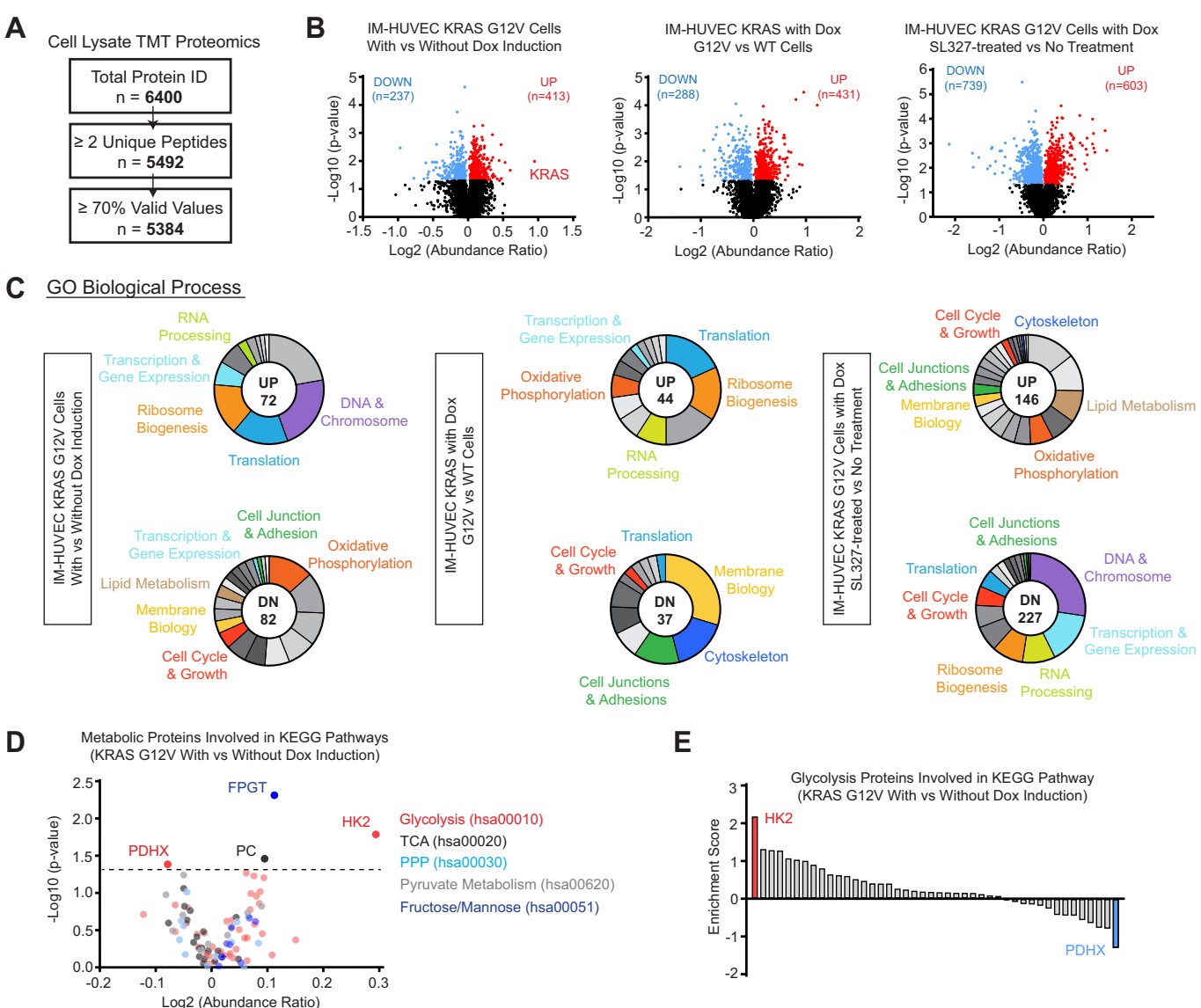

**Figure 4. Proteomics demonstrates that HK2 is a specific mediator of glycolytic changes in KRAS-mutant ECs.**

(A) Number of proteins identified from the TMT-labeled cell lysate proteomics runs. $n = 4$. (B) Volcano plots of differentially expressed proteins between various comparison groups. Significantly upregulated and downregulated proteins are shown in red and blue, respectively. Statistics are computed using Proteome Discoverer, which uses ANOVA hypothesis testing. $n = 4$. (C) Major enrichment themes among differentially expressed proteins between (1) IM-HUVEC KRAS G12V cells with vs without Dox induction; (2) IM-HUVEC KRAS G12V vs WT with Dox induction; and (3) IM-HUVEC KRAS G12V (+ Dox) with vs without 10 μM SL327 treatment. Themes were derived by categorizing all enriched Gene Ontology Biological Process (GO-BP) terms (adj $P$ value of <0.05) from Enrichr (Fisher exact tests with Benjamini–Hochberg corrections for statistics). The number inside each pie chart represents the total number of GO-BP terms passing the adjusted $P$ value cutoff. UP and DN represented terms enriched by upregulated and downregulated proteins, respectively. Selected themes were labeled. (D) Volcano plot of differentially expressed proteins (ANOVA tests) involved in five metabolic KEGG pathways between IM-HUVEC KRAS G12V cells (−/+ Dox). Each color represents proteins in a specific metabolic KEGG pathway as labeled in the graph. $n = 4$. (E) Glycolysis proteins in the KEGG hsa00010 pathway were ranked based on enrichment score with upregulated proteins in red (positive) and downregulated proteins in blue (negative) comparing KRAS G12V ECs (−/+ Dox). Differentially expressed proteins with $P$ value < 0.05 (ANOVA tests) were colored in and labeled. Source data are available online for this figure.

oxidative phosphorylation, may explain why the increase in glycolytic flux is not accompanied by a corresponding increase in mitochondrial respiration.

Overall, both transcriptome- and proteome-level analyses positioned HK2 as a key mediator of glycolytic reprogramming in KRAS$^{G12V}$-expressing ECs, which may be implicated in sporadic AVM pathogenesis.

## Validating glycolytic phenotypes in human primary brain microvascular ECs expressing KRAS$^{G12V}$ and human bAVM tissues

To confirm our findings in the cellular context of primary brain ECs, we first transiently electroporated mScarlet-tagged *KRAS$^{WT}$* or *KRAS$^{G12V}$* plasmids into primary human brain microvascular

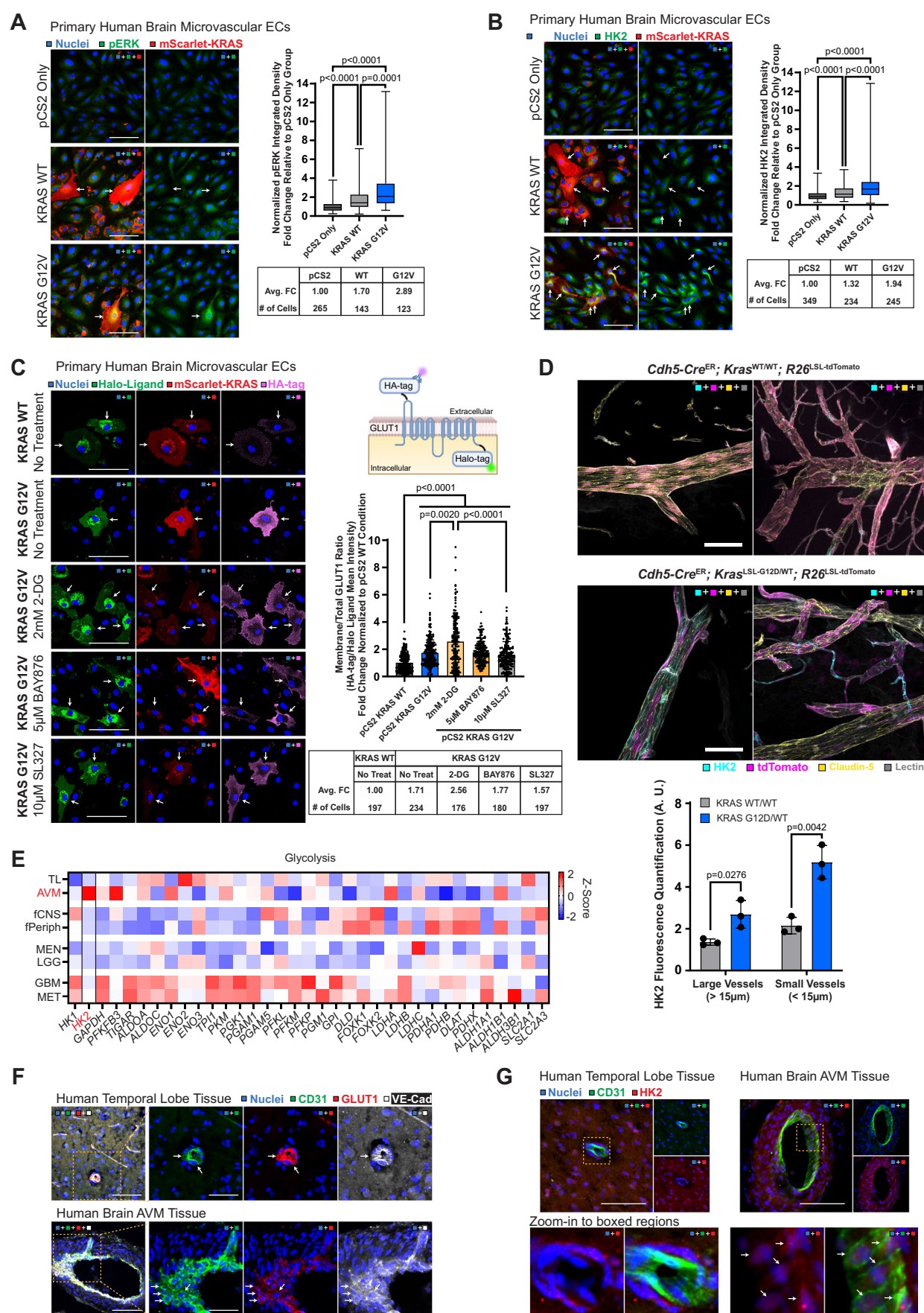

**A** Primary Human Brain Microvascular ECs

**B** Primary Human Brain Microvascular ECs

**C** Primary Human Brain Microvascular ECs

**D** *Cdh5-Cre*[ER]; *Kras*[WT/WT]; *R26*[LSL-tdTomato]

*Cdh5-Cre*[ER]; *Kras*[LSL-G12D/WT]; *R26*[LSL-tdTomato]

HK2 | tdTomato | Claudin-5 | Lectin

**E** Glycolysis

**F** Human Temporal Lobe Tissue

Human Brain AVM Tissue

**G** Human Temporal Lobe Tissue    Human Brain AVM Tissue

Zoom-in to boxed regions

◄

**Figure 5. Validation of HK2 upregulation and GLUT1 membrane localization in human brain microvascular ECs, mouse, and human bAVM tissue.**

(A) Representative IF staining and quantification of phospho-ERK1/2 (pERK; Thr202/Tyr204) in primary human brain microvascular ECs (hBMVECs) electroporated with control and *KRAS* constructs. Arrows indicated examples of transfected (mScarlet-positive) ECs. Average fold changes (FC) and number of cells quantified are summarized in the table below. Box and whiskers plot: min to max. Kruskal–Wallis test with Dunn's multiple comparisons. $n = 3$. Scale bars = 100 μm. (B) Representative IF staining and quantification of HK2 in primary hBMVECs electroporated with control and *KRAS* constructs. Analysis and statistics were done the same as (A). $n = 4$. Scale bars = 100 μm. (C) Representative IF staining and quantification of primary hBMVECs co-electroporated with HA-GLUT1-Halo and *KRAS* constructs. A schematic of the HA-GLUT1-Halo construct is shown above the graph. Average FC and number of cells quantified are summarized in the table below. Mean ± SD. Kruskal–Wallis test with Dunn's multiple comparisons. $n = 4$. Scale bars = 100 μm. (D) Brain sections from *Cdh5*-CreERT2; *Kras*^WT/WT^; *R26*^LSL-tdTomato^ and *Cdh5*-CreERT2; *Kras*^LSL-G12D/WT^; *R26*^LSL-tdTomato^ were stained with anti-HK2 (cyan) two months after tamoxifen treatment. LSL, lox-STOP-lox. Magenta indicated cells in which the tdTomato reporter was expressed, indicating Cre-mediated recombination. Claudin 5 (yellow) was stained to indicate the brain endothelium, and mice were perfused with lectin (gray). Scale bars = 50 μm. Quantification of HK2 fluorescence intensity (A.U.) is shown below. Vessels were divided into large vessels (diameter >15 μm) and small vessels (diameter <15 μm) for analysis. Mean ± SD. Unpaired multiple *t* tests. $n = 3$ animals per group. (E) Heatmap generated from published single-cell RNA-sequencing dataset by data ref: Wälchli et al, 2024. Z-score normalized expression of Glycolysis pathway component and related genes were compared in ECs isolated from human adult temporal lobe (TL), fetal brain (fCNS), fetal peripheral tissue (fPeriph) and various brain pathologies, including arteriovenous malformations (AVM), meningioma (MEN), low-grade glioma (LGG), glioblastoma (GBM) and lung cancer metastasized to the brain (MET). (F) Representative IF images of GLUT1 and VE-cadherin in human temporal lobe ($n = 2$) and brain AVM tissues ($n = 2$). Arrows show examples of junctional CD31, GLUT1, and VE-cadherin in the temporal lobe and bAVM tissue. Scale bars = 100 μm and 50 μm in the zoom-in images. (G) Representative IF images of HK2 in human temporal lobe ($n = 2$) and brain AVM tissues ($n = 2$). Arrows show examples of intracellular HK2 with zoomed-in images of boxed regions shown below. Scale bars = 100 μm. Source data are available online for this figure.

endothelial cells (phBMVEC). *KRAS*^WT^ and *KRAS*^G12V^ plasmids had similar electroporation efficiencies as assessed by mScarlet signal intensity using confocal microscopy (Appendix Fig. S10A). An empty plasmid vector (pCS2 only) was included as an additional control in all experiments. Immunostaining and confocal imaging showed that phBMVECs electroporated with *KRAS*^G12V^ constructs featured increased pERK levels and HK2 expression compared to cells receiving *KRAS*^WT^ constructs or empty vectors (Fig. 5A,B). To test if there was increased membrane localization of GLUT1 in phBMVECs, we utilized an engineered human GLUT1 construct with an extracellular HA-tag and an intracellular Halo-tag (Yazdani et al, 2022). We co-transfected this GLUT1 construct with our *KRAS* plasmids into phBMVECs and used anti-HA antibody and cell-permeable Halo-tag ligand to label HA and Halo tags, respectively (without permeabilization of the cells). A signal for HA will only be observed when HA-GLUT1 is localized to the membrane. By taking the signal ratio of the HA and Halo-tags, we were able to assess the amount of exogenous GLUT1 anchored in the plasma membrane versus the total amount of GLUT1 transfected into each cell. Analysis showed that brain ECs co-electroporated with mutant KRAS had a higher amount of GLUT1 membrane localization, compared to ECs co-electroporated with KRAS^WT^ (Fig. 5C). 2-DG treatment significantly elevated GLUT1 membrane localization, potentially as a compensatory mechanism to strong glycolytic suppression. BAY876 and SL327 treatments did not significantly alter GLUT1 trafficking to the plasma membrane. Interestingly, we observed that intracellular KRAS protein appeared to be enclosed within GLUT1^+^ vesicular structures (Appendix Fig. S10B). KRAS and GLUT1 also appeared to be co-transported and co-localized at the plasma membrane, suggesting close interactions between KRAS and GLUT1 (Appendix Fig. S10C,D).

To determine whether glycolysis might be modulated in a mouse model of bAVM (Fish et al, 2020), we assessed HK2 expression by immunofluorescence in adult mice with inducible endothelial-specific KRAS^G12D^ expression, driven by *Cdh5*-CreERT2-mediated recombination. This revealed that HK2 expression was indeed increased in bAVM vessels of mutant *Kras* mice, in comparison to vessels from the same brain region of *Kras*^WT^ animals (Fig. 5D).

We next extended these observations to human bAVM tissue samples. Wälchli and Ghobrial et al recently published a single-cell

RNA-sequencing dataset of human brain vasculature across developmental and pathological conditions (Wälchli et al, 2024). In comparison to tumor ECs, bAVM ECs had relatively lower expression of Ras/MEK/ERK signaling components, yet there was a strong MAPK/ERK signaling activation signature (Appendix Fig. S11), further highlighting the importance of this pathway in AVM pathogenesis. When we examined glycolysis genes, bAVM ECs had elevated levels of *HK2* and *PFKFB3* and downregulation of *PDHX*, while tumor ECs, particularly aggressive tumors, such as lung cancer metastasized to the brain (MET) and glioblastoma (GBM) had a distinct glycolytic signature where *HK1* was predominantly upregulated, along with many other glycolytic components (Fig. 5E) (data ref: Wälchli et al, 2024).

We then performed tissue immunostaining on human bAVM samples, using human temporal lobe sections as controls. Although CNS-specific transcriptional programs are downregulated in bAVM ECs (Wälchli et al, 2024), we observed membrane localization of GLUT1 in bAVM tissue as indicated by co-staining for the junctional protein CD31 (Fig. 5F). It is worth highlighting that VE-cadherin expression was weak and diffuse in bAVM tissue compared to temporal lobe tissue, where it was localized to cell junctions. This was unlikely to be caused by technical factors, such as tissue quality, as CD31 and tight junction protein, ZO-1, staining at the junctions was not impacted (Appendix Fig. S10E). Thus, adherens junctions are disrupted in clinical bAVM tissue, which matches our previous in vitro observations (Fish et al, 2020; Nikolaev et al, 2018). Yet, despite the loss of adherens junctions, GLUT1 was still localized to the membrane in bAVM tissue. Additionally, endothelial HK2 expression was elevated in the endothelium of bAVM vessels compared to temporal lobe vessels, as shown by IF (Fig. 5G). This direct evidence from human tissue further strengthens the potential importance of HK2 in modulating glycolytic changes in sporadic AVMs.

## EC glycolysis as a therapeutic vulnerability for sporadic AVM

While there are several therapeutic candidates currently under investigation for sporadic AVMs (Mansur and Radovanovic, 2024), there are no approved pharmacological treatments for this disease. To test if attenuating glycolysis is beneficial for AVM treatment, we first

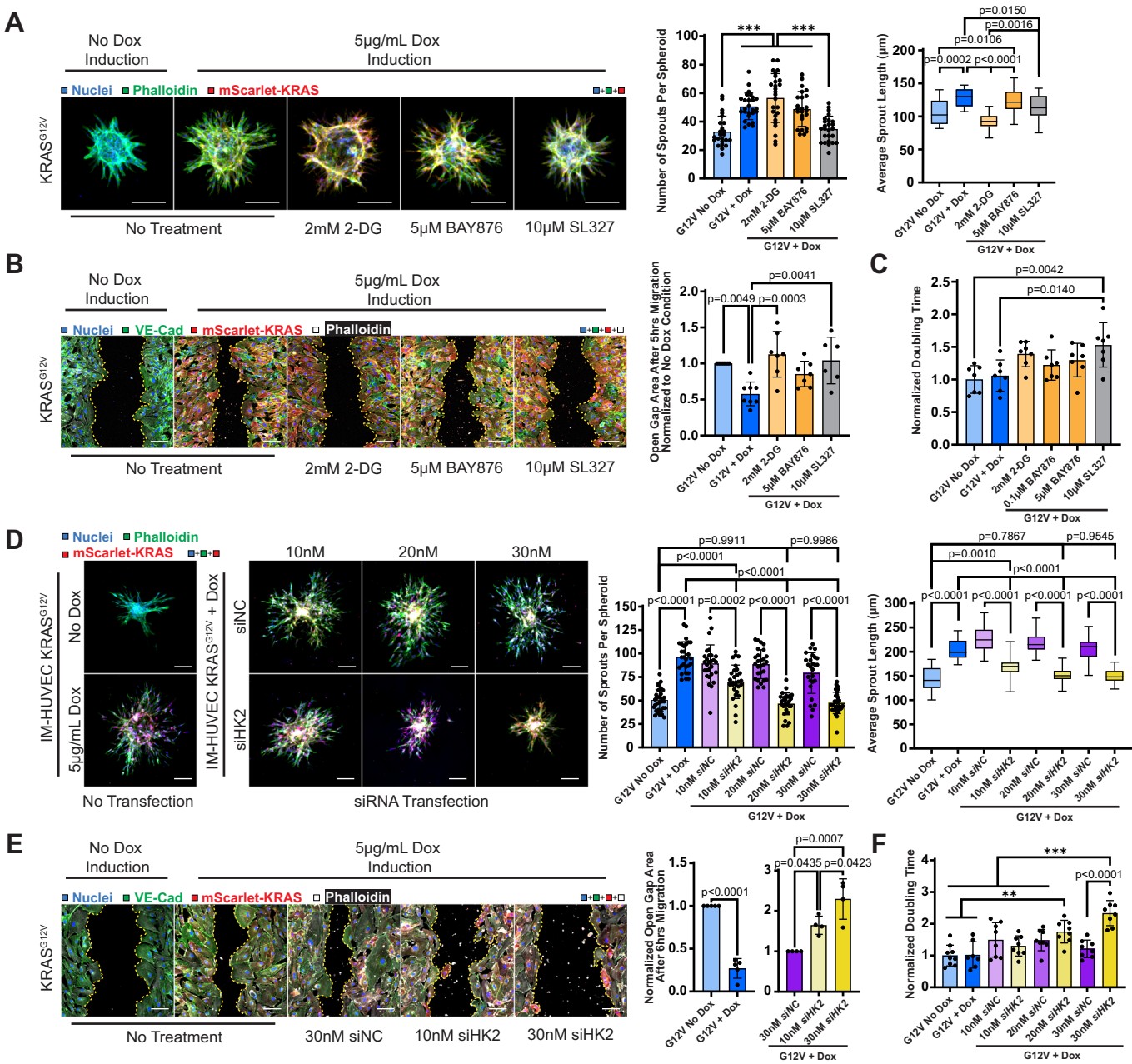

**Figure 6. Functional effects of glycolytic suppression using inhibitors or HK2 knockdown.**

Experiments were conducted with IM-HUVEC KRAS$^{G12V}$ cell line ($-$/$+$ Dox induction). (**A**) Representative spheroid sprouting assay (24 h) with inhibitor treatments. Bar graph: mean ± SD. Box and whiskers: min to max. One-way ANOVA with Tukey's post hoc tests. $n = 4$ (5–7 spheroids per condition per replicate). Scale bars = 200 µm. (**B**) Representative image of migration assay with inhibitor treatments. Yellow dotted lines highlight cell boundaries after migration. Mean ± SD. One-way ANOVA with Tukey's post hoc tests. $n = 6$–8. Scale bars = 100 µm. (**C**) Real-time xCELLigence impedance recordings with inhibitor treatments. Mean ± SD. One-way ANOVA with Tukey's post hoc tests. $n = 7$. (**D**) Representative images of the spheroid sprouting assay with siRNA-mediated *HK2* (siHK2) knockdown. siNC = negative control siRNA. Quantification and statistics are the same as (**A**). $n = 4$ (5–9 spheroids per condition per replicate). Scale bars = 200 µm. (**E**) Representative image of migration assay with siHK2 knockdown. Yellow dotted lines highlight cell boundaries after migration. Quantification is the same as (**B**). Two-tailed unpaired $t$ test or one-way ANOVA with Tukey's post hoc tests. $n = 4$–5. Scale bars = 100 µm. (**F**) Real-time xCELLigence impedance recordings with siHK2 knockdown. Quantification and statistics are the same as (**C**). $n = 7$–9. In this figure, **P value < 0.01, ***P value < 0.001 with exact P values shown in Appendix Table S2. Source data are available online for this figure.

examined the functional response of ECs expressing KRAS$^{G12V}$ to glycolysis inhibition. Treatment of KRAS$^{G12V}$ ECs with 2-DG reduced excessive angiogenic sprouting and migration (Fig. 6A,B). While there was a trend towards an effect for BAY876 on these processes, this did

not reach statistical significance. 2-DG and BAY876 did not significantly alter the doubling time of ECs as measured by real-time xCELLigence impedance assays, indicating that cell viability/proliferation was not affected (Fig. 6C). In agreement with previous data (Fish

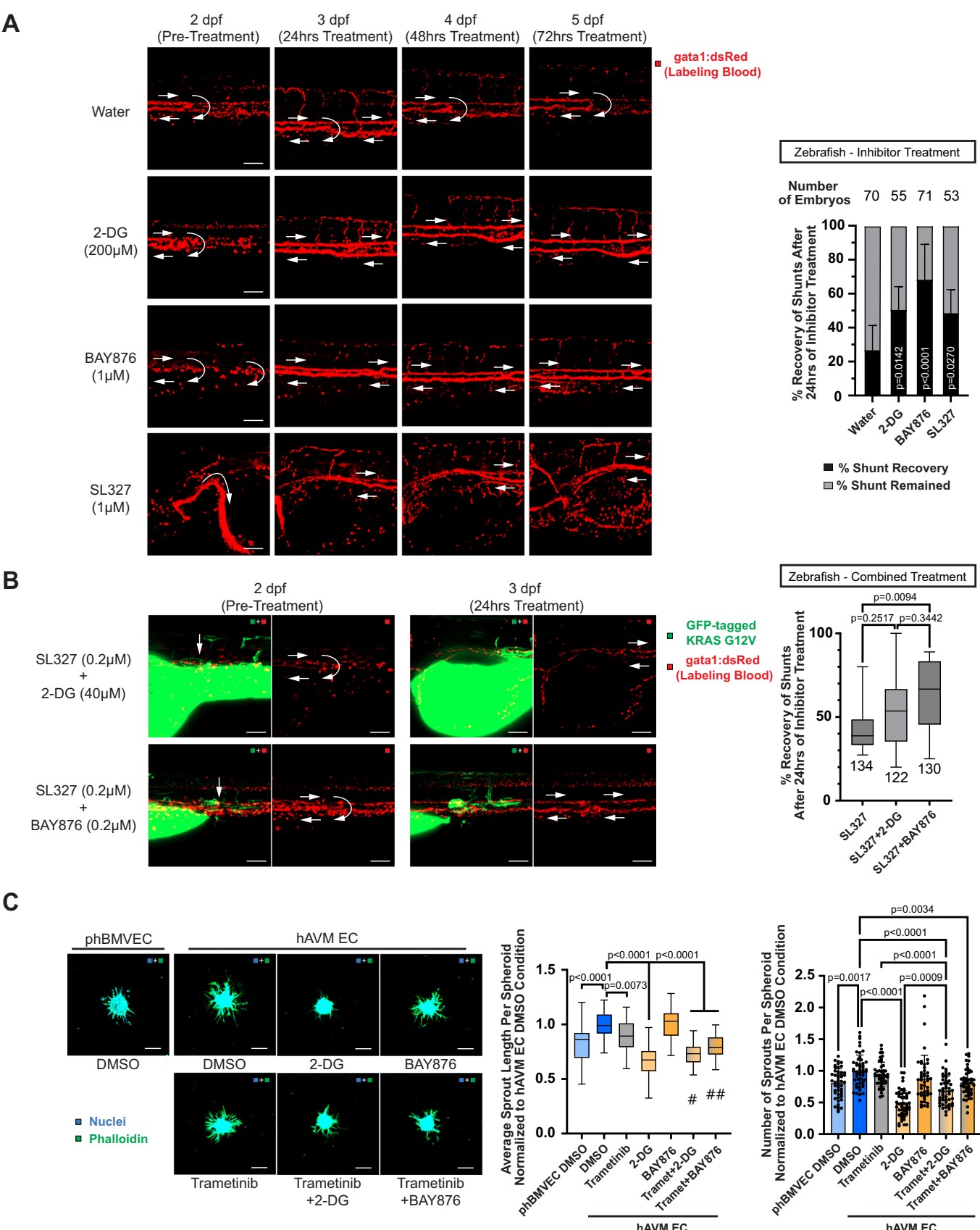

◄

**Figure 7.   Inhibition of glycolysis restores abnormal AVM phenotypes in zebrafish and human bAVM-isolated cells.**

(A) $Tg(gata1$:dsRed) zebrafish injected with GFP-tagged $KRAS^{G12V}$ constructs and imaged at 2 days-post-fertilization (dpf). Inhibitor treatments were initiated at 2 dpf (dosed in water) after confirming the presence of shunts. Zebrafish were imaged after 24–72 h of inhibitor treatment (representative images shown; left panels). Arrows indicate directions of blood flow. Scale bars = 100 μm. Shunt recovery quantification after 24 h of inhibitor treatments by a blinded researcher is shown on the right (right panel). Mean ± SD. One-way ANOVA with Tukey's post hoc tests. Statistical tests are in comparison to water treatment control. $n = 8$–11 independent replicates. The total number of embryos per condition is indicated at the top of the bar graph. (B) $Tg(gata1$:dsRed) zebrafish injected with GFP-tagged $KRAS^{G12V}$ constructs and imaged at 2 dpf. Inhibitor treatments were initiated at 2 dpf (dosed in water) after confirming the presence of arteriovenous shunts. Zebrafish were imaged before and after inhibitor treatment (representative images shown; left panels). Arrows indicate locations of shunts in the images and directions of blood flow. Scale bars = 100 μm. Quantifications of zebrafish shunt recovery with combined treatments are shown on the right. Numbers on the graph indicate the total number of zebrafish embryos analyzed for each treatment condition. Box and whiskers: min to max. One-way ANOVA with Tukey's post hoc test. $n = 14$–16 independent experiments. (C) Representative images of spheroid sprouting (24 h) of clinically-isolated bAVM ECs and commercially-purchased hBMVECs. Inhibitor treatments: 2-DG (2 mM), BAY876 (5 μM), trametinib (100 nM). Bar graph: mean ± SD. Box and whiskers: min to max. One-way ANOVA with Tukey's post hoc test. Only selected statistics are shown. # represents a significant $P$ value comparing 2-DG+Trametinib treatment with Trametinib treatment alone. ## represents significant $P$ values comparing BAY876+Trametinib treatment with both Trametinib or BAY876 treatment alone. Exact $P$ values are shown in Appendix Table S2. $n = 4$ patients for bAVM ECs and independent replicates (9–17 spheroids per condition per replicate). Scale bars = 200 μm. Source data are available online for this figure.

et al, 2020; Nikolaev et al, 2018), MEK inhibition reduced angiogenic sprouting and migration, but also significantly increased doubling time (Fig. 6A–C). We next performed the same assays following siRNA knockdown of *HK2*. Knockdown efficiency of *HK2* was confirmed (Fig. EV4). There were no effects on *HK1* levels or phosphorylation of ERK upon *HK2* knockdown (Fig. EV4A). siRNA knockdown of *HK2* resulted in a dose-dependent reduction of both average sprout length and number per EC spheroid (Fig. 6D). Knockdown of *HK2* also blunted the enhanced migratory behavior of mutant KRAS ECs (Fig. 6E). Higher concentrations of *HK2* siRNA did impact cell proliferation (Fig. 6F). We also examined the effect of glycolytic suppression on EC junctions (Appendix Fig. S12). 2-DG treatment appeared to improve monolayer and junctional organization, despite overall lower junctional VE-cadherin staining (Appendix Fig. S12A). Furthermore, western blot analysis revealed alterations to the molecular weight of VE-cadherin (Appendix Fig. S12D,E), perhaps indicating modulation of protein processing (Kurtoglu et al, 2007). BAY876 treatment did not restore VE-cadherin junctions (Appendix Fig. S12B). Finally, *HK2* knockdown improved monolayer and junctional organization in mutant KRAS ECs (Appendix Fig. S12C). While junctional staining appeared to be reduced, *HK2* knockdown did not impact the molecular weight of VE-cadherin (Appendix Fig. S12F). Overall, modulating EC glycolysis reversed many of the functional alterations caused by mutant KRAS signaling, suggesting its therapeutic potential in treating sporadic AVMs.

To translate our findings in vivo, we next turned to our established KRAS-driven AVM zebrafish animal model (Fish et al, 2020). Consistent with our previous report (Fish et al, 2020), injection of a Tol2 transgene driving endothelial-specific expression of fluorescently tagged KRAS$^{G12V}$ into 1–4 cell stage zebrafish embryos resulted in the formation of abnormally enlarged vessels and direct arteriovenous shunts by 36 h-post-fertilization (hpf) (Fig. 7A; Appendix Fig. S13). In embryos with established shunts, 24 h of treatment (36–60 hpf) with either glycolysis inhibitors (2-DG, BAY876) or MEK inhibitor (SL327) improved shunt recovery compared to untreated controls without impacting overall development (Fig. 7A; Appendix Fig. S13). To determine whether there might be a synergy between glycolysis and MEK inhibition, we tested combined treatments. We utilized a fivefold lower dose of 2-DG, BAY876, and SL327 compared to single inhibitor treatment. This experiment revealed that combined BAY876 + SL327 significantly improved shunt reversal compared to SL327 alone in our zebrafish model, and a similar trend (though not statistically significant) was seen with 2-DG + SL327 (Fig. 7B).

We next isolated primary ECs from resected clinical bAVM samples and performed an EC spheroid assay to test if glycolysis inhibitors could attenuate angiogenic sprouting in these pathological primary brain ECs. In this experiment, we used an FDA-approved MEK inhibitor, trametinib, which is currently under investigation for the treatment of bAVM (Mansur and Radovanovic, 2024). We observed that trametinib is more potent at inhibiting MAPK/ERK signaling than SL327 and may lead to compensatory AKT activation (Fig. EV5A). In comparison to phBMVEC from healthy donors, human bAVM ECs showed enhanced angiogenic potential with increased average sprout length and number per spheroid (Fig. 7C). Similar to the IM-HUVEC data (Fig. 6A), 2-DG treatment alone potently reduced sprouting angiogenesis of bAVM ECs, while BAY876 alone did not significantly impact sprouting behaviors. Intriguingly, ECs from two patients with *KRAS* mutations appeared to respond more strongly to trametinib than those without, yet combination treatment with BAY876 and trametinib reduced sprout length in all patient-derived EC samples (Figs. 7C and EV5B). Furthermore, we also observed that combined treatment of MEK inhibitors (SL327 or trametinib) and 2-DG was able to restore VE-cadherin junctions and improve overall monolayer organization in our mutant KRAS EC model (Fig. EV5C), indicating another potential benefit of combinatorial treatment.

Taken together, glycolytic inhibition, either by limiting glucose uptake or directly targeting HK2, effectively reversed pathological EC behaviors driven by mutant KRAS and may synergize with MEK inhibition.

# Discussion

Sporadic AVM is a devastating disease with a broad spectrum of negative consequences on the health and quality of life of patients, including facial deformity impacting vision and hearing for extracranial AVMs (Al-Olabi et al, 2018; Kim et al, 2017) and hemorrhagic stroke in the case of bAVM (Solomon and Connolly, 2017). The absence of approved non-surgical interventional strategies is largely due to an immense gap in knowledge regarding the underlying molecular pathogenesis of this disease. However, with the identification of *KRAS* as a genetic driver in most cases of sporadic bAVM (Nikolaev et al, 2018), and subsequent animal model development (Fish et al, 2020), targeted therapies are on the

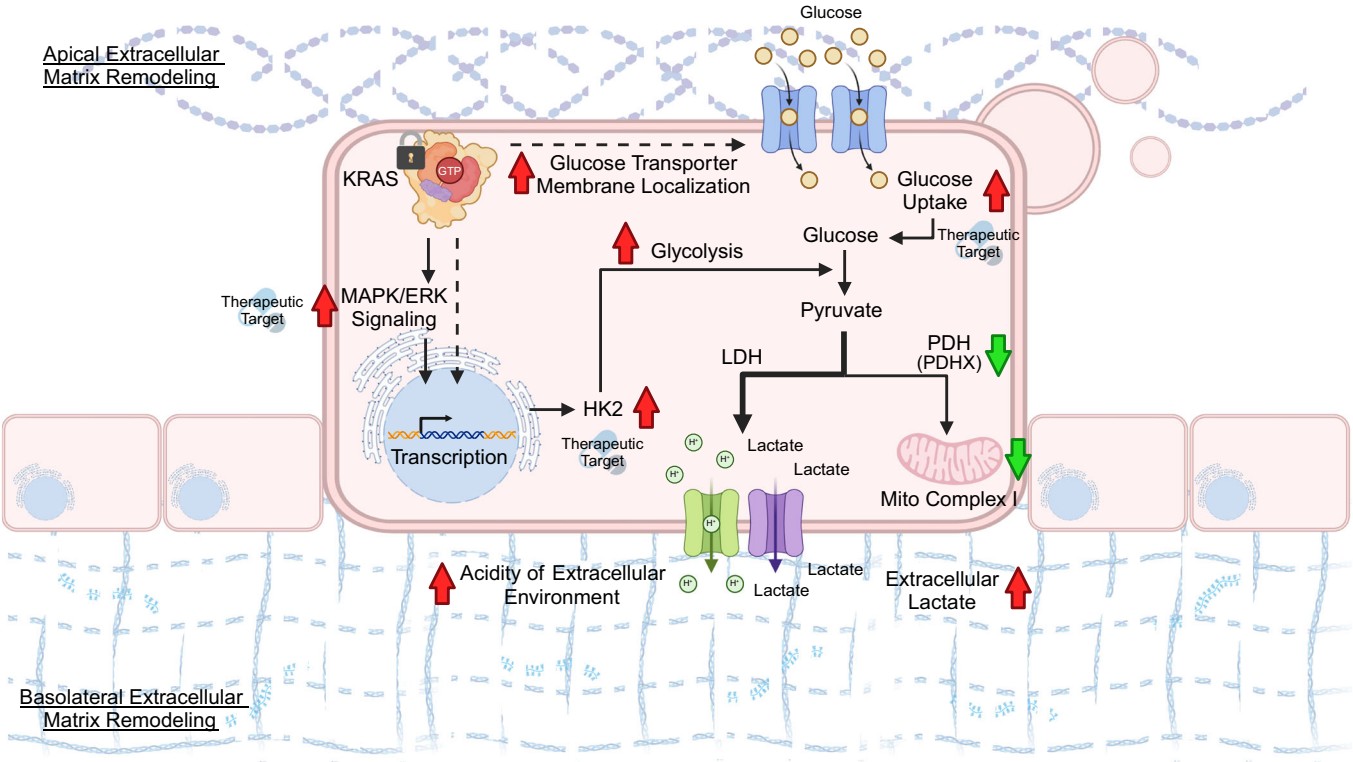

**Figure 8. KRAS induces metabolic reprogramming in endothelial cells.**

Graphical summary of glycolytic alterations in ECs expressing mutant KRAS proteins (G12V). In mutant KRAS ECs, there is elevated glucose uptake due to increased levels of glucose transporters at the plasma membrane. This is accompanied by the upregulation of a key glycolytic enzyme, HK2, which collectively leads to a higher glycolytic flux and the secretion of downstream products (protons and lactate) into the extracellular environment. This shift towards glycolysis contributes to pathological behaviors of ECs in the context of sporadic AVMs, including enhanced angiogenic sprouting and migratory phenotypes. The secretion of glycolytic products may alter the extracellular environment surrounding mutant KRAS ECs, which could contribute to extracellular matrix remodeling and phenotypic modulations of neighboring cells. Mechanistically, HK2 expression was found to be dependent on MAPK/ERK signaling (solid line). However, ERK activation alone was insufficient to promote HK2 upregulation, indicating that mutant KRAS is also modulating HK2 expression through additional mechanisms (dotted line). Targeting MAPK/ERK signaling or glycolysis provides therapeutic benefits in restoring pathological changes driven by mutant KRAS, and combined suppression may lead to synergistic benefits.

horizon. In the current study, we used a human EC model with inducible mutant KRAS[G12V] expression and profiled the transcriptional, proteomic, and functional consequences on EC dynamics downstream of mutant KRAS signaling. In particular, we closely investigated the metabolic changes driving enhanced glycolysis, demonstrating the presence of the Warburg Effect in mutant KRAS ECs (Fig. 8). We showed that this metabolic adaptation was primarily driven by increased membrane localization of glucose transporters, particularly GLUT1, and enhanced HK2 expression, resulting in elevated glycolytic flux. The induction of HK2 expression was confirmed in primary brain ECs, and mouse and human bAVM tissues. Notably, there appears to be a distinct glycolytic signature in human bAVMs compared to ECs in aggressive tumors, which predominantly feature HK1 upregulation. Interestingly, recent case reports demonstrated that high-flow AVMs have remarkably elevated 18-F-fluorodeoxyglucose uptake observed by positron emission tomography/computed tomography imaging, providing direct evidence of increased glucose uptake within an AVM nidus (Gungor and Yakar 2021; Quddus et al, 2021). Therapeutically, we confirmed that targeting EC glycolysis had functional benefits by reducing excessive angiogenic and migratory behaviors in mutant KRAS ECs. We further validated the

therapeutic efficacy of glycolytic suppression using an in vivo AVM zebrafish model and primary ECs isolated from bAVM patients, which revealed a potential synergy with MEK inhibition.

Aside from surgical procedures, much of the therapeutic development efforts for sporadic AVMs have focused on re-purposing existing drugs used in oncology and other settings. Some of the promising candidates include trametinib (MEK inhibitor) (Cooke et al, 2021; Nguyen et al, 2023; Suarez et al, 2024), thalidomide (angiogenesis inhibitor) (Boon et al, 2022), bevacizu-mab (antibody against VEGF) (Seebauer et al, 2024), sirolimus (mTOR inhibitor) (Hammer et al, 2018) and pazopanib (multi-kinase inhibitor) (Moon et al, 2022). While small cohort studies have shown encouraging clinical outcomes for some of these inhibitors, patient response to treatment has been highly variable (Al-Samkari and Eng, 2022). This may in part be due to differences in underlying causative genetic variants among patient cohorts. Some inhibitors, such as sirolimus, showed efficacy for slow-flow vascular anomalies (e.g., venous and lymphatic malformations), but unsatisfactory outcomes for sporadic AVMs (Triana et al, 2017). Experience from the oncology field has also raised concerns for drug resistance when targeting some of these pathways (i.e., VEGF, BRAF, etc.) through activation of compensatory signaling and

feedback mechanisms (Krebs et al, 2023; Wee et al, 2009). We have observed evidence of this in the current study. While trametinib inhibits ERK signaling more potently than SL327, it also elicited greater activation of the PI3K/AKT pathway (Fig. EV5A). This argues for the utilization of combination therapy in AVMs, much like in oncology. Recent advancement in the development of mutant KRAS-specific inhibitors (Lanman et al, 2020; Ou et al, 2022; Wang et al, 2022) marks another potentially exciting therapeutic breakthrough for sporadic AVMs (Fraissenon et al, 2024). However, most compounds are still under clinical testing for oncology applications, and it is unclear if they will provide benefits for sporadic AVMs. Sotorasib (Lanman et al, 2020) and Adagrasib (Ou et al, 2022), two KRAS inhibitors with accelerated approvals from the United States Food and Drug Agency, both target the G12C mutation, but this mutation is extremely rare in sporadic AVMs (Al-Olabi et al, 2018; Fraissenon et al, 2024; Hong et al, 2019; Priemer et al, 2019). In addition, reports of drug resistance and patient relapse are already emerging for these novel KRAS-directed therapeutics in cancer (Awad et al, 2021; Koga et al, 2021).

Here, we showed that targeting EC glycolysis ameliorates pathological changes in mutant KRAS ECs without impacting upstream signaling activation, as both glycolysis inhibitors and HK2 knockdown had no effect on ERK phosphorylation levels (Fig. EV4A; Appendix Figs. S5A and S12E,F). This might be advantageous for preventing drug resistance and may have broad applications to diseases featuring AVMs with different underlying genetic mutations. Additionally, our data highlighted HK2 as a specific mediator of metabolic reprogramming in mutant KRAS ECs, identifying this as a potential therapeutic target in sporadic AVMs. There are two approaches to modulate HK2: (1) attenuate its activity through chemical inhibitors, and (2) reduce HK2 expression. Throughout our study, 2-DG potently inhibited glycolysis in the presence of normal glucose levels. 2-DG has been tested in clinical trials for different applications, including a recent trial for COVID-19, demonstrating therapeutic potential (Bhatt et al, 2022). Importantly, 2-DG is capable of crossing the blood-brain barrier (Priebe et al, 2018), which is a favorable drug property for central nervous system diseases. One caveat with 2-DG, however, is its potent inhibition of protein glycosylation due to its structural similarity with D-mannose (Kurtoglu et al, 2007). In ECs, we found that the top pathways enriched after 2-DG treatment were N-linked glycosylation, protein processing, and the unfolded protein response (Fig. EV3B). Interestingly, combined treatment of MEK inhibitor with 2-DG seemed to alleviate the effect of 2-DG on VE-cadherin junctions (Fig. EV5C). Further molecular studies are needed to understand the molecular mechanisms underlying these changes. Alternatively, we demonstrated that directly lowering HK2 expression provided potent therapeutic benefits in vitro, without affecting protein processing.

Lastly, our work identified potential synergistic benefits when inhibition of glucose uptake (using BAY876) and MAPK/ERK activity was combined using low doses of inhibitors. While the MAPK/ERK pathway is a major signaling axis dysregulated in sporadic AVMs, it is not the sole mechanism driving pathogenesis. There are both MEK-dependent and MEK-independent changes in mutant KRAS ECs (Fish et al, 2020). For example, VE-cadherin junctional integrity is a MEK-dependent phenotype that can be rescued with various MEK/ERK inhibitors. Conversely, mutant KRAS-driven GLUT1 membrane localization is not affected by SL327 treatment and is thus a MEK-independent phenotype (Fig. 5C). BAY876, a GLUT1 inhibitor,

potently reduced glucose uptake, lactate, and proton secretion in mutant KRAS ECs, potentially compensating for some of the pathological alterations not fully rescued by MEK inhibition. Indeed, our studies using patient-derived bAVM ECs suggested that samples without known KRAS mutations did not respond to MEK inhibition alone, while all samples responded well to combined glycolysis and MEK inhibition. However, this result should be interpreted with caution, given the low number of samples. Whether a combination of glycolysis and MEK inhibition will be efficacious in other vascular anomalies that also feature precocious angiogenesis (Girard et al, 2018; Kobialka et al, 2022) remains to be seen, but it is an exciting possibility.

Taken together, our study pinpointed EC glycolysis as a therapeutic vulnerability capable of restoring pathological abnormalities downstream of mutant KRAS signaling in sporadic AVM. Dual-targeting of EC glycolysis and the MAPK/ERK pathway provided combinatorial benefits, as evident using patient-derived ECs and a zebrafish model, paving the way for further in vivo and clinical testing.

## Limitations

The primary mechanistic insights in our study were made using an immortalized HUVEC cellular system, with validation in primary brain ECs. The absence of fluid flow and perivascular supportive cells is a limitation of this model system. Further studies using a larger number of bAVM patient-derived primary ECs with various genetic mutations will be needed to determine whether glycolysis and MEK inhibition are broadly able to suppress angiogenic phenotypes. This approach should also be assessed in other types of vascular anomalies that feature excessive angiogenesis, such as Hereditary Hemorrhagic Telangiectasia. This will reveal whether this therapeutic approach may be widely applicable to AVM patients with different underlying genetic causes. AVM treatments are typically initiated after years of disease progression with extensive vascular remodeling. Our studies assessing the reversal of arteriovenous shunts in the zebrafish represent a very early stage of disease, which may be easier to restore than advanced disease. The mechanisms of shunt reversal remain unknown. We also used SL327 for these zebrafish studies, rather than the more clinically relevant MEK inhibitor, trametinib. Hence, the effectiveness of dual glycolysis/MEK inhibition in reversing pathology should be validated in animal models with more established and complex AVM lesions using clinically relevant drugs and doses. Encouragingly, HK2 is upregulated in a mouse model of KRAS-dependent bAVM, which will facilitate these studies.

## Methods

**Reagents and tools table**

| Reagent/resource | Reference or source | Identifier or catalog number |
| --- | --- | --- |
| **Experimental models** | | |
| Immortalized Human Umbilical Vein Endothelial Cells | PMID: 27842057 | N/A |
| Immortalized Human Umbilical Vein Endothelial Cells – F-tractin Labeled | PMID: 27842057 | N/A |
| Immortalized Human Umbilical Vein Endothelial Cells – Dox-inducible KRAS WT | Current study | N/A |

| Reagent/resource | Reference or source | Identifier or catalog number |
| --- | --- | --- |
| Immortalized Human Umbilical Vein Endothelial Cells – Dox-inducible KRAS G12V | Current study | N/A |
| Primary Human Brain Microvascular Endothelial Cells | Cell Systems | ACBRI 376 |
| Patient-Derived CD31+ Endothelial Cells | Current study; Derived from resected bAVM tissues. | N/A |
| Transgenic Zebrafish Line | PMID: 14608381 | Tg(gata1:dsRed)[sd2] |
| EC-specific KRAS[WT] Mouse Model | PMID: 32552404 | Cdh5-CreERT2; KRAS[WT/WT] |
| EC-specific KRAS[G12D] Mouse Model | PMID: 32552404 | Cdh5-CreERT2; KRAS[G12D/WT] |
| **Recombinant DNA** | | |
| pCK-hyPBase (JDW638) | Current study; Dr. Joshua D. Wythe | N/A |
| pCS2 Only (JDW365) | Current study; Dr. Joshua D. Wythe | N/A |
| pCS2-V5-mScarlet-I-KRAS4A-WT (JDW836) | PMID: 32552404; Dr. Joshua D. Wythe | Addgene ID: 156411 |
| pCS2-V5-mScarletI-KRAS4A-G12V (JDW837) | PMID: 32552404; Dr. Joshua D. Wythe | Addgene ID: 156410 |
| pBA_TA_ERN_mScarlet_KRASWT_rtTA (JDW890) | Current study; Dr. Joshua D. Wythe | N/A |
| pBA_TA_ERN_mScarlet_KRASG12V_rtTA (JDW891) | Current study; Dr. Joshua D. Wythe | N/A |
| hGLUT1-HA-pcDNA3.1-Halo | PMID: 35921166; Dr. Amira Klip | N/A |
| pTol2-kdrl:EGFP-KRAS4BG12V-ac/Y (JDW770) | PMID: 32552404; Dr. Joshua D. Wythe | Addgene ID: 156414 |
| **Antibodies** | | |
| KRAS Polyclonal antibody – Dilution 1:1000 for WB | ProteinTech | 12063-1-AP |
| GAPDH (D16H11) XP® Rabbit mAb #5174 – Dilution 1:2000 for WB | Cell Signaling Technologies | 5174S |
| Phospho-VE-cadherin (Tyr658) Polyclonal Antibody – Dilution 1:1000 for WB | Invitrogen | 44-1144 G |
| VE-cadherin Antibody (F-8) – Dilution 1:200 for IF and 1:2000 for WB | Santa Cruz | sc-9989 |
| Phospho-p44/42 MAPK (Erk1/2) (Thr202/Tyr204) Antibody – Dilution 1:200 for IF and 1:2000 for WB | Cell Signaling Technologies | 9101S |
| p44/42 MAPK (Erk1/2) (137F5) Rabbit mAb – Dilution 1:2000 for WB | Cell Signaling Technologies | 4695S |
| Phospho-Akt (Ser473) (D9E) XP® Rabbit mAb – Dilution 1:100 for IF and 1:1000 for WB | Cell Signaling Technologies | 4060S |
| Akt Antibody – Dilution 1:1000 for WB | Cell Signaling Technologies | 9272S |
| Hyaluronan Binding Protein (HABP), recombinant Versican G1 domain (Biotin Conj.) – Dilution 1:100 for IF | Cosmo Bio LTD | HKD-BC41 |
| Anti-Fibronectin antibody – Dilution 1:100 for IF | Abcam | ab2413 |
| Anti-mouse IgG, HRP-linked Antibody – Dilution 1:5000 for WB | Cell Signaling Technologies | 7076S |
| Anti-rabbit IgG, HRP-linked Antibody – Dilution 1:5000 for WB | Cell Signaling Technologies | 7074S |

| Reagent/resource | Reference or source | Identifier or catalog number |
| --- | --- | --- |
| Goat anti-Rabbit IgG (H + L) Cross-Adsorbed Secondary Antibody, Alexa Fluor™ 488 – Dilution 1:200 for IF | Invitrogen | A-11008 |
| Goat anti-Mouse IgG (H + L) Cross-Adsorbed Secondary Antibody, Alexa Fluor™ 488 – Dilution 1:200 for IF | Invitrogen | A-11001 |
| Goat anti-Rabbit IgG (H + L) Cross-Adsorbed Secondary Antibody, Alexa Fluor™ 568 – Dilution 1:200 for IF | Invitrogen | A-11011 |
| Goat anti-Mouse IgG (H + L) Cross-Adsorbed Secondary Antibody, Alexa Fluor™ 647 – Dilution 1:200 for IF | Invitrogen | A-21235 |
| Donkey anti-Rabbit IgG (H + L) Highly Cross-Adsorbed Secondary Antibody, Alexa Fluor™ 647 – Dilution 1:200 for IF (1:500 for the HA-tag experiment) | Invitrogen | A-31573 |
| Alexa Fluor® 647 Streptavidin – Dilution 1:100 for IF | Jackson ImmunoResearch | 016-600-084 |
| β-Actin Antibody – Dilution 1:2000 for WB | Cell Signaling Technologies | 4967S |
| Ki-67 (8D5) Mouse mAb – Dilution 1:200 for IF | Cell Signaling Technologies | 9449S |
| BrdU (Bu20a) Mouse mAb – Dilution 1:200 for IF | Cell Signaling Technologies | 5292S |
| HK1 Monoclonal Antibody (R.457.3) – Dilution 1:2000 for WB | Invitrogen | MA5-14789 |
| Recombinant Anti-Hexokinase II antibody [EPR20839] – Dilution 1:100-1:200 for IF and 1:2000 for WB | Abcam | ab209847 |
| Recombinant Anti-Glucose Transporter GLUT1 antibody [EPR3915] – Dilution 1:200 for IF and 1:2000 for WB | Abcam | ab115730 |
| GLUT3 Polyclonal antibody – Dilution 1:2000 for WB | ProteinTech | 20403-1-AP |
| Anti-Actin, α-Smooth Muscle - Cy3™ antibody, Mouse monoclonal – Dilution 1:500 for IF | Millipore-Sigma | C6198-100UL |
| alpha 1 Sodium Potassium ATPase/ ATP1A1 Antibody (F-2) – Dilution 1:2000 for WB | Santa Cruz | sc-514614 |
| α-Tubulin (DM1A) Mouse mAb – Dilution 1:2000 for WB | Cell Signaling Technologies | 3873S |
| Alexa Fluor® 488 AffiniPure Donkey Anti-Guinea Pig IgG (H + L) – Dilution 1:200 for IF | Jackson ImmunoResearch | 706-545-148 |
| HA-Tag (C29F4) Rabbit mAb – Dilution 1:500 for IF | Cell Signaling Technologies | 3724S |
| ZO-1 Monoclonal Antibody (ZO1-1A12) – Dilution 1:100 for IF | Invitrogen | 33-9100 |
| CD31 antibody – Dilution 1:200 for IF | Synaptic Systems | 351-004 |
| PDHX Polyclonal Antibody – Dilution 1:1000 for WB | ProteinTech | 10951-1-AP |
| Claudin 5 Monoclonal Antibody (4C3C2), Alexa Fluor™ 488 – Dilution 1:250 for IF | Invitrogen | 352588 |
| Goat anti-Rabbit IgG (H + L) Secondary Antibody, DyLight™ 405 – Dilution 1:500 for IF | Invitrogen | 35551 |
| Lycopersicon Esculentum (Tomato) Lectin (LEL, TL), DyLight® 649 (as suggested by the manufacturer: 5-20 μg/mL) | Vector Laboratories | DL-1178-1 |
| **Oligonucleotides and other sequence-based reagents** | | |
| CD31 qPCR Primers | Current study; Integrated DNA Technologies (IDT) | Supplementary Methods (Appendix Table 1) |

| Reagent/resource | Reference or source | Identifier or catalog number |
|---|---|---|
| CDH5 qPCR Primers | Current study; Integrated DNA Technologies (IDT) | Supplementary Methods (Appendix Table 1) |
| vWF qPCR Primers | Current study; Integrated DNA Technologies (IDT) | Supplementary Methods (Appendix Table 1) |
| NOS3 qPCR Primers | Current study; Integrated DNA Technologies (IDT) | Supplementary Methods (Appendix Table 1) |
| EGFL7 qPCR Primers | Current study; Integrated DNA Technologies (IDT) | Supplementary Methods (Appendix Table 1) |
| SPRY4 qPCR Primers | Current study; Integrated DNA Technologies (IDT) | Supplementary Methods (Appendix Table 1) |
| EGR1 qPCR Primers | Current study; Integrated DNA Technologies (IDT) | Supplementary Methods (Appendix Table 1) |
| TWIST1 qPCR Primers | Current study; Integrated DNA Technologies (IDT) | Supplementary Methods (Appendix Table 1) |
| SOX2 qPCR Primers | Current study; Integrated DNA Technologies (IDT) | Supplementary Methods (Appendix Table 1) |
| SNAI1 qPCR Primers | Current study; Integrated DNA Technologies (IDT) | Supplementary Methods (Appendix Table 1) |
| KLF4 qPCR Primers | Current study; Integrated DNA Technologies (IDT) | Supplementary Methods (Appendix Table 1) |
| ACTA2 qPCR Primers | Current study; Integrated DNA Technologies (IDT) | Supplementary Methods (Appendix Table 1) |
| TBP qPCR Primers | Current study; Integrated DNA Technologies (IDT) | Supplementary Methods (Appendix Table 1) |
| GAPDH qPCR Primers | Current study; Integrated DNA Technologies (IDT) | Supplementary Methods (Appendix Table 1) |
| HK1 qPCR Primers | Current study; Integrated DNA Technologies (IDT) | Supplementary Methods (Appendix Table 1) |
| HK2 qPCR Primers | Current study; Integrated DNA Technologies (IDT) | Supplementary Methods (Appendix Table 1) |
| TIGAR qPCR Primers | Current study; Integrated DNA Technologies (IDT) | Supplementary Methods (Appendix Table 1) |
| PFKFB3 qPCR Primers | Current study; Integrated DNA Technologies (IDT) | Supplementary Methods (Appendix Table 1) |
| SLC2A1/GLUT1 qPCR Primers | Current study; Integrated DNA Technologies (IDT) | Supplementary Methods (Appendix Table 1) |

| Reagent/resource | Reference or source | Identifier or catalog number |
|---|---|---|
| SLC2A3/GLUT3 qPCR Primers | Current study; Integrated DNA Technologies (IDT) | Supplementary Methods (Appendix Table 1) |
| Negative Control siRNA (5 nmol) | Qiagen | 1022076 |
| FlexiTube GeneSolution GS3099 for HK2 | Qiagen | 1027416 |
| **Chemicals, enzymes, and other reagents** | | |
| Lipofectamine™ RNAiMAX Transfection Reagent | Thermo Fisher Scientific | 13778150 |
| P5 Primary Cell 4D-Nucleofector™ X Kit L | Lonza | V4XP-5024 |
| Endothelial Cell Media | ScienCell | 1001 |
| EGMTM -2 MV Microvascular Endothelial Cell Growth Medium-2 BulletKitTM | Lonza | CC-3202 |
| Doxycycline | Sigma-Aldrich | D3072-1ML |
| Rat Tail Collagen Type I | Sigma-Aldrich | C3867-1VL |
| Attachment Factor | Gibco | S-006-100 |
| Fibronectin | Roche | 11051407001 |
| 2-Deoxy-D-glucose | Sigma-Aldrich | D8375-1G |
| BAY 876 | Tocris | 6199 |
| SL327 | Sigma-Aldrich | S4069-5MG |
| Trametinib | SelleckChem | S2673 |
| 3-PO | Sigma-Aldrich | 525330-25MG |
| Plasma Membrane Protein Extraction Kit | Abcam | 65400 |
| VEGF | Gibco | PHC9394 |
| Thrombin | Roche | 10602400001 |
| Phorbol 12-Myristate 13-Acetate (PMA) | Tocris | 1201-1 mg |
| Glucose Uptake-Glo™ Assay Kit | Promega | J1342 |
| Lactate-Glo™ Assay Kit | Promega | J5021 |
| 2-NBDG | Cayman Chemical | 11046 |
| In Situ Cell Death Detection Kit, Fluorescein | Roche | 11684795910 |
| Proteome Profiler Human Angiogenesis Array Kit | R&D Systems | ARY007 |
| Seahorse XFe24 FluxPak | Agilent | 102340-100 |
| 70 kDa FITC Dextran | Sigma-Aldrich | 46945-100MG-F |
| CellTracker Green CMFDA | Invitrogen | C7025 |
| Methylcellulose | Sigma-Aldrich | M0512-100G |
| Cultrex 3-D Culture Matrix Rat Collagen I | R&D Systems | 3447-020-01 |
| Alexa Fluor™ 488 Phalloidin | Invitrogen | A12379 |
| Alexa Fluor™ 647 Phalloidin | Invitrogen | A22287 |
| Hoechst 33342 | Thermo Fisher Scientific | 62249 |
| ProLong™ Gold Antifade Mountant | Invitrogen | P36930 |
| SuperSignal™ West Pico PLUS Chemiluminescent Substrates | Thermo Fisher Scientific | 34580 |
| BLUeye Prestained Protein Ladder | FroggaBio | PM007-0500F |
| BCA Protein Assay Kit | Pierce | 23227 |
| RIPA Lysis Buffer 10X | Sigma-Aldrich | 20-188 |
| Protease Inhibitor Tablets | Roche | 11836170001 |
| Phosphatase Inhibitor Mini Tablets | Pierce | A32957 |
| LightCycler® 480 SYBR Green I Master Mix | Roche | 4707516001 |
| PowerTrack™ SYBR Green Master Mix | Applied Biosystems | A46112 |

| Reagent/resource | Reference or source | Identifier or catalog number |
|---|---|---|
| High-Capacity cDNA Reverse Transcription Kit | Applied Biosystems | 4368814 |
| TRIzol Reagent | Invitrogen | 15596018 |
| Chloroform | Caledon Laboratories | 3000-1-10 |
| G418 | BioShop | GEN418.1 |
| TrypLE™ Express | Gibco | 12604-021 |
| Oregon Green HaloTag® Ligands | Promega | G2801 |
| Citrate Buffer, pH 6.0, 10X, Antigen Retriever | Sigma-Aldrich | C9999-100ML |
| TMTpro16 | Thermo Fisher Scientific | A44521 |
| Collagenase | Sigma-Aldrich | C6885 |
| CD31 Dynabeads | Invitrogen | 11155D |
| Phenylthiourea (PTU) | Sigma-Aldrich | P7629 |
| Tricaine/MS-222 | Sigma-Aldrich | A5040 |
| Tetrahydrofuran | Sigma-Aldrich | 109-99-9 |
| Tamoxifen | Sigma-Aldrich | T5648 |
| **Software** | | |
| FIJI ImageJ (version 2.9.0) | PMID: 22743772 | https://imagej.net/software/fiji/downloads |
| Adobe Illustrator | Adobe | https://www.adobe.com/ca/products/illustrator.html |
| GraphPad Prism 10 | GraphPad Software | https://www.graphpad.com/features |
| RTCA Software 2.0 | Agilent | https://www.agilent.com/en/product/cell-analysis/real-time-cell-analysis/rtca-software |
| Wave software (version 2.6.3) | Agilent | https://www.agilent.com/en/product/cell-analysis |
| Primer3Plus | PMID: 22730293 | https://www.primer3plus.com/index.html |
| Primer-BLAST | PMID: 22708584 | https://www.ncbi.nlm.nih.gov/tools/primer-blast/ |
| DESeq2 R package (1.20.0) | PMID: 25516281 | https://bioconductor.org/packages/release/bioc/html/DESeq2.html |
| Proteome Discoverer 2.2 | Thermo Fisher Scientific | https://documents.thermofisher.com/TFS-Assets/CMD/manuals/Man-XCALI-97808-Proteome-Discoverer-User-ManXCALI97808-EN.pdf |
| Enrichr | PMID: 33780170 | https://maayanlab.cloud/Enrichr/ |
| RNAlysis | PMID: 37024838 | https://github.com/GuyTeichman/RNAlysis |

| Reagent/resource | Reference or source | Identifier or catalog number |
|---|---|---|
| The STRING Database | PMID: 33237311 | https://string-db.org/ |
| Perseus | PMID: 27348712 | https://www.biochem.mpg.de/6304220/perseus |
| BioRender | BioRender | https://www.biorender.com/ |
| **Other** | | |
| 4D-Nucleofector® Core Unit and X Unit | Lonza | AAF-1003B; AAF-1003X |
| Olympus Fluoview 1000 Confocal Microscope (Olympus IX81 Inverted Stand) | Olympus | FV1000 |
| Nikon A1R Confocal Microscope (Nikon Eclipse Ti Inverted Stand) | Nikon | A1R HD25 |
| Leica SP8 Laser Scanning Confocal Microscope | Leica | SP8 |
| AxioZoom V16 Epifluorescence Macroscope | Zeiss | AxioZoom V16 |
| xCELLigence RTCA DP Analyzer | Agilent | N/A |
| Orion Star A111 pH Meter | Thermo Fisher Scientific | STARA1110 |
| Cytation 5 Imaging Reader | BioTek | N/A |
| Simple Plex Ella | ProteinSimple | 600-100 |
| Seahorse XFe24 Analyzer | Agilent | N/A |
| PowerPac Basic system | Bio-Rad | 1645050 |
| Trans-Blot SD Semi-Dry Transfer Cell and PowerPac 1000 | Bio-Rad | 1703940 |
| ChemiDoc Imaging System | Bio-Rad | 12003153 |
| LightCycler 480 Real Time PCR Instrument | Roche | 5015243001 |
| QuantStudio 5 Real-Time PCR Instrument | Applied Biosystems | A28135 |
| ProFlex PCR System | Applied Biosystems | 4483636 |
| DS-11+ Spectrophotometer | DeNovix | N/A |
| Orbitrap Fusion™Lumos™Tribrid™ | Thermo Fisher Scientific | N/A |

Additional detailed analytical methods are provided in the Supplementary Appendix.

## Study design

The overall objective of this study was to understand the molecular and cellular effects of mutant KRAS signaling on endothelial biology and to identify novel therapeutic avenues for the treatment of KRAS-driven sporadic AVMs. In particular, we delved in-depth into the metabolic changes in ECs expressing mutant KRAS and performed extensive analysis through a combination of in vitro and in vivo experiments with engineered cell lines, primary cells, mouse, zebrafish, and clinical samples (resected tissues from bAVM patients). All experiments were performed in a controlled laboratory environment. Cell culture experiments were not blinded; however, quantitative measurements were taken whenever possible to minimize human bias. For zebrafish experiments, zebrafish were assigned randomly to various treatment groups in equal numbers. Inhibitor treatments and assessments of arteriovenous shunts following treatment were conducted by blinded researchers. Exclusion criteria

and outliers were defined prior to experiments. Briefly, for experiments involving expression of mutant KRAS (stable or transient transfections), an elevated pERK level was confirmed to ensure that the transfection was successful and that the mutant KRAS proteins were active. For experiments involving siRNA knock-down of HK2, knock-down efficiency was assessed by either western blots or immunostaining. For inhibitor treatments, the effects of inhibitors were confirmed (e.g., checking pERK levels for MEK inhibitors, validating the effects of 2-Deoxy-D-Glucose [2-DG]) and BAY876 through RNA-sequencing. For zebrafish experiments, embryos with excessive gross morphological abnormalities were removed. As only embryos were assessed, no assessment of sex as a biological variable was possible. For mouse experiments, brains exhibiting venous or arterial dilation and evidence of arteriovenous shunting were scored as positive for AVMs and were used for immunofluorescence staining. Wild-type and *Kras* mutant mice were sex-matched. For studies involving clinical samples, patients with a family history of AVMs and patients with other genetic vascular diseases were excluded. A minimum of three biological replicates/independent experiments were utilized, except for technical parameter validations and optimizations (e.g., hyaluronidase treatment in Appendix Fig. S1F to validate the HABP staining) or due to limitations with available clinical resources. The number of replicates is indicated in the figure legends.

## Endothelial cell culture and KRAS induction

Immortalized human umbilical vein endothelial cells (IM-HUVEC/ IM-HUVEC F-Tractin) were cultured in complete Endothelial Cell Media (EC media) from ScienCell (Cat #: 1001). IM-HUVEC and IM-HUVEC F-Tractin lines were gifted from Dr. Arnold Hayer (McGill University) (Hayer et al, 2016). Primary human brain microvascular endothelial cells (phBMVEC) and patient-derived endothelial cells were cultured in complete EGM™ -2 MV Microvascular Endothelial Cell Growth Medium-2 BulletKitTM (Cat #: CC-3202). phBMVEC (Cell Systems, Cat #: ACBRI 376) were purchased commercially. phBMVEC were used before P10, and IM-HUVEC were used before P25. Cell lines were periodically tested for mycoplasma contamination using the Mycoplasma Detection Kit (Lonza, Cat #: LT07-118). Cells were maintained in incubators (ThermoFisher, Heracell VIOS 160i $CO_2$ incubator) at 37 °C with 5% $CO_2$. Doxycycline (Dox) (Sigma, Cat #: D3072-1ML) induction for stably transfected IM-HUVEC KRAS cell lines was conducted at a working concentration of 5 µg/mL. When applicable, serum starvation was done using basal media with 0.1% fetal bovine serum (FBS) (Gibco, Cat #: 12484-028). Cell culture coating solutions included: 150 µg/mL rat tail collagen type I (Sigma, C3867-1VL), 1× Attachment Factor (Gibco, Cat #: S-006-100), or 10 µg/mL fibronectin (Roche, Cat #: 11051407001), as appropriate for the experiment. Generation of the stable cell lines was previously described (Soon et al, 2022). For transient electroporation, KRAS constructs used were KRAS WT [JDW836; pCS2-V5-mScarlet-I-KRAS4A-WT] and KRAS G12V [JDW837; pCS2-V5-mScarlet-I-KRAS4A-G12V] (Fish et al, 2020). Electroporation was conducted using Lonza's 4D-Nucleofector Unit and the P5 Primary Cell 4D-Nucleofector™ X Kit L (Cat #: V4XP-5024) with 2 µg per reaction. Cells were seeded directly into experimental plates/ chambers after electroporation. All experiments were performed within 4 days of transient electroporation.

## RNA extraction, cDNA synthesis, and quantitative polymerase chain reaction (qPCR)

RNA was isolated using TRIzol reagent (Invitrogen, Cat #: 15596018) according to the manufacturer's recommendations. Unless otherwise specified, cells were serum starved with 0.1% FBS in basal EC media for 6 h prior to RNA collection. 1 µg of RNA was used for each cDNA synthesis reaction. cDNA synthesis was performed using the High Capacity cDNA Reverse Transcription Kit (Applied Biosystems, Cat #: 4374966) according to the manufacturer's recommendations with the ProFlex PCR System from Applied Biosystems (Cat # 4483636). Cycle time for cDNA synthesis was as follows: (1) 25 °C for 10 min (1×); (2) 37 °C for 2 h (1×); (3) 85 °C for 5 min (1×); (4) hold at 4 °C. A list of primers can be found in Appendix Table S1. qPCR was run using either the LightCycler 480 Real Time PCR Instrument from Roche (Cat #: 5015243001) or the QuantStudio 5 Real-Time PCR instrument from Applied Biosystems (Cat #: A28135) using default programs (i.e., Comparative Ct with Melt program in QuantStudio 5). The Sybr green reagents used were LightCycler® 480 SYBR Green I Master Mix (Cat #: 4707516001) or PowerTrack™ SYBR Green Master Mix (Applied Biosystems, Cat #: A46112).

## Protein collection and western blots

For protein collection, cells were washed with ice-cold PBS and lysed with 2× RIPA buffer (Millipore, RIPA Lysis Buffer 10×, Cat # 20-188) with protease inhibitors (Roche, Protease inhibitor tablets, Cat # 11836170001) and phosphatase inhibitors (Thermo, Pierce™ Phosphatase Inhibitor Mini Tablets, A32957). For glycolysis inhibitor experiments, treatments were typically performed for 24 h prior to protein collection. Unless otherwise specified, cells were serum starved with 0.1% FBS in basal EC media for 6 h prior to protein collection. For siRNA knockdown studies, cells were serum-starved for 3 h. To concentrate cell culture media, samples were transferred to Amicon Ultra-4 Centrifugal Filter Unit (10 kDa cutoff, Millipore, Cat #: UFC801096D) and concentrated by centrifuging at $3900 \times g$, 4 °C until the volume was ~300 µL. Protease inhibitors were added to the media samples afterwards. BCA measurement was performed after protein collection using Pierce's BCA Protein Assay Kit (Cat #: 23227). Before gel loading, protein samples were mixed with either 2× or 4× Laemmli buffer (Bio-Rad, Cat #: 161-0737 or 161-0747) and 2.5% of beta-mercaptoethanol (Thermo, Cat #: 125472500) and heated at 95 °C for 10 min. For cell lysate analysis, ~2–5 µg of proteins were loaded onto 8–12% gels, depending on the molecular weights of target proteins. Gels were run using Bio-Rad's PowerPac Basic system at 80 V for 30 min, followed by 120 V for 60 min. Semi-dry gel transfer was done using Bio-Rad's Trans-Blot SD Semi-Dry Transfer Cell and PowerPac 1000 and run at 15 V for 60–90 min. Proteins in gels were transferred to PVDF membranes (GE Healthcare, Cat #: 10600023). Membrane blocking was done using 5% milk (BioShop Cat #: SKI400.500) diluted in 1X TBS-0.2% Tween 20 (TBST; 10× TBS, Fisher Scientific, Cat #: BP2471-1; Tween 20, BioShop, Cat #: TWN510.500) for 1 h at room temperature (RT), with shaking. For phosphorylated proteins, blocking was done using 5% BSA (BioShop, Cat #: ALB001.500) diluted in 1× TBST. Primary antibodies were diluted in 1% milk or BSA in 1× TBST and incubated at 4 °C overnight, with shaking. The

next day, membranes were washed with 1× TBST three times, and secondary antibodies (diluted in 1% milk or BSA-1X TBST) were added and incubated for 1 h at RT with shaking. For chemiluminescence development, SuperSignal™ West Pico PLUS Chemiluminescent Substrate (Thermo, Cat #: 34580) was used, and membranes were imaged using Bio-Rad's ChemiDoc Imaging System. A detailed list of primary and secondary antibodies is included separately in the Reagents and Tools Table. Uncropped western blot images are included in the Source Data.

## Immunofluorescence (IF) staining—cell culture

Cells were fixed with 4% paraformaldehyde (PFA) (Electron Microscopy Sciences, Cat #: 15710) at RT for 20 min. After fixation, cells were washed with PBS and permeabilized with 0.5% Triton X-100 (Sigma, Cat #: X100-500ML) at RT for 10 min. Blocking was done with 5% BSA-0.1% Triton X-100 in PBS (PBST) for 1 h at RT, followed by primary antibody incubation, diluted in 1% BSA-PBST, overnight at 4 °C. On the second day, cells were washed with PBST three times. Then, appropriate secondary antibodies, diluted in 1% BSA-PBST, were added and incubated at RT for 2 h, followed by PBST washes. In some experiments, phalloidin (Alexa Fluor™ 488 Phalloidin, Invitrogen, Cat #: A12379; Alexa Fluor™ 647 Phalloidin, Invitrogen, Cat #: A22287) was diluted to 1× following the manufacturer's instructions in PBS and added to cells for 30 min at RT or overnight at 4 °C. Cells were washed again with PBST after phalloidin staining. For nuclear staining, Hoechst 33342 (20 µM, Thermo, Cat #: 62249) was added to cells and incubated at RT for 10 min. Mounting was done with ProLong™ Gold Antifade Mountant (Invitrogen, Cat # P36930) when using chamber slides and coverslips for staining. All incubation steps were performed with protection from light. A detailed list of primary and secondary antibodies used is included in the Reagents and Tools Table.

For fibronectin staining, cell culture dishes were pre-coated with 10 µg/mL of fibronectin (Roche, Cat #: 11051407001) and incubated for 1 h at 37 °C. Fibronectin solution was removed before cell seeding. For hyaluronic acid staining, cells were not permeabilized after fixation, and Triton was not added to the blocking and antibody dilution buffers. Dox induction was performed for 3 days prior to fixation and staining.

For BrdU assays, 10 µM of BrdU was added to cells for 24 h before cell fixation for analysis. Dox induction was performed for the same period of time. During the staining workflow, an additional acid hydrolysis step—incubation with 2 N HCl for 30 min at RT—was performed between permeabilization and blocking steps.

For combined MEK and glycolysis inhibitor treatments in vitro, KRAS expression was induced for 2 days prior to inhibitor treatment with 2 mM 2-DG alone or 2 mM 2-DG combined with either 10 µM SL327 or 100 nM trametinib for another 2 days before fixation and staining.

CellTracker Green staining was performed according to the manufacturer's protocol using CellTracker™ Green CMFDA (Invitrogen, Cat #: C7025).

## Spheroid sprouting assay

A 1% methylcellulose stock solution (Sigma, Cat #: M0512-100G) was made in basal EGM-2 media. After mixing, the solution was centrifuged at $500 \times g$ for 5 min at RT. After spinning, the top layer was transferred to a new tube. This 1% methylcellulose stock solution was further diluted with basal EGM-2 to make 0.2% methylcellulose solution or diluted with FBS at 3:2 ratio to make a methylcellulose solution with 40% FBS. Both solutions were sterile-filtered through 0.22µm filters (Millipore, Cat #: SLGP033RS). The 3D collagen matrix mixture was prepared fresh every time before use. For each mL of collagen matrix mixture, the mixture contained: 550 µL of methylcellulose solution with 40% FBS + 150 µL of 15.6 mg/mL NaHCO$_3$ in water + 10 µL of 1 M NaOH + 300 µL of Cultrex 3-D Culture Matrix Rat Collagen I (R&D Systems, Cat #: 3447-020-01). IM-HUVEC KRAS$^{WT\ or\ G12V}$ cells were counted, and 200,000 cells for each condition were pelleted at $500 \times g$ for 5 min at RT. Cells were resuspended in 5 mL of 0.2% methylcellulose solution. Cells were then seeded droplet-wise onto the lids of tissue culture dishes (25 µL per droplet = 1000 cells per spheroid). After seeding, lids were flipped upright and placed back onto tissue culture dishes. Droplets were placed in 37 °C incubators overnight for spheroids to form. On day 2, spheroids were collected into Falcon tubes by rinsing with 10% FBS in PBS. Spheroids were centrifuged at $300 \times g$ for 5 min at RT. Supernatant was removed. Spheroids were resuspended in a collagen matrix mixture and transferred into a 24-well plate (500 µL/well). The matrix was incubated at 37 °C for 5 min to solidify. After incubation, 500 µL of appropriate cell culture media (complete ScienCell EC media with or without Dox and/or inhibitors) was added to each well. Spheroids embedded in a collagen matrix were placed in the tissue culture incubator (37 °C) for 24 h to allow for sprouting (Dox induction time = 24 h). Afterward, spheroids were fixed with 4% PFA at RT for 30 min, permeabilized with 0.5% Triton, and blocked with 5% BSA. Phalloidin and Hoechst were diluted together in PBS and incubated at 4 °C overnight. After staining, spheroids were washed with PBS and stored in PBS until imaging.

KRAS expression was induced once the spheroids were embedded in a 3D collagen matrix for the initial characterization and inhibitor treatment studies, except for siHK2 experiments, where KRAS expression was induced for 1 day prior to siHK2 knockdown, followed by spheroid formation and sprouting after 2 days of siHK2 knockdown. For inhibitor studies, inhibitor conditions were 2 mM 2-DG, 5 µM BAY876, and 10 µM SL327.

## Migration assay

Cell migration assays were performed with Ibidi's µ-Dish 35 mm (Cat #: 81156) with 2 Well Culture-Inserts for self-insertion (Cat #: 80209), or the pre-inserted Culture-Insert 2 Well in µ-Dish 35 mm (Cat #: 81176). KRAS expression was induced with Dox the day prior to migration experiments. For experiments involving CellTracker staining, cells were first incubated with cell culture media containing CellTracker Green and Hoechst 33342 for 30 min at 37 °C. After staining, cells were gently washed once with PBS, and Ibidi inserts were lifted. In total, 2 mL of fresh media was supplied to each dish, and cells were allowed to migrate in the incubator (37 °C) for 5–6 h, before fixation with 4% PFA. Migration was performed in serum-starved media (0.1% FBS EC media) for the initial characterization experiment. For experiments with glycolysis inhibitors and siHK2 knockdown, migration was performed in complete EC media. Dox and inhibitors were kept in the media during cell migration. Inhibitor

conditions were 2 mM 2-DG, 5 µM BAY876, or 10 µM SL327. For siHK2 studies, migration assay was performed following 2 days of siRNA knockdown.

## Transwell leak assay

Experiments were performed with the 6.5 mm Transwell® with 0.4 µm Pore Polycarbonate Membrane Insert (Corning, Cat #: 3413). Transwells were pre-coated with Attachment Factor prior to cell seeding (incubation at 37 °C for 1 h). Cells were seeded in the top chamber of the transwell at high density to ensure monolayer formation (150,000 cells per well). KRAS$^{WT}$ or $^{G12V}$ expression was induced for 4 days before leakage assessments. Cells were incubated with 1 mg/mL of 70 kDa FITC solution (Sigma, Cat #: 46945-100MG-F) in the top chamber of the transwells at 37 °C for 30 min (150 µL volume). The bottom chamber of the transwell was filled with 500 µL of HBSS. Following incubation, the top chamber was removed, and 100 µL of the bottom chamber solution was sampled to assess FITC leakage through the cell monolayer. Fluorescent measurements were done in a black-walled, clear-bottom 96-well plate (Corning, Cat # 3603) using the BioTek's Cytation 5 Imaging Reader (excitation = 485 nm; emission = 535 nm). A control well was included without cell seeding to assess FITC leakage through the transwell membrane. Experiments were done in technical duplicate wells, and measurements were taken in duplicates for each well ($2 \times 2 = 4$ measurements per condition per individual experiment).

## Seahorse assay

Mitochondrial stress test was performed using the Seahorse XF DMEM assay medium pack, pH 7.4 (Agilent, Cat #: 103680-100) according to the manufacturer's recommendations. Test media was prepared in basic DMEM + 10 mM glucose + 1 mM pyruvate + 2 mM L-glutamine. Glycolysis stress test media was prepared in basic DMEM + 2 mM L-glutamine. Seahorse culture plates, cartridges, and calibration solution were included in the Seahorse XFe24 FluxPak (Agilent, Cat #: 102340-100). Seahorse XF24 cell culture microplates were pre-coated with Attachment Factor and incubated at 37 °C for 1 h before removal. Cells were then seeded at 150,000/well and grown until confluency. KRAS expression was induced with Dox treatment shortly after cell seeding/attachment for 24 h prior to the experiment. Each experimental condition was seeded in duplicates for each independent experiment. Hydration of the Seahorse XFe24 sensor cartridge was done by adding 1 mL of the Seahorse XF Calibrant Solution per well and incubated overnight at 37 °C (no CO$_2$). Oxygen consumption rate (OCR) and extracellular acidification rate (ECAR) were measured in real-time for 100 min at default intervals using the Seahorse XFe24 Analyzer (Agilent). Three measurements were taken after each injection of drug compounds. Drug concentrations for the glycolysis stress tests were: 10 mM glucose, 1.5 µM Oligomycin, and 50 mM 2-DG. Drug concentrations for the mitochondrial stress tests were: 1.5 µM Oligomycin, 2 µM FCCP, and 0.5 µM Rotenone/Antimycin A. Data were visualized using Wave software (version 2.6.3). For quantifications, duplicate measurements were first averaged for each experimental condition at each time point. Then, average values of triplicate time points within each experimental phase (i.e., baseline, 1st, 2nd, and 3rd injections) were calculated. Using these values, glycolysis and mitochondrial respiration parameters were calculated according to the manufacturer's instructions.

## Proteome profiler dot blot

IM-HUVEC KRAS$^{G12V}$ cells (with and without 4 days of Dox induction) were serum starved (0.1% FBS EC media) for 16 h prior to sample collection. Cell lysates were collected with 2× RIPA buffer (with protease/phosphatase inhibitors). Cell culture media was concentrated using Amicon Ultra-15 Centrifugal Filter Unit (3 kDa cutoff, Millipore, Cat #: UFC900324) as above. Proteome Profiler Human Angiogenesis Array Kit was purchased from R&D Systems (Cat #: ARY007), and the assay was performed following the manufacturer's protocol. In total, 150 µg of protein samples were used for each condition. Membranes were imaged using Bio-Rad's ChemiDoc Imaging System.

## ELISA measurements of ANGPT2 and sTIE2

IM-HUVEC KRAS$^{G12V}$ cells (with and without 4 days of Dox induction) were serum starved (0.1% FBS EC media) for 16 h prior to sample collection. Cell culture media was processed as above. 100 µg of protein samples in 110 µL (diluted in PBS) per condition were used for the ELISA assay. Levels of soluble angiopoietin-2 (sAngpt2; Lower limit of quantification [LLOQ] 9.91 pg/mL; diluted 1:50) and soluble Tie-2 (sTie-2; LLOQ 72.2 pg/mL; diluted 1:1) were quantified in cell culture media using the Simple Plex Ella (ProteinSimple, San Jose, CA, USA) multiplex platform according to the manufacturer's instructions; values were reported as the average of triplicate readings per sample.

## TUNEL cell death assay

TUNEL assay was performed using the In Situ Cell Death Detection Kit, Fluorescein (Roche, Cat #: 11684795910) following the manufacturer's protocol. Nuclei were stained with Hoechst.

## 2-NBDG glucose uptake assay

IM-HUVEC KRAS$^{WT \text{ or } G12V}$ cells (with and without Dox induction) were cultured in 8-well chamber slides. Cells were first washed with PBS twice and switched to glucose-free EC media for 2 h of glucose starvation. Following starvation, cells were incubated with 150 µM of 2-NBDG (Cayman Chemical, Cat #: 11046) in glucose-free media for 1 h in the incubator. After incubation, Hoechst dye was added to stain the nuclei and incubated for 10 min at 37 °C. Cells were then washed with PBS, and live cell imaging was performed.

## Glucose uptake-Glo™ assay

A glucose uptake assay was conducted using the Glucose Uptake-Glo™ Assay Kit (Promega, Cat #: J1342) following the manufacturer's instructions. Briefly, IM-HUVEC KRAS$^{WT \text{ or } G12V}$ cells were seeded into two 96-well plates simultaneously at 7500 cells per well (one plate was used for measurement of glucose uptake, while the second plate was used for cell counting). Dox was added 2 days after seeding to induce KRAS expression. Inhibitors (5 µM BAY876 or 10 µM SL327) were added 1 day following Dox induction, and the glucose uptake assay

was performed 2 days following Dox induction (4 days after seeding). On the day of the assay, cells were first washed with PBS and serum/glucose starved for 2 h using the glucose-free EC media basal media (ScienCell, Cat #: 1001-GF). All measurements were done in duplicates. Luminescence reading was recorded with BioTek's Cytation 5 Imaging Reader (1 s integration). The final readout was calculated as Relative Light Unit (RLU)/cell number/minute and normalized to KRAS$^{G12V}$ no Dox condition.

## Lactate-Glo™ assay

Extracellular lactate was measured using the Lactate-Glo™ Assay Kit (Promega, Cat #: J5021) following the manufacturer's instructions. Following serum/glucose starvation as described above for the Glucose Uptake-Glo Assay, 100 μL of 5 mM glucose (diluted in glucose-free EC media basal media) was added to each well and incubated for 1 h prior to media sample collection and measurements.

## MAPK pathway stimulation

IM-HUVEC KRAS$^{WT \text{ or } G12V}$ cells (with and without Dox induction) were cultured for 2 days with Dox induction of KRAS expression. Before protein or RNA collection, cells were serum-starved with 0.1% FBS EC media for 6 h. During the last 30 min of serum starvation, cells were treated with 50 ng/mL VEGF (Gibco, Cat #: PHC9394), 0.02 U/mL thrombin (Roche, Cat #: 10602400001), or 1 μM Phorbol 12-myristate 13-acetate (PMA) (Tocris, Cat #: 1201-1 mg). Following stimulations, cells were washed with PBS, and lysates were collected for either RNA or protein isolation.

## Membrane fractionation

Membrane fractionation was conducted using the Plasma Membrane Protein Extraction Kit (Abcam, Cat #: 65400), following the manufacturer's protocol. Protein samples (Cytoplasmic [Cyto], total membrane [TM], and plasma membrane [PM] fractions) were processed and analyzed using a western blot without heat denaturing.

## pH measurements

IM-HUVEC KRAS$^{G12V}$ cells (with and without Dox induction) were cultured in six-well plates for 5 days. Inhibitor treatments were done at the following concentrations: 2 mM 2-DG, 5 μM BAY876, and 10 μM SL327. For pH measurements, cell culture media was removed and centrifuged at $1000 \times g$ for 5 min to remove any floating cells. Supernatant was transferred to a new tube, and pH of the cell culture media was measured immediately with Orion Star A111 pH Meter (Thermo Scientific).

## xCELLigence system to assess proliferation

xCELLigence experiments were performed with the xCELLigence RTCA DP analyzer (Agilent). Prior to cell seeding, xCELLigence plates (E-Plate 16 PET, Agilent, Cat #: 300-600-890) were first coated with 1X Attachment Factor and analyzed on the system with cell culture media for an hour to record baseline readings. Cells were seeded in xCELLigence plates in duplicates at 7500 cells per well in 150 μL of cell culture media (with/without Dox and/or glycolysis inhibitors). KRAS expression was induced prior to seeding. Impedance readings were recorded every 10 min for 90 h (glycolysis inhibitor study) or every 15 min for 96 h (siHK2 study). Cell doubling times were analyzed and calculated using the RTCA software 2.0. The time range for doubling time calculations was set based on the time period between initial cell seeding and maximum normalized cell index for each run.

## HK2 siRNA knockdown

Knockdown of HK2 was performed using siRNA purchased from Qiagen: Negative Control siRNA (Cat #: 1022076) and FlexiTube GeneSolution GS3099 for *HK2* (Cat #: 1027416). Transfection was done using Lipofectamine™ RNAiMAX Transfection Reagent (Thermo, Cat #: 13778150). Dox-induced expression of KRAS was initiated the day prior to transfection. The final concentration of siRNA is indicated in the figure legends. All subsequent experiments were conducted within 4 days of siRNA knockdown.

## GLUT1-Halo IF staining in phBMVECs

Primary hBMVECs were co-electroporated with 2 μg of pCS2 mScarlet-KRAS WT or mScarlet-KRAS G12V (JDW836 or JDW837) and 1 μg of human GLUT1-HA-pcDNA3.1-Halo plasmids (Yazdani et al, 2022) using Lonza's 4D-Nucleofector Unit and the P5 Primary Cell 4D-Nucleofector™ X Kit L. Cells were fixed 2 days after electroporation and inhibitor treatments (2 mM 2-DG, 5 μM BAY876, 10 μM SL327) were conducted for 24 h prior to fixation.

Oregon Green HaloTag® Ligands (Promega, Cat #: G2801) and Hoechst 33342 were used to label cells at 37 °C for 15 min. Following incubation, cells were gently washed three times with cell culture media. Then, the primary antibody against HA-tag was diluted 1:500 in cell culture media and added to the cells. Cells were incubated with the primary HA-tag antibody for 1 h in the 37 °C incubator and then gently washed twice with PBS. Following PBS washes, cells were fixed with 4% PFA and incubated at RT for 30 min. Cells were then blocked with 5% BSA in PBS (no Triton) for 1 h at RT. After blocking, the appropriate secondary antibody (1:500 in PBS) was added and incubated at RT for 2 h. Lastly, cells were washed with PBS three times and stored in PBS for confocal imaging.

## Bulk RNA sequencing

IM-HUVEC KRAS$^{G12V}$ cells were treated with 5 μg/mL of Dox for 2 days with 1 day of inhibitor treatments (2 mM 2-DG, 5 μM BAY876, 10 μM 3-PO [Sigma, Cat #: 525330-25MG] and 10 μM SL327) prior to RNA collection (using Trizol reagent). Cells were cultured in complete media without serum starvation. RNA samples (1 μg/μL; 10 μL) were submitted to Novogene Corporation Inc (Sacramento, CA) for bulk RNA sequencing. Briefly, messenger RNA was purified from total RNA using poly-T oligo-attached magnetic beads, followed by cDNA library construction. Quality control of the library was performed with Qubit and real-time PCR for quantification and a bioanalyzer for size distribution detection. The quantified library was sequenced on Illumina platforms, and paired-end reads were generated. Data analysis, including pathway analysis, is described in the Supplementary Appendix.

## Proteomics

IM-HUVEC KRAS$^{WT \text{ or } G12V}$ cells (with and without 2 days of Dox induction) were serum starved (0.1% FBS EC media) for 6 h prior to sample collection. SL327 treatment was used at 10 μM for 24 h. Cell pellets were sent to the Network Biology Collaborative Centre (Mount Sinai Hospital, Toronto) for sample processing and mass spectrometry analysis.

For sample processing, cell pellets were lysed in 500 μL of 5% SDS + 50 mM TEAB and sonicated. In total, 25 μg of protein material was reduced with 20 mM DTT for 10 min at 95 °C and alkylated with 40 mM iodoacetamide for 30 min in the dark. Phosphoric acid was added to a final concentration of 1.2%. In all, 165 μL of S-Trap protein binding buffer (90% methanol, 100 mM TEAB) was added to 27.5 μL of acidified lysate. The resulting mixture was passed through the microcolumn at $4000 \times g$. The microcolumn was washed 4× with the S-Trap protein binding buffer. Each sample was digested with 1 μg of trypsin (in 20 μL of 50 mM TEAB) for 1 h at 47 °C. Prior to elution, 40 μL of 50 mM TEAB pH 8 was added to the column. Peptides were eluted by centrifugation at $4000 \times g$. Peptides were eluted two more times with 40 μL 2% formic acid and 40 μL of 50% acetonitrile+2% formic acid. Eluted peptides were dried down and stored at −40 °C.

For tandem mass tag labeling, TMTpro16 (Thermo Scientific, Cat #: A44521) was used. After peptides were recovered from S-Trap digest, ~10 μg was labeled with 80 μg of its respective TMTpro16 channel. Peptides were resuspended in 10 μL of 100 mM HEPES pH 8.5 and mixed with 80 μg TMTpro16 label that was resuspended in 4 μL of acetonitrile. This was incubated for 1 h at RT. The reaction was quenched with 4 μL of 5% hydroxylamine (diluted in HEPES) for 15 min. The sample was lyophilized, and 1/20th of each labeled sample was combined for testing. For sample fractionation, 24 μg of combined peptides (1.5 μg per labeled sample) were fractionated using an Agilent 1260 Infinity HPLC system. A fritted self-packed 200μm ID (360μm OD) packed with Agilent Zorbax 300 Extend C18 High pH material (5μm beads) with a column length of 15 cm was used to separate peptides. Mass spectrometry data acquisition and analysis, including pathway analysis, are described in the Supplementary Appendix.

## Mouse—housing and animal care

All mouse protocols were approved by the Institutional Animal Care and Use Committee at the University of Virginia (protocol #4446). Mice were housed with access to food (normal chow diet) and water ad libitum on a 12-h light–12-h dark cycle at 21 °C and 50–60% humidity.

## Mouse—tamoxifen induction and tissue processing

$Kras^{Lox-STOP-Lox(LSL)-G12D/WT}$; $Rosa26^{LSL-tdTomato/LSL-tdTomato}$ females were crossed with males harboring the pan endothelial CreERT2 transgene, $Cdh5$-Cre$^{ERT2}$ (Fish et al, 2020). Adult mice ($Cdh5$-Cre$^{ERT2}$; $Kras^{WT/WT}$; $Rosa26^{tdTomato}$ or $Cdh5$-Cre$^{ERT2}$; $Kras^{LSL-G12D/WT}$; $Rosa26^{tdTomato}$; 2 months of age) received 0.015 mg tamoxifen/g bodyweight (diluted 1:10 in warm sesame oil [Sigma-Aldrich, Cat #: C8267]) by i.p. injection. 2 months post tamoxifen injection, mice were retro-orbitally injected with 50 μL of far-red (649 nm) fluorescently labeled tomato lectin (Lycopersicon esculentum; Vector Laboratories, Cat #: DL-1178-1). The lectin was allowed to circulate for 15 min to label the vasculature. Animals were then

anesthetized with inhaled isoflurane until a surgical plane of anesthesia was achieved. The chest was opened, the rib cage reflected, and 150 μL of fluorescently labeled lectin was perfused into the left ventricle, followed by 8 mL of ice-cold 1× PBS and 8 mL of 4% PFA. Brains were then carefully removed from the skull and briefly rinsed in 1× PBS.

Whole brains were imaged using an AxioZoom V16 epifluorescence macroscope (Zeiss) with a 1× Plan Neofluar Z objective (total magnification 11.3×, exposure time 500 ms) in the far-red channel to visualize AVMs. Brains exhibiting venous or arterial dilation and evidence of arteriovenous shunting were scored as positive for AVM. Following visualization of AVMs, brains were drop-fixed in 4% PFA (w/v) overnight at 4 °C with gentle agitation (40 rpm), followed by immunostaining and confocal microscopy.

## Immunofluorescence (IF) staining—mouse tissue

After fixation, control or AVM-containing brains were washed extensively in 1× PBS, embedded in 1% agarose, and sectioned at 300 μm thickness (oscillation: 5, speed: 5) using a Compresstome (Precisionary Instruments, Cat #: VF-300-Z). Thick brain sections were transferred to 20 mL scintillation vials (Sigma-Aldrich, Cat #: DWK986541) and delipidated in a 1:1 tetrahydrofuran (THF; Sigma-Aldrich, Cat #: 109-99-9) and deionized water solution for 16 h at RT with gentle agitation (40 rpm). Samples were washed three times in deionized water (30 min per wash at RT with gentle agitation). Tissues were permeabilized in permeabilization solution (PBS/0.2% Triton X-100 [Sigma-Aldrich, Cat #: X100-500ML]/20% DMSO [Sigma-Aldrich, Cat #: D8418]) for 4 h at RT with gentle agitation, then transferred to blocking solution (PBS/0.2% Triton X-100/10% DMSO/6% donkey serum [Sigma-Aldrich, Cat #: D9663-10ML]) and incubated overnight at 4 °C with gentle agitation. Samples were incubated in primary antibody solution containing HK2 (1:100; Abcam, Cat #: ab209847) and pre-conjugated Claudin-5–488 (1:250, Invitrogen, Cat #: 352588) diluted in antibody diluent (5% DMSO/3% donkey serum in PTwH [PBS/0.2% Tween-20 with 10 μg/mL heparin]) for 72 h at 32 °C with gentle agitation. After primary incubation, samples were washed three times in PTwH at RT and subsequently incubated with secondary antibody solution containing goat anti-rabbit–405 (1:500; Invitrogen, Cat #: 35551) diluted in the same antibody diluent as above. Secondary incubation was performed at 37 °C for 48 h with gentle agitation, followed by three washes in PTwH at room temperature with gentle agitation. Samples were mounted onto glass microscope slides (Fisher Scientific, Cat #: 22-0370246) using ProLong Gold Antifade Mounting Medium (Thermo Fisher, Cat #: P36930), and a coverslip was applied (Fisher Scientific, Cat #: 12-541-054). Slides were cured horizontally at RT overnight prior to imaging.

## Confocal imaging and analysis—mouse tissue

Confocal images were acquired using a Leica SP8 laser-scanning confocal microscope equipped with a ×40 oil immersion objective (NA 1.4). Z-stacks were collected at 1 μm step size over a total depth of 150–180 μm (laser power: 405 nm, 4.0%; 488 nm, 10.2%; 560 nm, 2.39%; 650 nm, 2.0%; pinhole: 51 μm). Three image stacks were acquired per animal. Images were processed in FIJI (ImageJ, 1.54p) to generate maximum intensity projections and exported as TIFF files for figure preparation in Adobe Illustrator. Confocal images were analyzed

using FIJI. Control brain regions were matched to the AVM regions in mutant KRAS animals, both within the cerebral cortex. Individual vessels were manually traced to generate regions of interest (ROIs). Vessels were categorized based on diameter, with those >15 µm classified as large vessels and those <15 µm classified as small vessels. For each ROI, mean pixel intensity was calculated to quantify fluorescence, and background subtraction was applied to correct for non-specific signal. Three confocal image stacks were collected per animal, with a total of three bAVM-bearing mutant mice and three control littermates analyzed.

## Zebrafish husbandry, transgenesis, and treatment

Zebrafish study protocols were approved by the Animal Care Committee at the University Health Network (Toronto, AUP #: 6055). Norecopa's PREPARE guidelines (Smith et al, 2018) were followed. Zebrafish were housed in separate tanks on a recirculating aquatic housing system (Aquaneering). The Tg(gata1:dsRed)$^{sd2}$ (Traver et al, 2003) transgenic line was utilized in this study. Zebrafish were bred from age of 3 months to 18 months by setting up crosses in false-bottomed tanks. For transient transgenesis, 1–4 cell stage embryos were injected with 1 nL of transgenesis mixture: 100–150 ng of plasmid DNA pTol2-kdrl:EGFP-KRAS4BG12V-ac/Y (JDW #770) or pTol2-kdrl:V5-mScarlet-I-KRAS4AG12V-1ac/mC (JDW #859) (both available at Addgene; https://www.addgene.org/browse/article/28211645) (Fish et al, 2020), 150 ng Tol2 transposase mRNA (InVivo Biosystems), 1 µL phenol Red, and HEPES 10 mM up to final volume of 11 µL. Embryos were manually dechorionated using forceps for imaging or drug treatment prior to hatching and were housed at 25–28 °C in E3 medium. Pigment formation was inhibited using a final concentration of 0.003% Phenylthiourea (PTU) (Millipore-Sigma, Cat #: P7629) added to the media at 20–24 h-post-fertilization (hpf). Phenotypic analysis was performed under anesthesia using Tricaine/MS-222 (Sigma, Cat #: A5040) at a concentration of 166 mg/L.

Inhibitor treatments were performed with direct dosing of the inhibitors in E3 media beginning at 36 hpf following confirmation of arteriovenous shunts. For single inhibitor treatments, inhibitor concentrations were 200 µM 2-DG, 1 µM BAY876, and 1 µM SL327. For combined treatments, inhibitor concentrations were 40 µM 2-DG, 0.2 µM BAY876, and 0.2 µM SL327. Treatments were performed over 24–72 h. Shunt recovery was assessed by a blinded reviewer after 24 h of treatment and reported as a percentage of the number of embryos without shunts following treatment/total number of embryos per treatment group.

## Confocal microscopy imaging and image analysis for cell culture and zebrafish

Confocal microscopy was performed using either an Olympus Fluoview 1000 Confocal microscope (Olympus IX81 inverted stand) or a Nikon A1R Confocal microscope with resonant scanner (Nikon Eclipse Ti inverted stand) or a Leica SP8 Laser Scanning Confocal Microscope. Objective lenses for the Olympus microscope were UPlan SApo 4×/0.16 NA, UPlan Apo 10×/0.40 NA, UPlan SApo 20×/0.75NA, Plan Apo 40×/1.35 NA oil immersion and Plan Apo 63×/1.42 NA oil immersion. Objective lenses for the Nikon microscope were Plan Apo 10×/0.45 NA, Plan Apo 20×/0.75 NA,

Plan FL 40×/1.3 NA oil, and Plan Apo, nano-crystal, 60x/1.4 NA, oil immersion. The objective lens for the Leica microscope was a 40× oil immersion objective (NA 1.4). Image analysis was done using FIJI ImageJ (version 2.9.0) (Schindelin et al, 2012). Image labeling and figure panels were generated using Adobe Illustrator.

## Patient samples and research ethics

Clinical samples used in this study included human temporal lobe and bAVM tissues, as well as endothelial cell cultures isolated from bAVM tissues. All patients provided written informed consent, and the study was conducted in compliance with the University Health Network Institutional Research Ethics Board regulations (CAPCR protocol ID 13-6009BE). The research was conducted in accordance with the principles set out in the WMA Declaration of Helsinki (World Medical Association, 2013) and the Department of Health and Human Services Belmont Report. Human temporal lobe specimens were obtained from adult patients (over the age of 18) who were undergoing temporal lobectomy for seizure management. Human bAVM specimens were obtained from adult patients (over the age of 18) who were undergoing surgical bAVM resections. Surgeries were performed by the Division of Neurosurgery, Department of Surgery, University Health Network at the Toronto Western Hospital in Toronto, Canada. All bAVM patients had sporadic, unifocal lesions with evidence of arteriovenous shunting on digital subtraction angiography. Clinical and surgical information was obtained from the patients' electronic charts by the study team. For in vitro studies, two samples were from male patients and two were from female patients between the ages of 20–60. KRAS mutations were detected by the clinical team using droplet digital polymerase chain reactions.

## Immunofluorescence (IF) staining—human tissue

Human temporal lobe and brain AVM tissues were embedded in OCT and cryo-sectioned into 20 or 40-µm sections. For target retrieval, Citrate Buffer, pH 6.0, 10×, Antigen Retriever (Sigma, Cat #: C9999-100ML) was diluted to 1× and heated to >95 °C in a water bath. Tissue slides were inserted into the pre-heated buffer and incubated at >95 °C for 15 min. After incubation, slide containers were removed from the heated water bath and left at RT to cool for 30 min. Slides were then washed with PBS once (10 min on a shaker) and blocked with 10% donkey serum-0.3% Triton X-100 in PBS (blocking buffer) for 90 min at RT. The appropriate primary antibodies were then diluted in blocking buffer and added to the slides overnight at 4 °C. On day 2, slides were washed with PBS three times, followed by secondary antibody incubation (diluted in blocking buffer) at RT for 90 min. Slides were washed again with PBS. The primary and secondary antibody incubation steps were repeated when staining multiple targets sequentially. Afterwards, slides were washed once with 70% ethanol for 5 min. Then, slides were quenched with 0.1% Sudan Black B in 70% ethanol for 30 min at RT. Following quenching, slides were washed with PBS again (3 × 10 min each), and Hoechst 33342 (1:200 diluted in PBS) was added to stain cell nuclei (10 min incubation at RT). After nuclear staining, slides were mounted with ProLong™ Gold Antifade Mountant and sealed. All incubation steps were protected from light. Z-stack images were taken with confocal microscopy.

## Human brain AVM EC isolation

Human brain AVM tissue was processed, and endothelial cell isolation was done by Dr. Ann Mansur (CAPCR protocol ID 13-6009BE). Briefly, human bAVM tissue was first digested mechanically with blades and rinsed with HBSS several times to remove blood. Digested tissue was then incubated with collagenase (Sigma, Cat #: C6885) at 37 °C for 20 min with periodic gentle shaking. After incubation, the tissue was further mixed and filtered through 100 μm strainers into 50 mL Falcon tubes. Cells were pelleted by centrifuging at 1700 rpm for 5 min at RT. Supernatant was removed, and the cell pellet was washed once with endothelial cell media (Lonza, Cat # CC-3202), followed by another centrifugation at 1700 rpm for 5 min at RT. Afterwards, the cell pellet was resuspended in endothelial cell media and seeded into 10-cm dishes at roughly 500,000 cells per plate. Cells were grown until 80% confluency prior to sorting in complete endothelial cell media. CD31 Dynabeads (Invitrogen, Cat #: 11155D) were used to isolate endothelial cells from the cell mixture. Aliquoted magnetic Dynabeads were washed once with 0.1% BSA in PBS and kept on ice in 0.1% BSA in PBS. The cell mixture was detached from the culturing plates using trypsin. The cells were pelleted at 1000 rpm for 5 min at RT. After spinning, the cell pellets were resuspended in 800 μL of 0.1% BSA in PBS and counted to ensure the cell concentration was <2,000,000 cells/mL. The cells were transferred to Eppendorf tubes and combined with 25 μL of washed beads in 0.1% BSA in PBS, followed by incubation on a rotator at 4 °C for 20 min. After incubation, another 200 μL of 0.1% BSA in PBS was added to the cell-beads mixture and placed on a magnetic rack for 2 min at RT. Afterwards, the supernatant was removed. For CD31-positive cells attached to magnetic beads, the beads were washed three times with 0.1% BSA in PBS. After the third wash, the beads were resuspended in endothelial cell media and seeded into 10 cm plates. The cells were cultured until confluency and frozen down for storage or seeded directly for experiments.

## Statistical analysis and data presentation

All statistical analyses were performed using GraphPad Prism 10 (GraphPad Software Inc., USA). Unless otherwise specified, comparisons between two groups were performed using two-tailed Student's *t* test (parametric) or Mann–Whitney test (non-parametric). Comparisons between more than two groups were done using one-way ANOVA with Tukey's post hoc test (parametric) or Kruskal–Wallis test with Dunn's multiple comparisons test (non-parametric). Normal distribution of datasets was tested using the D'Agostino-Pearson omnibus test, Anderson–Darling test, Shapiro–Wilk test, or Kolmogorov–Smirnov test with Dallal-Wilkinson-Lillie for *P* value. Statistical significance was considered when *P* values were <0.05. Statistical details are listed in the figure legend for each figure panel. All bar graphs depict mean ± standard deviation (SD), unless otherwise noted. Box and whisker plots show min to max. Details of RNA-sequencing and proteomics data analysis are described in the Supplementary Appendix.

## Graphics

Schematic and graphic figures were created using Biorender.com. Figure panels were generated using Adobe Illustrator.

# Data availability

Bulk RNA sequencing data is deposited at the Gene Expression Omnibus (GEO) repository (Barrett et al, 2013) with the accession number GSE292538. The TMT16 mass spectrometry proteomics data is deposited to the ProteomeXchange Consortium via the PRIDE partner repository (Perez-Riverol et al, 2025) with identifier PXD062375. The single-cell RNA-sequencing dataset was previously published (Wälchli et al, 2024) and deposited under the GEO accession number GSE256493.

The source data of this paper are collected in the following database record: biostudies:S-SCDT-10_1038-S44321-026-00383-y.

# Peer review information

# References

Alharbi S, Merkle S, Hammill AM, Waters AM, Le Cras TD (2025) RAS pathway mutations and therapeutics in vascular anomalies. Pediatr Blood Cancer 72(5):e31605

---

**The paper explained**

**Problem**

An arteriovenous malformation (AVM) is an abnormal tangle of blood vessels where arteries connect directly to veins. Over time, these vessels remodel and carry a lifelong risk of hemorrhage. Most brain AVMs are sporadic and are caused by somatic activating mutations in the *KRAS* gene in the endothelial cell (EC) lining of these vessels. While surgical treatment options are available, these can be very risky and are not an option for many patients. Novel non-invasive therapies are urgently needed.

**Results**

We extensively characterized the molecular alterations downstream of mutant KRAS signaling in ECs. We found that mutant KRAS reprograms cellular metabolism to elevate glycolytic flux, which is linked to various pathological phenotypes, such as dysregulated migratory and angiogenic capacities. This was primarily driven by upregulation of hexokinase 2 expression and increased membrane localization of glucose transporters. Targeting endothelial glycolysis, either alone or in combination with MEK inhibitors, provided promising therapeutic benefits in a zebrafish model of AVMs and in patient-derived ECs.

**Impact**

This study reveals how *KRAS* mutations drive AVM development and identifies endothelial glycolysis as a promising new treatment target. These findings could lead to safer, non-invasive alternatives to current surgical therapies for AVM patients.

Al-Olabi L, Polubothu S, Dowsett K, Andrews KA, Stadnik P, Joseph AP, Knox R, Pittman A, Clark G, Baird W et al (2018) Mosaic RAS/MAPK variants cause sporadic vascular malformations which respond to targeted therapy. J Clin Invest 128(4):1496–1508

Al-Samkari H, Eng W (2022) A precision medicine approach to hereditary hemorrhagic telangiectasia and complex vascular anomalies. J Thromb Haemost 20(5):1077–1088

Al-Shahi R, Fang JS, Lewis SC, Warlow CP (2002) Prevalence of adults with brain arteriovenous malformations: a community based study in Scotland using capture-recapture analysis. J Neurol Neurosurg Psychiatry 73(5):547–551

Awad MM, Liu S, Rybkin II, Arbour KC, Dilly J, Zhu VW, Johnson ML, Heist RS, Patil T, Riely GJ et al (2021) Acquired resistance to KRAS$^{G12C}$ inhibition in cancer. N Engl J Med 384(25):2382–2393

Bameri O, Salarzaei M, Parooie F (2021) KRAS/BRAF mutations in brain arteriovenous malformations: a systematic review and meta-analysis. Interv Neuroradiol 27(4):539–546

Barrett T, Wilhite SE, Ledoux P, Evangelista C, Kim IF, Tomashevsky M, Marshall KA, Phillippy KH, Sherman PM, Holko M et al (2013) NCBI GEO: archive for functional genomics data sets—update. Nucleic Acids Res 41(Database issue):D991–D995

Bhatt AN, Shenoy S, Munjal S, Chinnadurai V, Agarwal A, Vinoth Kumar A, Shanavas A, Kanwar R, Chandna S (2022) 2-deoxy-D-glucose as an adjunct to standard of care in the medical management of COVID-19: a proof-of-concept and dose-ranging randomised phase II clinical trial. BMC Infect Dis 22(1):669

Boon LM, Dekeuleneer V, Coulie J, Marot L, Bataille AC, Hammer F, Clapuyt P, Jeanjean A, Dompmartin A, Vikkula M (2022) Case report study of thalidomide therapy in 18 patients with severe arteriovenous malformations. Nat Cardiovasc Res 1(6):562–567

Cantelmo AR, Conradi LC, Brajic A, Goveia J, Kalucka J, Pircher A, Chaturvedi P, Hol J, Thienpont B, Teuwen LA et al (2016) Inhibition of the glycolytic activator PFKFB3 in endothelium induces tumor vessel normalization, impairs metastasis, and improves chemotherapy. Cancer Cell 30(6):968–985

Cooke DL, Frieden IJ, Shimano KA (2021) Angiographic evidence of response to trametinib therapy for a spinal cord arteriovenous malformation. J Vasc Anom 2(3):pe018

Cura AJ, Carruthers A (2010) Acute modulation of sugar transport in brain capillary endothelial cell cultures during activation of the metabolic stress pathway. J Biol Chem 285(20):15430–15439

De Bock K, Georgiadou M, Schoors S, Kuchnio A, Wong BW, Cantelmo AR, Quaegebeur A, Ghesquière B, Cauwenberghs S, Eelen G et al (2013) Role of PFKFB3-driven glycolysis in vessel sprouting. Cell 154(3):651–663

Debela DT, Muzazu SG, Heraro KD, Ndalama MT, Mesele BW, Haile DC, Kitui SK, Manyazewal T (2021) New approaches and procedures for cancer treatment: current perspectives. SAGE Open Med 9:20503121211034366

Derdeyn CP, Zipfel GJ, Albuquerque FC, Cooke DL, Feldmann E, Sheehan JP, Torner JC, American Heart Association Stroke Council (2017) Management of brain arteriovenous malformations: a scientific statement for healthcare professionals from the American Heart Association/American Stroke Association. Stroke 48(8):e200–e224

Du W, Ren L, Hamblin MH, Fan Y (2021) Endothelial cell glucose metabolism and angiogenesis. Biomedicines 9(2):147

El Sissy FN, Wassef M, Faucon B, Salvan D, Nadaud S, Coulet F, Adle-Biassette H, Soubrier F, Bisdorff A, Eyries M (2022) Somatic mutational landscape of extracranial arteriovenous malformations and phenotypic correlations. J Eur Acad Dermatol Venereol 36(6):905–912

Falkenberg KD, Rohlenova K, Luo Y, Carmeliet P (2019) The metabolic engine of endothelial cells. Nat Metab 1(10):937–946

Fish JE, Flores Suarez CP, Boudreau E, Herman AM, Gutierrez MC, Gustafson D, DiStefano PV, Cui M, Chen Z, De Ruiz KB et al (2020) Somatic gain of KRAS

function in the endothelium is sufficient to cause vascular malformations that require MEK but not PI3K signaling. Circ Res 127(6):727–743

Fraissenon A, Bayard C, Morin G, Benichi S, Hoguin C, Protic S, Zerbib L, Ladraa S, Firpion M, Blauwblomme T et al (2024) Sotorasib for vascular malformations associated with KRAS G12C mutation. N Engl J Med 391(4):334–342

Girard R, Zeineddine HA, Koskimaki J, Fam MD, Cao Y, Shi C, Moore T, Lightle R, Stadnik A, Chaudagar K et al (2018) Plasma biomarkers of inflammation and angiogenesis predict cerebral cavernous malformation symptomatic hemorrhage or lesional growth. Circ Res 122:1716–1721

Gungor S, Yakar Hİ (2021) Usefulness of dynamic fluorodeoxyglucose positron emission tomography/computed tomography in diagnosing pulmonary arteriovenous malformation mimicking a lung tumour. Interact Cardiovasc Thorac Surg 33(4):665–667

Hammer J, Seront E, Duez S, Dupont S, Van Damme A, Schmitz S, Hoyoux C, Chopinet C, Clapuyt P, Hammer F et al (2018) Sirolimus is efficacious in treatment for extensive and/or complex slow-flow vascular malformations: a monocentric prospective phase II study. Orphanet J Rare Dis 13(1):191

Hayer A, Shao L, Chung M, Joubert LM, Yang HW, Tsai FC, Bisaria A, Betzig E, Meyer T (2016) Engulfed cadherin fingers are polarized junctional structures between collectively migrating endothelial cells. Nat Cell Biol 18(12):1311–1323

Hong T, Yan Y, Li J, Radovanovic I, Ma X, Shao YW, Yu J, Ma Y, Zhang P, Ling F et al (2019) High prevalence of KRAS/BRAF somatic mutations in brain and spinal cord arteriovenous malformations. Brain 142(1):23–34

Huang L, Guo Z, Wang F, Fu L (2021) KRAS mutation: from undruggable to druggable in cancer. Signal Transduct Target Ther 6(1):386

Inoue J, Kishikawa M, Tsuda H, Nakajima Y, Asakage T, Inazawa J (2021) Identification of PDHX as a metabolic target for esophageal squamous cell carcinoma. Cancer Sci 112(7):2792–2802

Kalucka J, Bierhansl L, Conchinha NV, Missiaen R, Elia I, Brüning U, Scheinok S, Treps L, Cantelmo AR, Dubois C et al (2018) Quiescent endothelial cells upregulate fatty acid β-oxidation for vasculoprotection via redox homeostasis. Cell Metab 28(6):881–894.e13

Kim SH, Han SH, Song Y, Park CS, Song JJ (2017) Arteriovenous malformation of the external ear: a clinical assessment with a scoping review of the literature. Braz J Otorhinolaryngol 83(6):683–690

Kobialka P, Sabata H, Vilalta O, Gouveia L, Angulo-Urarte A, Muixí L, Zanoncello J, Muñoz-Aznar O, Olaciregui NG, Fanlo L et al (2022) The onset of PI3K-related vascular malformations occurs during angiogenesis and is prevented by the AKT inhibitor miransertib. EMBO Mol Med 14(7):e15619

Koga T, Suda K, Fujino T, Ohara S, Hamada A, Nishino M, Chiba M, Shimoji M, Takemoto T, Arita T et al (2021) KRAS secondary mutations that confer acquired resistance to KRAS G12C inhibitors, sotorasib and adagrasib, and overcoming strategies: insights from in vitro experiments. J Thorac Oncol 16(8):1321–1332

Krebs FS, Moura B, Missiaglia E, Aedo-Lopez V, Michielin O, Tsantoulis P, Bisig B, Trimech M, Zoete V, Homicsko K (2023) Response and resistance to trametinib in MAP2K1-mutant triple-negative melanoma. Int J Mol Sci 24(5):4520

Kurtoglu M, Gao N, Shang J, Maher JC, Lehrman MA, Wangpaichitr M, Savaraj N, Lane AN, Lampidis TJ (2007) Under normoxia, 2-deoxy-D-glucose elicits cell death in select tumor types not by inhibition of glycolysis but by interfering with N-linked glycosylation. Mol Cancer Ther 6(11):3049–3058

Lanman BA, Allen JR, Allen JG, Amegadzie AK, Ashton KS, Booker SK, Chen JJ, Chen N, Frohn MJ, Goodman G et al (2020) Discovery of a covalent inhibitor of KRASG12C (AMG 510) for the treatment of solid tumors. J Med Chem 63(1):52–65

Li H, Nam Y, Huo R, Fu W, Jiang B, Zhou Q, Song D, Yang Y, Jiao Y, Weng J et al (2021) De novo germline and somatic variants convergently promote

endothelial-to-mesenchymal transition in simplex brain arteriovenous malformation. Circ Res 129(9):825–839

Li X, Yang Y, Zhang B, Lin X, Fu X, An Y, Zou Y, Wang JX, Wang Z, Yu T (2022) Lactate metabolism in human health and disease. Signal Transduct Target Ther 7(1):305

Mansur A, Radovanovic I (2023) Vascular malformations: an overview of their molecular pathways, detection of mutational profiles and subsequent targets for drug therapy. Front Neurol 14:1099328

Mansur A, Radovanovic I (2024) Defining the role of oral pathway inhibitors as targeted therapeutics in arteriovenous malformation care. Biomedicines 12(6):1289

Moon JY, Ajebo EM, Gossage JR, Belcher MD (2022) Improvement of cutaneous hereditary hemorrhagic telangiectasia with pazopanib-A multikinase inhibitor. JAMA Dermatol 158(2):214–216

Nguyen HL, Boon LM, Vikkula M (2023) Trametinib as a promising therapeutic option in alleviating vascular defects in an endothelial KRAS-induced mouse model. Hum Mol Genet 32(2):276–289

Nikolaev SI, Vetiska S, Bonilla X, Boudreau E, Jauhiainen S, Rezai Jahromi B, Khyzha N, DiStefano PV, Suutarinen S, Kiehl TR et al (2018) Somatic activating KRAS mutations in arteriovenous malformations of the brain. N Engl J Med 378(3):250–261

Ou SI, Jänne PA, Leal TA, Rybkin II, Sabari JK, Barve MA, Bazhenova L, Johnson ML, Velastegui KL, Cilliers C et al (2022) First-in-human phase I/IB dose-finding study of adagrasib (MRTX849) in patients with advanced KRASG12C solid tumors (KRYSTAL-1). J Clin Oncol 40(23):2530–2538

Park ES, Kim S, Huang S, Yoo JY, Körbelin J, Lee TJ, Kaur B, Dash PK, Chen PR, Kim E (2021) Selective endothelial hyperactivation of oncogenic KRAS induces brain arteriovenous malformations in mice. Ann Neurol 89(5):926–941

Perez-Riverol Y, Bandla C, Kundu DJ, Kamatchinathan S, Bai J, Hewapathirana S, John NS, Prakash A, Walzer M, Wang S et al (2025) The PRIDE database at 20 years: 2025 update. Nucleic Acids Res 53(D1):D543–D553

Potter MD, Barbero S, Cheresh DA (2005) Tyrosine phosphorylation of VE-cadherin prevents binding of p120- and beta-catenin and maintains the cellular mesenchymal state. J Biol Chem 280(36):31906–31912

Priebe W, Zielinski R, Fokt I, Felix E, Radjendirane V, Arumugam J, Tai Khuong M, Krasinski M, Skora S (2018) EXTH-07. Design and evaluation of WP1122, an inhibitor of glycolysis with increased CNS uptake. Neuro Oncol 20:vi86

Priemer DS, Vortmeyer AO, Zhang S, Chang HY, Curless KL, Cheng L (2019) Activating KRAS mutations in arteriovenous malformations of the brain: frequency and clinicopathologic correlation. Hum Pathol 89:33–39

Quddus A, Karia P, Khurram R, Parthipun A, Brookes J (2021) Illuminating the nidus: The role of FDG PET/CT in high flow arteriovenous vascular malformations. Radiol Case Rep 16(6):1374–1377

Schindelin J, Arganda-Carreras I, Frise E, Kaynig V, Longair M, Pietzsch T, Preibisch S, Rueden C, Saalfeld S, Schmid B et al (2012) Fiji: an open-source platform for biological-image analysis. Nat Methods 9(7):676–682

Schmidt VF, Kapp FG, Goldann C, Huthmann L, Cucuruz B, Brill R, Vielsmeier V, Seebauer CT, Michel AJ, Seidensticker M et al (2024) Extracranial vascular anomalies driven by RAS/MAPK variants: spectrum and genotype-phenotype correlations. J Am Heart Assoc 13(8):e033287

Seebauer CT, Wiens B, Hintschich CA, Platz Batista da Silva N, Evert K, Haubner F, Kapp FG, Wendl C, Renner K, Bohr C et al (2024) Targeting the microenvironment in the treatment of arteriovenous malformations. Angiogenesis 27(1):91–103

Smith AJ, Clutton RE, Lilley E, Hansen KEA, Brattelid T (2018) PREPARE: guidelines for planning animal research and testing. Lab Anim 52(2):135–141

Solomon RA, Connolly ES (2017) Arteriovenous malformations of the brain. N Engl J Med 376(19):1859–1866

Soon K, Li M, Wu R, Zhou A, Khosraviani N, Turner WD, Wythe JD, Fish JE, Nunes SS (2022) A human model of arteriovenous malformation (AVM)-on-a-chip reproduces key disease hallmarks and enables drug testing in perfused human vessel networks. Biomaterials 288:121729

Suarez CF, Harb OA, Robledo A, Largoza GE, Ahn JJ, Alley EK, Wu T, Veeraragavan S, McClugage ST, Iacobas I et al (2024) MEK signaling represents a viable therapeutic vulnerability of KRAS-driven somatic brain arteriovenous malformations. Preprint at https://www.biorxiv.org/content/10.1101/2024.05.15.594335v1.full

Sugiyama T, Grasso G, Torregrossa F, Fujimura M (2022) Current concepts and perspectives on brain arteriovenous malformations: a review of pathogenesis and multidisciplinary treatment. World Neurosurg 159:314–326

Sun Z, Kemp SS, Lin PK, Aguera KN, Davis GE (2022) Endothelial k-RasV12 expression induces capillary deficiency attributable to marked tube network expansion coupled to reduced pericytes and basement membranes. Arterioscler Thromb Vasc Biol 42(2):205–222

Traver D, Paw BH, Poss KD, Penberthy WT, Lin S, Zon LI (2003) Transplantation and in vivo imaging of multilineage engraftment in zebrafish bloodless mutants. Nat Immunol 4(12):1238–1246

Triana P, Dore M, Cerezo VN, Cervantes M, Sánchez AV, Ferrero MM, González MD, Lopez-Gutierrez JC (2017) Sirolimus in the treatment of vascular anomalies. Eur J Pediatr Surg 27(1):86–90

Wälchli T, Bisschop J, Carmeliet P, Zadeh G, Monnier PP, De Bock K, Radovanovic I (2023) Shaping the brain vasculature in development and disease in the single-cell era. Nat Rev Neurosci 24(5):271–298

Wälchli T, Ghobrial M, Schwab M, Takada S, Zhong H, Suntharalingham S, Vetiska S, Gonzalez DR, Wu R, Rehrauer H et al (2024) Single-cell atlas of the human brain vasculature across development, adulthood and disease. Nature 632(8025):603–613

Wang X, Allen S, Blake JF, Bowcut V, Briere DM, Calinisan A, Dahlke JR, Fell JB, Fischer JP, Gunn RJ et al (2022) Identification of MRTX1133, a noncovalent, potent, and selective KRASG12D inhibitor. J Med Chem 65(4):3123–3133

Warburg O (1925) The metabolism of carcinoma cells. J Cancer Res 9(1):148–163

Webb BA, Chimenti M, Jacobson MP, Barber DL (2011) Dysregulated pH: a perfect storm for cancer progression. Nat Rev Cancer 11(9):671–677

Wee S, Jagani Z, Xiang KX, Loo A, Dorsch M, Yao YM, Sellers WR, Lengauer C, Stegmeier F (2009) PI3K pathway activation mediates resistance to MEK inhibitors in KRAS mutant cancers. Cancer Res 69(10):4286–4293

Winkler EA, Birk H, Burkhardt JK, Chen X, Yue JK, Guo D, Rutledge WC, Lasker GF, Partow C, Tihan T et al (2018) Reductions in brain pericytes are associated with arteriovenous malformation vascular instability. J Neurosurg 129(6):1464–1474

Winkler EA, Kim CN, Ross JM, Garcia JH, Gil E, Oh I, Chen LQ, Wu D, Catapano JS, Raygor K et al (2022) A single-cell atlas of the normal and malformed human brain vasculature. Science 375(6584):eabi7377

World Medical Association (2013) World Medical Association Declaration of Helsinki: ethical principles for medical research involving human subjects. JAMA 310(20):2191–2194

Wu D, Harrison DL, Szasz T, Yeh CF, Shentu TP, Meliton A, Huang RT, Zhou Z, Mutlu GM, Huang J et al (2021) Single-cell metabolic imaging reveals a SLC2A3-dependent glycolytic burst in motile endothelial cells. Nat Metab 3(5):714–727

Yazdani S, Bilan PJ, Jaldin-Fincati JR, Pang J, Ceban F, Saran E, Brumell JH, Freeman SA, Klip A (2022) Dynamic glucose uptake, storage, and release by human microvascular endothelial cells. Mol Biol Cell 33(12):ar106

Yetkin-Arik B, Vogels IMC, Nowak-Sliwinska P, Weiss A, Houtkooper RH, Van Noorden CJF, Klaassen I, Schlingemann RO (2019) The role of glycolysis and mitochondrial respiration in the formation and functioning of endothelial tip cells during angiogenesis. Sci Rep 9(1):12608

## Acknowledgements

Bulk RNA-sequencing was performed as a service by Novogene Corporation, Inc (Sacramento, CA). The proteomics study was conducted with the service and support from the Network Biology Collaborative Centre (NBCC; Lunenfeld-Tanenbaum Research Institute, Toronto; RRID: SCR_025375). The authors thank Cassandra Wong and Brett Larsen at NBCC for their help with the proteomics sample processing and LC-MS. The facility is supported by the Canada Foundation for Innovation and the Ontario Government. Additionally, the authors thank Kai Ellis (University of Toronto), Steven Botts (University of Toronto), Dr. Uros Kuzmanov (University of Toronto) and Dr. Sasha A. Singh (Brigham and Women's Hospital and Havard Medical School) for their guidance and consultation about RNA-sequencing and proteomics data analysis. The authors thank Dr. Arnold Hayer (McGill University) and his team for providing the immortalized HUVEC cell lines and Dr. Minna Woo (University of Toronto) and her team for the use of their Seahorse Instrument. All confocal microscopy was performed in the Advanced Optical Microscopy Facility, University Health Network, with training and access provided by the staff members. All schematic figures were created with Biorender.com. This work was supported by grants from the Canadian Institutes of Health Research (CIHR PJT155922 to IR, JDW and JEF and CIHR PJT173489 to JEF) and the National Institutes of Health (1R01HL159159 to JDW and JEF). RW received funding support from the following studentships: Canadian Graduate Scholarships – Doctoral (CGS D) – CIHR, Ontario Graduate Scholarship, Queen Elizabeth II/ Graduate Scholarships in Science and Technology - Heart and Stroke Foundation, and the PRiME Next-Generation Precision Medicine Fellowship program from the University of Toronto. NK received funding support from the following studentships: Canadian Graduate Scholarships – Doctoral (CGS D) – CIHR.

## Author contributions

**Ruilin Wu**: Conceptualization; Data curation; Formal analysis; Investigation; Visualization; Writing—original draft; Writing—review and editing. **Negar Khosraviani**: Investigation; Writing—review and editing. **Ann Mansur**: Investigation; Writing—review and editing. **Emilie Boudreau**: Formal analysis; Investigation; Writing—review and editing. **Gabrielle E Largoza**: Formal analysis; Investigation; Visualization; Writing—review and editing. **Suejean Park**: Investigation; Writing—review and editing. **Dakota Gustafson**: Investigation; Writing—review and editing. **Sneha Raju**: Investigation; Writing—review and editing. **Crizza Ching**: Investigation; Writing—review and editing. **Amira Klip**: Resources; Methodology; Writing—review and editing. **Thomas Wälchli**: Resources; Writing—review and editing. **Kathryn L Howe**: Supervision; Writing—review and editing. **Ivan Radovanovic**: Resources; Supervision; Writing—review and editing. **Joshua D Wythe**: Supervision; Funding acquisition; Writing—review and editing. **Jason E Fish**: Conceptualization; Supervision; Funding acquisition; Visualization; Writing—original draft; Project administration; Writing—review and editing.

Source data underlying figure panels in this paper may have individual authorship assigned. Where available, figure panel/source data authorship is listed in the following database record: biostudies:S-SCDT-10_1038-S44321-026-00383-y.

## Disclosure and competing interests statement

The authors declare no competing interests.

# Expanded View Figures

**Figure EV1. Stable Dox-inducible KRAS cell line construction and validation.**

(A) Schematic representation of stable cell line generation. Immortalized human umbilical vein endothelial cells (IM-HUVECs) were electroporated with plasmid constructs containing a Doxycycline (Dox)-inducible wild-type (WT) or G12V-mutant *KRAS4A* expression cassette and a constitutive Neomycin resistance cassette, along with a piggyBac transposon vector to incorporate *KRAS* constructs into the genome of these cells. After electroporation, transfected cells were selected with G418 and expanded for future use. (B, C) qPCR of endothelial cell markers (*CD31, EGFL7, vWF, NOS3, CDH5*) and mesenchymal markers (*TWIST1, SOX2, SNAI1, KLF4, ACTA2*) between stable or control IM-HUVEC cell lines ($-/+$ Dox). Housekeeping genes: *TBP, GAPDH*. MAPK/ERK pathway activation: *SPRY4, EGR1*. Mean ± standard deviations (SD). One-way ANOVA with Tukey's post hoc tests. *n* (independent experiments) = 4. (D) Representative western blots showing mScarlet-tagged KRAS expression in stable IM-HUVEC KRAS cell lines ($-/+$ Dox, 2 days). MEK inhibitor treatment: SL327 (10 µM) for 24 h. $n = 3$. (E) Representative western blots showing pERK, pAKT and KRAS levels in the stable IM-HUVEC KRAS$^{G12V}$ cell line over time from 15 min to 5 days. (F) Quantifications of western blots in (E). All groups were normalized to no Dox condition, except for the mScarlet/endogenous KRAS comparison. Mean ± SD. One-way ANOVA with Tukey's post hoc tests. $n = 3$ (for KRAS), $n = 4$ (for pERK and pAKT). In this figure, *$P$ value < 0.05, **$P$ value ≤ 0.001, ***$P$ value ≤ 0.0004 with exact $P$ values shown in Appendix Table S2.

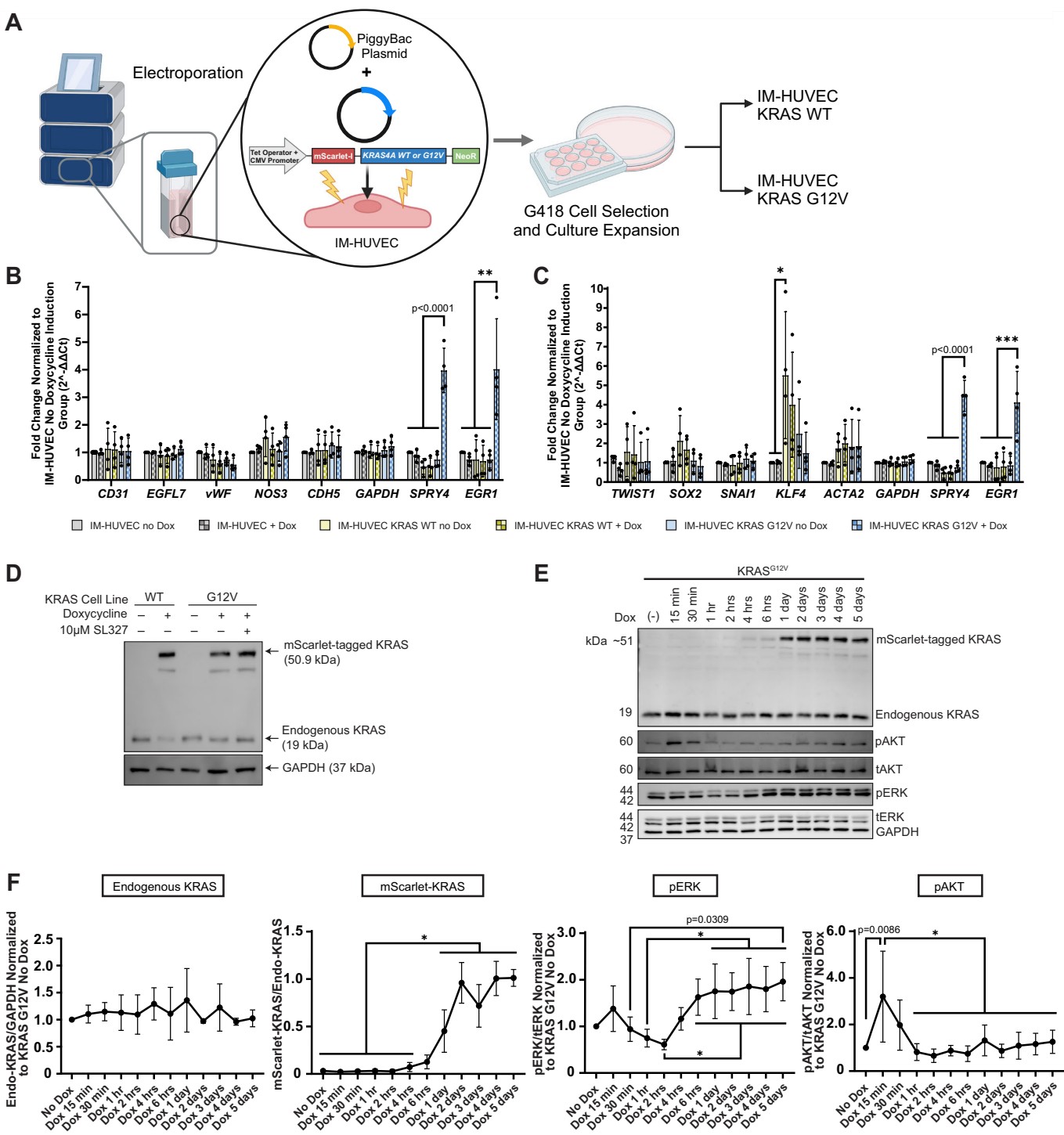

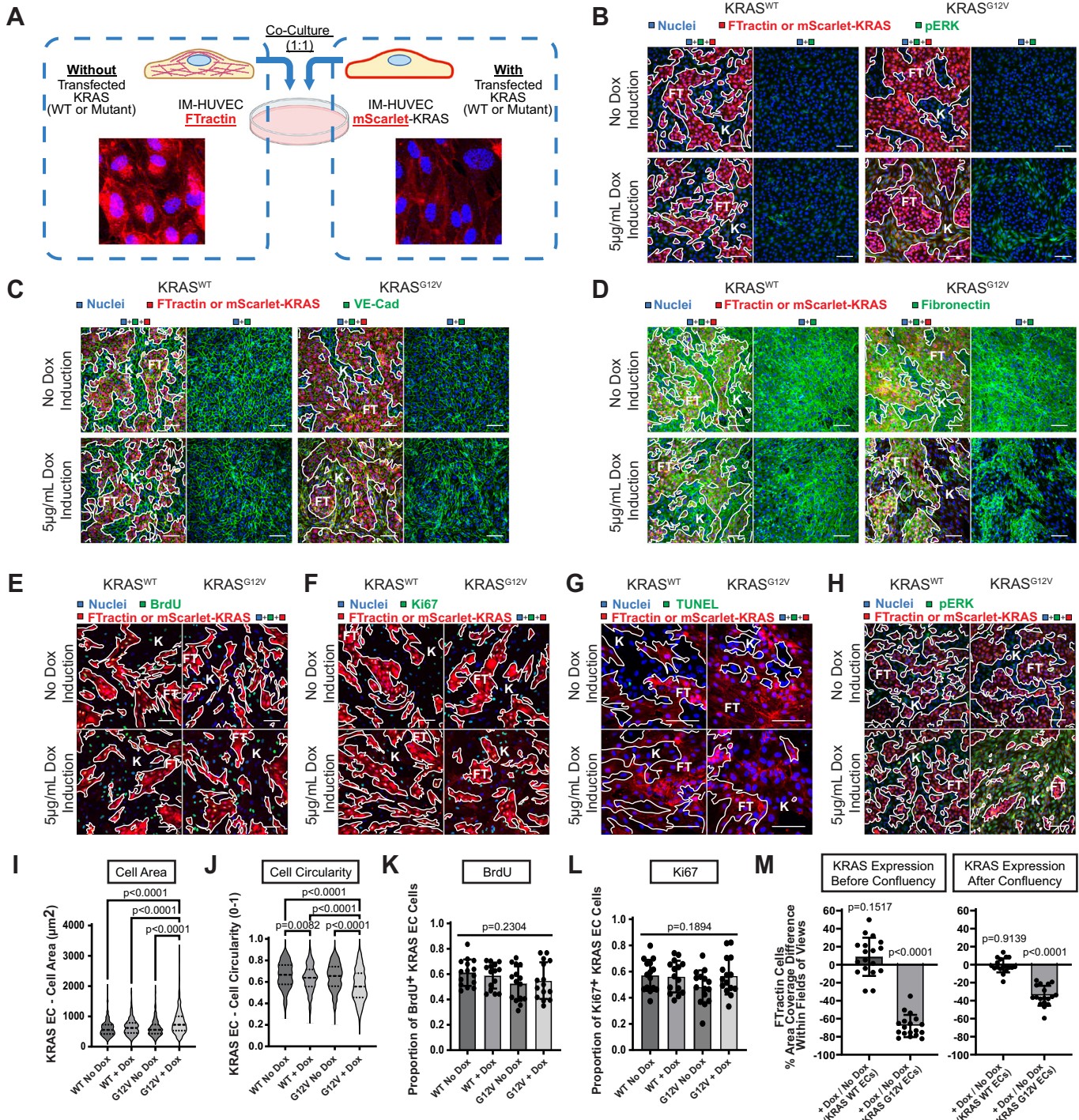

◄ **Figure EV2. Endothelial cells expressing mutant KRAS out-compete wild-type endothelial cells in a co-culture model.**

(A) Schematic of co-culturing IM-HUVEC KRAS and IM-HUVEC F-Tractin cells. Example images showing the difference between the F-Tractin and mScarlet-KRAS signals. (B–G) Representative IF staining of (B) pERK ($n = 3$), (C) VE-cadherin ($n = 3$), (D) Fibronectin (FN1) ($n = 3$), (E) BrdU ($n = 3$), (F) Ki67 ($n = 3$), (G) TUNEL ($n = 3$) within the co-culture system (−/+ Dox). Time of KRAS induction varies depending on the experiment: (i) proliferation assays (i.e., BrdU, Ki67) = 24 h; (ii) FN1 and TUNEL assays = 3 days; (iii) pERK and VE-cadherin assays = 4 days. White asterisks in (C) indicate examples of barrier gaps. Arrows in (D) indicate regions under WT ECs that have reduced FN1 levels. (H) Representative IF staining of pERK within the co-culture system showing changes to the areas occupied by IM-HUVEC F-Tractin cells after 4 days of co-culture with IM-HUVEC KRAS cells ($n = 3$). In all images, "FT" marks regions of IM-HUVEC F-Tractin cells, while "K" marks regions of IM-HUVEC KRAS WT or G12V cells. White solid lines show the boundary between IM-HUVEC KRAS and F-Tractin cells. All scale bars = 100 µm. (I, J) Quantifications of cell area (I) and circularity (J) by tracing VE-cadherin staining images from the same experiments as (C). Violin plots with dotted lines show interquartile range. One-way ANOVA with Tukey's post hoc tests. $n = 3$ (540 cells were quantified per condition). (K, L) Quantifications of proportions of (K) BrdU and (L) Ki67 positive IM-HUVEC KRAS cells from the same experiments as (E, F). Mean ± SD. One-way ANOVA tests. $n = 3$ (5 fields of view per condition per replicate). (M) Quantification of area occupied by IM-HUVEC F-Tractin cells after co-culture with IM-HUVEC KRAS cells (−/+ Dox). Two groups of experiments were conducted: (1) KRAS expression was induced right after seeding (prior to 100% confluency) (left); (2) KRAS expression was induced after the formation of confluent monolayers (right). Mean ± SD. Unpaired, two-tailed t tests. $n = 3$ (6 fields of view per condition per replicate).

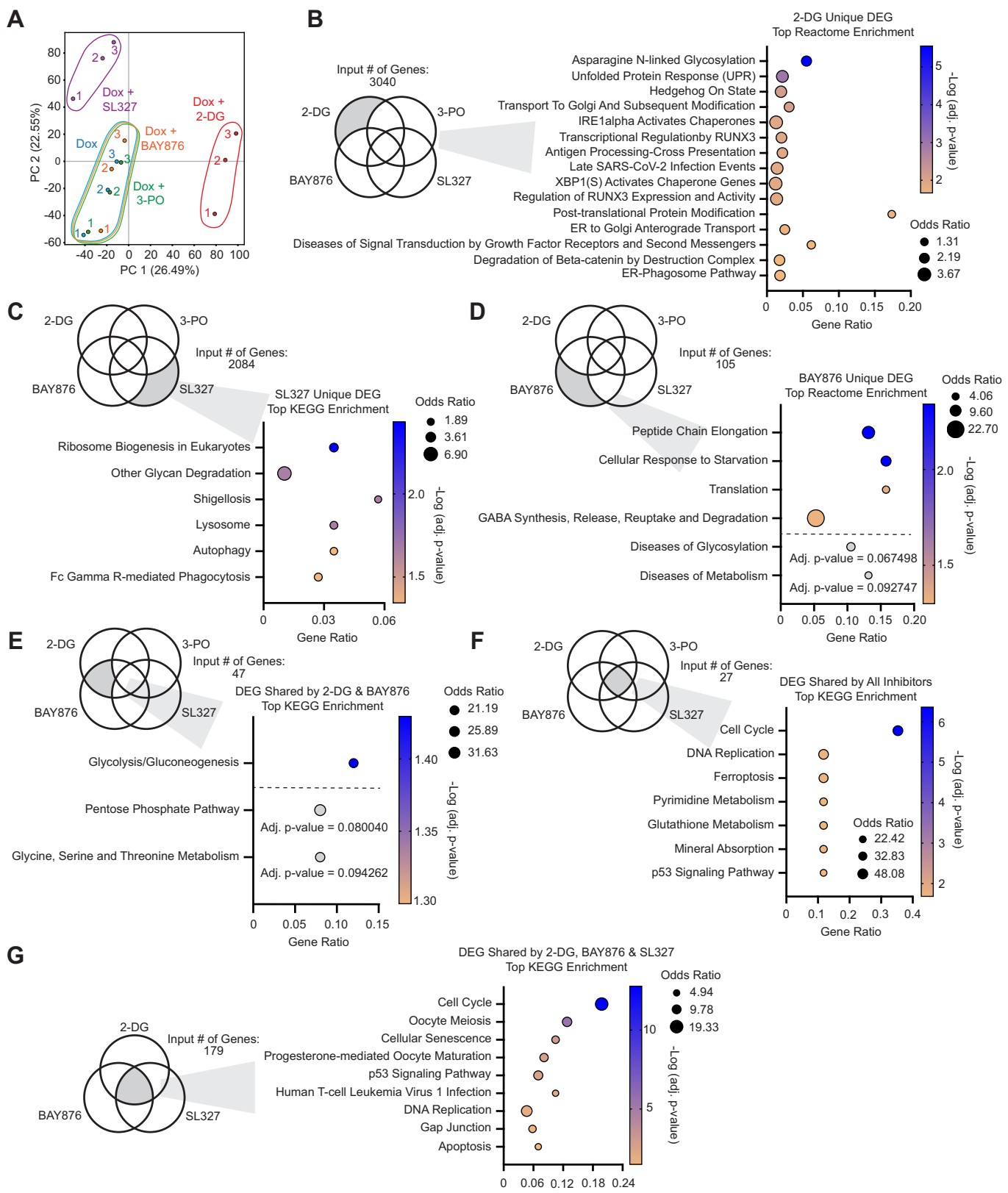

◀ **Figure EV3.  Pathway enrichment of transcriptional profiling data from KRAS^G12V ECs treated with glycolysis and MEK inhibitors.**

(A) RNA-seq Principal Component Analysis (PCA) plot of IM-HUVEC KRAS^G12V cells with Dox induction, comparing between no treatment and inhibitor treatments. Numbers in the graph represent experimental replicates. (B) Top Reactome pathway enrichments using unique DEGs (*P* value < 0.05) altered by 2-DG treatment, in comparison to no treatment. (C) Top Kyoto Encyclopedia of Genes and Genomes (KEGG) pathway enrichments using unique DEGs (*P* value < 0.05) altered by SL327 treatment, in comparison to no treatment. (D) Top Reactome pathway enrichments using unique DEGs (*P* value < 0.05) altered by BAY876 treatment, in comparison to no treatment. (E) Top KEGG pathway enrichments using shared DEGs (*P* value < 0.05) altered by both 2-DG and BAY876 treatments, in comparison to no treatment. (F) Top KEGG pathway enrichments using shared DEGs (*P* value < 0.05) altered by all treatments in comparison to no treatment. (G) Top KEGG pathway enrichments using shared DEGs (*P* value < 0.05) altered by all treatments, excluding 3-PO, in comparison to no treatment. All enrichments were done using Enrichr. Color of the circle presents -log10 of adjusted *P* value for pathway enrichment. Size of the circle represents odds ratio. Gene ratio was calculated as the number of genes used to enrich for each GO term divided by the total number of genes used to enrich for all GO terms. Venn diagrams highlight the set of genes used for pathway enrichment (shaded area), along with number of genes used for enrichment indicated. In all panels, the statistics for DEGs are calculated by DESeq2. Pathway enrichment statistics are computed by Enrichr, which utilizes Fisher exact tests with Benjamini–Hochberg corrections. Adjusted *P* value cutoff of <0.05 is considered significant for pathway enrichment.

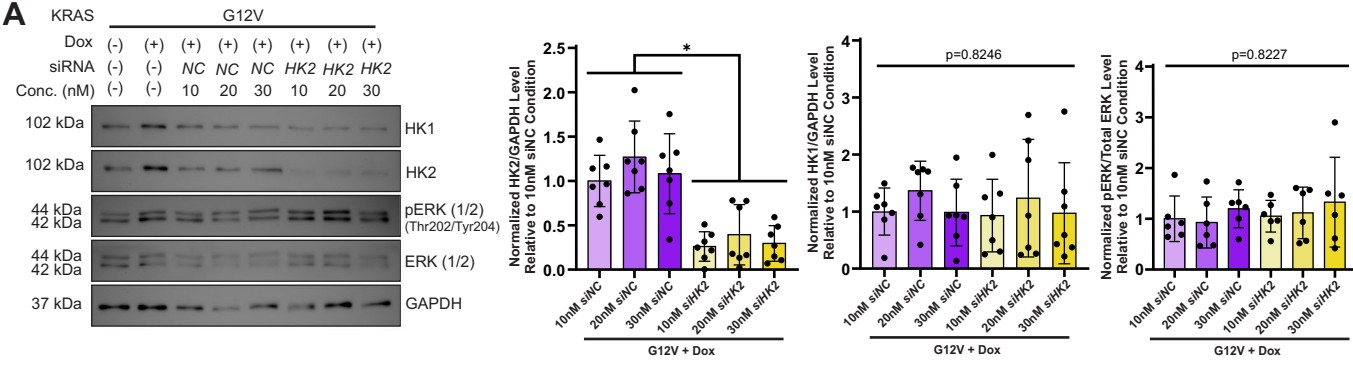

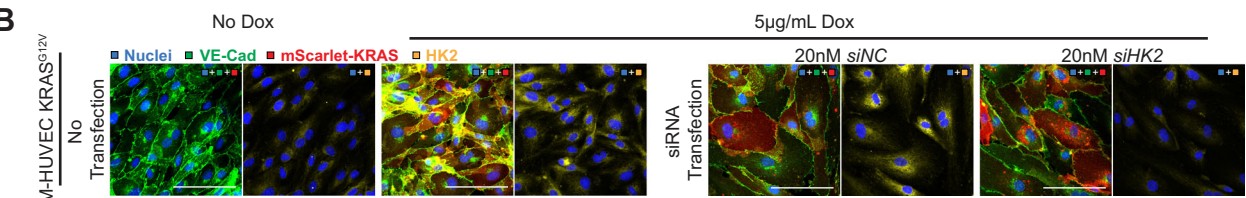

**Figure EV4. Confirmation of HK2 knock-down by siRNA.**

(A) Representative western blots of siHK2 knock down (10–30 nM) in IM-HUVEC KRAS^G12V cells (−/+ Dox induction). Mean ± SD. One-way ANOVA with Tukey's post hoc tests. HK1 and HK2 were normalized to GAPDH levels ($n = 7$). pERK was normalized to total ERK levels ($n = 6$). (B) Representative IF staining of VE-cadherin and HK2 with 20 nM siHK2 knock down in IM-HUVEC KRAS^G12V cells (−/+ Dox). $n = 2$. Scale bars = 100 μm. In this figure, *$P$ value ≤ 0.0150 with exact $P$ values shown in Appendix Table S2.

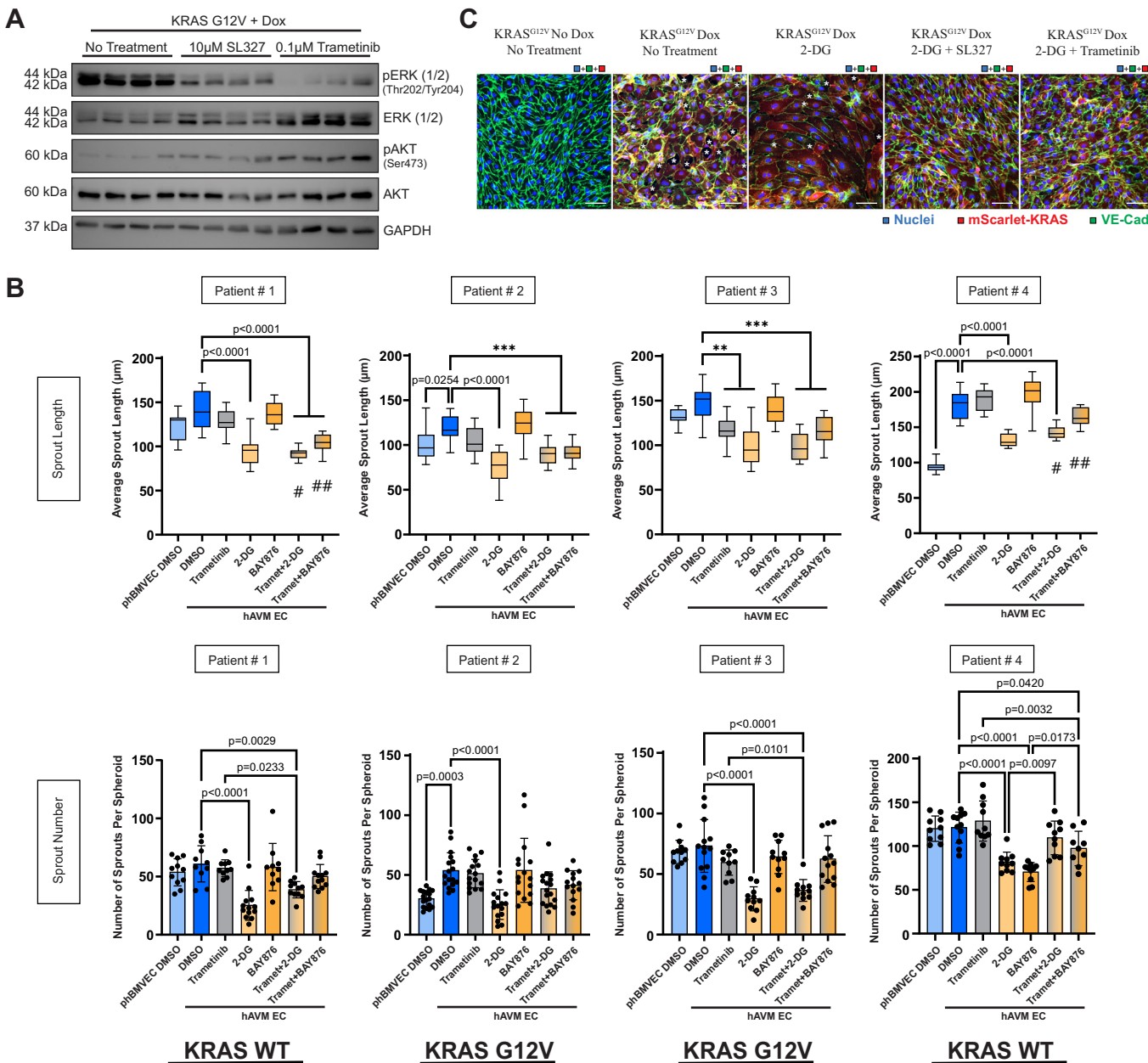

Figure EV5. MEK inhibition comparisons between SL327 and trametinib and patient-specific response to inhibitor treatments on bAVM EC spheroids sprouting.

(A) Western blots of cell lysate samples collected from IM-HUVEC KRAS[G12V] cells (5 μg/mL Dox), comparing between 10 μM SL327 and 100 nM Trametinib treatments. $n = 4$. (B) Quantifications of the bAVM EC spheroid sprouting assay, as shown in Fig. 7C, for each individual patient. Inhibitor treatments: 2-DG (2 mM), BAY876 (5 μM) and trametinib (100 nM) for 24 h. Bar graph: mean ± SD. Box and whiskers: min to max. One-way ANOVA with Tukey's post hoc test. Only selected statistics are shown. # represents significant $P$ value comparing 2-DG+Trametinib treatment with Trametinib treatment alone. ## represents significant $P$ values comparing BAY876+Trametinib treatment with both Trametinib or BAY876 treatment alone. $n = 4$ patients (9–17 spheroids per condition per replicate). (C) IF images showing VE-cadherin in the stable IM-HUVEC KRAS[G12V] cell line with 2 mM 2-DG treatment alone or 2-DG treatment combined with either 10 μM SL327 or 100 nM Trametinib for 2 days. White asterisks indicate examples of obvious gaps between cells. $n = 1$. Scale bars = 100 μm. In this figure, **$P$ value < 0.01, ***$P$ value ≤ 0.0003 with exact $P$ values shown in Appendix Table S2.

