## [Peer Review File · EMBO Molecular Medicine]

KRAS-Dependent Glycolytic Reprogramming of Endothelial Cells in Sporadic Arteriovenous Malformations

Ruilin Wu, Negar Khosraviani, Ann Mansur, Emilie Boudreau, Gabrielle Largoza, Suejean Park, Dakota Gustafson, Sneha Raju, Crizza Ching, Amira Klip, Thomas Wächli, Kathryn Howe, Ivan Radovanovic, Joshua Wythe, and Jason Fish

Corresponding author: Jason Fish (jason.fish@utoronto.ca)

Review Timeline:

Submission Date:	19th Jun 25
Editorial Decision:	10th Jul 25
Revision Received:	24th Oct 25
Editorial Decision:	21st Nov 25
Revision Received:	23rd Jan 26
Accepted:	27th Jan 26

Editor: Lise Roth

Transaction Report:

10th Jul 2025

Dear Dr. Fish,

Thank you for submitting your manuscript to EMBO Molecular Medicine. We have now received feedback from the three reviewers who agreed to evaluate your manuscript. As you will see from the reports below, the referees acknowledge the interest of the study and are overall supporting publication of your work pending appropriate revisions.

We further consulted with the referees regarding referee #2's request to confirm the findings in an endothelial-specific mouse model harboring the KRAS G12V mutation, as well as in primary ECs from patients with this mutation. While the addition of a new mouse model or patients' samples would be very valuable, we agreed that it would not be requested for further consideration. However, we would like you to strengthen the characterization of the treatment effects in the zebrafish model and discuss the limitations of your study in the manuscript text.

Addressing the reviewers' concerns in full will be necessary for further considering the manuscript in our journal, and acceptance of the manuscript will entail a second round of review. EMBO Molecular Medicine encourages a single round of revision only and therefore, acceptance or rejection of the manuscript will depend on the completeness of your responses included in the next, final version of the manuscript. For this reason, and to save you from any frustrations in the end, I would strongly advise against returning an incomplete revision.

We are expecting your revised manuscript within three months, if you anticipate any delay, please contact us.

We require:

Additional information on source data and instruction on how to label the files are available

4) A .docx formatted letter INCLUDING the reviewers' reports and your detailed point-by-point responses to their comments. As part of the EMBO Press transparent editorial process, the point-by-point response is part of the Review Process File (RPF), which will be published alongside your paper.

5) A complete author checklist, which you can download from our author guidelines (<https://www.embopress.org/page/journal/17574684/authorguide#submissionofrevisions>). Please insert information in the checklist that is also reflected in the manuscript. The completed author checklist will also be part of the RPF.

6) All Materials and Methods need to be described in the main text using our 'Structured Methods' format. According to this format, the Methods section includes a Reagents and Tools Table (listing key reagents, experimental models, software and relevant equipment and including their sources and relevant identifiers) followed by a Methods and Protocols section describing the methods, ideally using a step-by-step protocol format. The aim is to facilitate adoption of the methodologies across labs. Please download and fill our Reagents and Tools Table template (.docx), which you can find in our author guidelines: <https://www.embopress.org/page/journal/14693178/authorguide#structuredmethods>.

7) Please note that all corresponding authors are required to supply an ORCID ID for their name upon submission of a revised manuscript.

8) It is mandatory to include a 'Data Availability' section after the Materials and Methods. Before submitting your revision, primary

datasets produced in this study need to be deposited in an appropriate public database, and the accession numbers and database listed under 'Data Availability'. Please remember to provide a reviewer password if the datasets are not yet public (see <https://www.embopress.org/page/journal/17574684/authorguide#dataavailability>).

9) For data quantification: please specify the name of the statistical test used to generate error bars and P values, the number (n) of independent experiments (specify technical or biological replicates) underlying each data point and the test used to calculate p-values in each figure legend. The figure legends should contain a basic description of n, P and the test applied. Graphs must include a description of the bars and the error bars (s.d., s.e.m.). Please provide exact p values.

10) Our journal encourages inclusion of *data citations in the reference list* to directly cite datasets that were re-used and obtained from public databases. Data citations in the article text are distinct from normal bibliographical citations and should directly link to the database records from which the data can be accessed. In the main text, data citations are formatted as follows: "Data ref: Smith et al, 2001" or "Data ref: NCBI Sequence Read Archive PRJNA342805, 2017". In the Reference list, data citations must be labeled with "[DATASET]". A data reference must provide the database name, accession number/identifiers and a resolvable link to the landing page from which the data can be accessed at the end of the reference. Further instructions are available at .

11) We replaced Supplementary Information with Expanded View (EV) Figures and Tables that are collapsible/expandable online. EV Figures should be cited as 'Figure EV1, Figure EV2' etc... in the text and their respective legends should be included in the main text after the legends of regular figures.

12) The paper explained: EMBO Molecular Medicine articles are accompanied by a summary of the articles to emphasize the major findings in the paper and their medical implications for the non-specialist reader. Please provide a draft summary of your article highlighting

13) Author contributions: CRedit has replaced the traditional author contributions section because it offers a systematic machine readable author contributions format that allows for more effective research assessment. Please remove the Authors Contributions from the manuscript and use the free text boxes beneath each contributing author's name in our system to add specific details on the author's contribution. More information is available in our guide to authors.

Please also suggest a visual abstract to illustrate your article as a PNG file 550 px wide x 300-600 px high. A cropped portion of this image will serve as thumbnail for the table of content on our webpage.

16) As part of the EMBO Publications transparent editorial process initiative (see our Editorial at <http://embomolmed.embopress.org/content/2/9/329>), EMBO Molecular Medicine will publish online a Review Process File (RPF) to accompany accepted manuscripts.

In the event of acceptance, this file will be published in conjunction with your paper and will include the anonymous referee

reports, your point-by-point response and all pertinent correspondence relating to the manuscript. Let us know whether you agree with the publication of the RPF and as here, if you want to remove or not any figures from it prior to publication. Please note that the Authors checklist will be published at the end of the RPF.

I look forward to receiving your revised manuscript.

Yours sincerely,

Lise Roth

**** Reviewer's comments ****

Referee #1 (Comments on Novelty/Model System for Author):

In this comprehensive and methodologically rigorous study, Wu et al. identify endothelial cell (EC) glycolysis as a novel therapeutic vulnerability in KRASG12V-driven arteriovenous malformations (AVMs), as demonstrated both in in vitro and preclinical in vivo models. The authors employ elegant and ambitious experimental strategies with clear translational relevance, making this work highly suitable for the readership of EMBO Molecular Medicine. To further enhance the translational impact of the study, I suggest the following revisions, as outlined to the authors.

Referee #1 (Remarks for Author):

In this study by Wu R et al, the authors elegantly and comprehensively identify EC glycolysis as a pathomechanism and therapeutic vulnerability in KRASG12V-driven AVMs. The authors demonstrate in rigorous in vitro and in vivo models that inhibition of glycolysis, either alone or in combination with MEK inhibition, could be a novel therapeutic approach for AVMs. The findings of the study are intriguing as they i) help understand underlying pathomechanisms in AVMs, ii) are of translational relevance and may iii) offer an additional or complementary approach to existing strategies. To further strengthen and extend the impact of the study, the following considerations are proposed:

Major:

With the exception of Figure 7B, the MEK-inhibitor SL327 instead of trametinib was used throughout the study. To robustly validate the hypothesis that dual targeting of EC glycolysis alone and in combination with MAPK/ERK signaling may provide a novel therapeutic vulnerability as well as synergistic benefits, a zebrafish study including trametinib alone and in combination with BAY876 would be helpful. This would enhance translational applicability and clarify potential synergy further supporting the data in Figure 7B.

Minor:

1. To specifically highlight the metabolic pathway implicated in AVM pathology, the authors may consider including the term glycolysis in the title.
2. The study would benefit from a discussion on whether the dependence on glycolysis represents a unique vulnerability in KRASG12V-mutant AVM ECs or a broader feature shared across other AVM genotypes and even vascular anomaly entities. Elucidating whether this metabolic dependency may be a common theme of EC pathology could have broader implications for the field of vascular anomalies and, specifically, precision medicine in VAs.

Referee #2 (Remarks for Author):

The study reveals KRAS-driven metabolic reprogramming in endothelial cells (ECs) of brain arteriovenous malformations (bAVMs), identifying HK2-mediated glycolysis as a key pathway-representing a novel therapeutic vulnerability not previously emphasized in AVM biology. The authors used a comprehensive experimental approach, combining in vitro models (immortalized and primary ECs), in vivo validation (zebrafish), omics analyses (RNA-seq and proteomics), and clinical samples

(human AVMs). The manuscript is well written, and the authors thoughtfully differentiate between MEK-dependent (HK2 induction) and MEK-independent (GLUT1 membrane localization) components of metabolic reprogramming, providing nuanced insight into KRAS signaling.

They also explore therapeutic implications, demonstrating the potential for dual inhibition of glycolysis and the MEK pathway to reverse pathological angiogenesis in AVMs. The reported synergy between BAY876 (a GLUT1 inhibitor) and trametinib (a MEK inhibitor) is particularly noteworthy. The inclusion of patient-derived ECs and tissues further enhances the clinical relevance of the findings.

However, I have a specific concern regarding the main conclusion. The PI3K-AKT pathway is a well-established regulator of glycolysis, enhancing it through multiple mechanisms-including upregulation and membrane translocation of GLUT1, which facilitates glucose uptake. It is also well documented that AKT activates hexokinases, particularly HK2, which phosphorylates glucose, committing it to the glycolytic pathway.

Although the authors report that MEK inhibitors reduce HK2 and GLUT1 expression at both the mRNA and protein levels, these results should be interpreted with caution. At the doses used, MEK inhibitors may have off-target effects, potentially influencing the PI3K-AKT pathway. It is widely recognized that oncogenic KRAS mutations activate PI3K-AKT signaling-a fact that has led to targeted therapeutic strategies such as those developed by BBO, with promising preclinical data recently published for KRAS-mutant cancers (PMID: 40504949).

Thus, the finding that KRAS mutations do not activate AKT in vitro is unexpected and may not accurately reflect in vivo biology or patient-derived cellular responses.

To support their claims, the authors must confirm all these findings using an endothelial-specific mouse model harboring the KRAS G12V mutation, as well as primary ECs from patients with this mutation.

Indeed, while the data presented are compelling, I believe the study underestimates the role of the AKT pathway in KRAS-mediated glycolysis and may have taken a direction that overlooks a key signaling axis associated with the Warburg effect. Further investigation into PI3K-AKT involvement is warranted to provide a more complete picture of KRAS-driven metabolic reprogramming in AVMs.

Referee #3 (Remarks for Author):

Sporadic Arteriovenous malformations are difficult to treat, with no approved pharmacological treatment options. The authors propose that glycolytic inhibition can effectively reverse pathological EC behaviors driven by mutant KRAS and may synergize with MEK inhibition. This may offer new therapeutic opportunities to minimize deleterious effects on healthy endothelial cells. This study extends earlier work by the researchers, where they have shown that KrasG12V is found in some AVMs and is associated with elevated MEK/ERK activity. Mechanisms underlying the effect of KrasG12V on endothelial cells are explored in the current study, using immortalized cell line, patient-derived endothelial cells and a zebrafish model. Extensive functional, histological, transcriptomic and proteomic assessments have been conducted.

Overall, the study was well-devised and advances our understanding of the role of Kras in endothelial cell metabolic programming. The authors provide numerous complementary types of evidence to build support for their conclusions. The inclusion of patient tissue analyses, and investigation of cells derived from patient tissue, is a strength of the study. The authors' main findings are that the functional effects of the KrasG12V mutation in promoting increased glycolysis, basement membrane degradation, migration, junctional organization and permeability can be improved either by MEK inhibition or by inhibition of glycolysis. However, the extent to which these treatments would benefit pre-established AVMs without harming normal vasculature will require more investigation.

There are several issues that the authors could address to strengthen the study.

1. In most experiments (ie. Fig. 1A, B, E, F, G, H, I), it would be useful to have clearer information about the duration of Dox induction prior to the assessment. Does the phenotype become more severe the longer KRAS mutant is expressed? For example, the change in VE cadherin is minimal (approx. 75% of no DOX based on the graph) and change in dextran leakage is also minimal. Was a time course done to ensure that maximal effect was detected?

2. The significance/relevance of the dot blots shown in Fig. 1E is not well explained, within the context of the rest of the study. This analysis appears to be a tangent that distracts from the main emphasis of the study. [Furthermore, were these outcomes validated by multiple biological replicates?]

3. The authors state that Kras mutant EC have a competitive advantage over WT EC independent of proliferation or apoptosis, based on data in Fig. S3. I find the lack of effect on proliferation to be surprising. Although I recognize that this outcome matches data published earlier from the group (Circ Res 2020), the transcriptomic data in the current study suggest that proliferation-associated genes are in fact enriched in the mutant EC, and affected by the drug treatments. Furthermore, increased glycolytic flux is well-associated with cell proliferation. Some additional in vitro proliferation assays would clarify this finding.

4. Figure 2 - what is the authors' explanation for the effect of SL327 treatment on decreasing glucose uptake (panel D) but not affecting lactate secretion (panel H)?

5. In the results text, the authors state that 2-DG treatment appeared to improve monolayer and junctional organization. And that BAY876 treatment did not restore VE-cadherin junctions. And that HK2 knockdown improved monolayer and junctional organization in mutant KRAS ECs. What data are these statements based on, considering that VE-cadherin staining was greatly reduced (Fig. S16 panel A, C) or seemingly unaltered (Fig. S16 panel B). Considering the ambiguity of the immunostaining, it seems that these statements are insufficiently justified. Was permeability affected?

6. It is extremely difficult to visualize the shunt structures (or their recovery) at the arrow sites in Figure 7A. Ideally, the magnification or the resolution of these images could be improved, which would help the reader to appreciate the effects being described by the authors.

7. Importantly, what does the recovery of a shunt represent at the cellular level? An altered cell size or sprouting? What is the impact of 2-DG on overall embryo development? I would think that cell proliferation and migration would be affected globally by this treatment (and by MEK inhibition), limiting the capacity of any cell to undergo remodeling. More detail regarding the overall impact of the drug treatments on the vascular growth would be useful.

Minor points:

1. Sprouting images in Fig. 6A are not very clear in terms of displaying sprouting. I recommend zooming in to make the sprouting easier to see (similar to Fig. 6D).

2. There is not a lot of evidence presented in the current paper to support the authors' conclusion that KRAS activity leads to increased extracellular matrix degradation- simply fibronectin and HABP immunostaining. In contrast, studies have shown that increasing glycolysis in tumor cells promotes matrix accumulation (<https://doi.org/10.1016/j.canlet.2024.217156>). Could the authors strengthen their observations through assessment of MMPs or analyses of degradation fragments in the culture media?

We thank the Editor and Reviewers for their constructive and helpful comments on our manuscript. We have revised the manuscript and have addressed all of the reviewers' concerns. In particular, we have acquired new data that shows the impact of combinatorial MEK and glycolysis inhibition in zebrafish (revised Figure 7B). We have also provided additional information regarding the timing of KRAS induction and subsequent molecular changes in our inducible *in vitro* system (revised Figure EV1E,F) and have provided information on the KRAS mutational status of patient-derived endothelial cells in the manuscript (revised Figure EV5B). We have improved the visualization of zebrafish phenotypes (revised Figure 7A) and provide staining of HK2 in the brain tissue of wild-type and mutant KRAS^{G12D} mice (revised Figure 5D). We also provide data to the reviewers that assesses the impact of AKT signaling in the regulation of HK2 induction and glucose uptake (Reviewer Figure 2). We also provide data to the reviewers using a 3D fibrin droplet vascular network assay that shows that combinatorial treatment with MEK and glycolysis inhibitors disrupts KRAS phenotypes (Reviewer Figure 1). Additionally, we have responded to all the editorial requirements. The original comments from the Editor and Reviewers are included, and our response is in red text. The revisions have improved our manuscript and strengthened our conclusions. We hope that the manuscript is now acceptable for publication in *EMBO Molecular Medicine*.

Comments from Editor:

"Thank you for submitting your manuscript to EMBO Molecular Medicine. We have now received feedback from the three reviewers who agreed to evaluate your manuscript. As you will see from the reports below, the referees acknowledge the interest of the study and are overall supporting publication of your work pending appropriate revisions."

We thank the Editor and Reviewers for their supportive comments. We have fully responded to the comments provided. This has further strengthened our manuscript.

"We further consulted with the referees regarding referee #2's request to confirm the findings in an endothelial-specific mouse model harboring the KRAS G12V mutation, as well as in primary ECs from patients with this mutation. While the addition of a new mouse model or patients' samples would be very valuable, we agreed that it would not be requested for further consideration. However, we would like you to strengthen the characterization of the treatment effects in the zebrafish model and discuss the limitations of your study in the manuscript text."

We appreciate this further clarification of the request from reviewers. While we are not able to include data from a KRAS-G12V mouse model given time restraints, we used our endothelial-specific KRAS^{G12D} mouse model and have stained HK2 in brain sections of control and mutant KRAS mice, which showed elevated HK2 expression in the brain blood vessels of mutant KRAS animals (revised Figure 5D). This data suggests that the mouse model can be used for further studies on glycolysis in future studies. Further, we have added information about the primary patient-derived ECs that were used in the manuscript. Their KRAS mutational status is now indicated (revised Figure EV5B). Importantly, patient ECs containing KRAS mutations appeared to be more sensitive to MEK inhibitor treatment, although the number of each is low (n=2 each). Interestingly, the combination of MEK and glycolysis inhibition appeared to be

effective, regardless of the mutational status (revised Figure 7C; revised Figure EV5B). This could be highly relevant to translation of our findings to the clinic. As requested, we have further characterized the treatment effects in the zebrafish model, including combination treatments of MEK and glycolysis inhibition (revised Figure 7B). We have also included images where the visualization of arteriovenous shunts is clearer (revised Figure 7A). The images have been better annotated to make the figure easier to understand. We have also expanded the limitations section of our study.

In addition to these revisions, we have responded to all of the editorial requests, including providing source data, a reagents and tools table and 'The Paper Explained' sections.

Referee #1 (Comments on Novelty/Model System for Author):

"In this comprehensive and methodologically rigorous study, Wu et al. identify endothelial cell (EC) glycolysis as a novel therapeutic vulnerability in KRASG12V-driven arteriovenous malformations (AVMs), as demonstrated both in in vitro and preclinical in vivo models. The authors employ elegant and ambitious experimental strategies with clear translational relevance, making this work highly suitable for the readership of EMBO Molecular Medicine. To further enhance the translational impact of the study, I suggest the following revisions, as outlined to the authors."

We thank the reviewer for their positive assessment of the quality and impact of our work.

Referee #1 (Remarks for Author):

"In this study by Wu R et al, the authors elegantly and comprehensively identify EC glycolysis as a pathomechanism and therapeutic vulnerability in KRASG12V-driven AVMs. The authors demonstrate in rigorous in vitro and in vivo models that inhibition of glycolysis, either alone or in combination with MEK inhibition, could be a novel therapeutic approach for AVMs. The findings of the study are intriguing as they i) help understand underlying pathomechanisms in AVMs, ii) are of translational relevance and may iii) offer an additional or complementary approach to existing strategies. To further strengthen and extend the impact of the study, the following considerations are proposed:"

Major:

"With the exception of Figure 7B, the MEK-inhibitor SL327 instead of trametinib was used throughout the study. To robustly validate the hypothesis that dual targeting of EC glycolysis alone and in combination with MAPK/ERK signaling may provide a novel therapeutic vulnerability as well as synergistic benefits, a zebrafish study including trametinib alone and in combination with BAY876 would be helpful. This would enhance translational applicability and clarify potential synergy further supporting the data in Figure 7B."

We thank the reviewer for this suggestion. In response to the comment from the reviewer, we have examined the effect of combined MEK and glycolysis inhibitor treatments in zebrafish (revised Figure 7B). Importantly, we used a five-fold lower dose of these inhibitors (0.2 μ M of SL327, 40 μ M of 2-DG, and 0.2

μM of BAY876) compared to when they were used individually in Figure 7A to examine if a synergistic benefit could be observed with lowered drug concentrations of each. This may have translational value because of reduce drug toxicity. In this experiment, we again used SL327 to match the initial single inhibitor data that was included in the original submission, thus allowing for comparative insights of our results. Because of the need to perform these experiments in KRAS-G12V-injected embryos with observable shunts, these experiments were labor-intensive and time consuming. Unfortunately, we were not able to fully test trametinib alone or in combination in zebrafish. We believe that the SL327 data provides valuable proof-of-concept, which can be expanded in the future using trametinib. Importantly, SL327 has been shown to inhibit MEK in zebrafish by us and others¹⁻⁴. In contrast, examples of the use of trametinib in zebrafish are scant. We found two publications with trametinib treatment in zebrafish using concentrations ranging from 0.1-0.4 μM . The inhibitor was dosed directly in E3 (embryo) media for both papers. One paper (*MAP2K1* mutation in ECs⁵) showed that a concentration of 0.4 μM led to a greater recovery of arteriovenous shunts in zebrafish, while the other (*NRAS* mutation in lymphatic ECs⁶) suggested that 0.4 μM was highly toxic. In our hands, we also found that higher concentrations of trametinib were toxic to the embryos, resulting in curved bodies, no blood flow and small hearts (data not shown). Further pharmacological testing is needed before an appropriate dose of trametinib (i.e., a therapeutic window) can be found that does not inhibit normal vascular development. We previously did this type of careful dose finding for SL327¹. We have mentioned the use of SL327 rather than the more clinically-relevant MEK inhibitor, trametinib, in the limitations section.

Revised Figure 7B. Combinatorial MEK and Glycolysis Treatment Improves Arteriovenous Shunt Recovery in Zebrafish. Tg(*gata1:dsRed*) zebrafish injected with GFP-tagged KRAS^{G12V} constructs and imaged at 2 days-post-fertilization (dpf). Inhibitor treatments were initiated at 2 dpf (dosed in water) after confirming the presence of arteriovenous shunts. Zebrafish were imaged before and after inhibitor treatment (representative images shown; left panels). Inhibitor concentrations were reduced by 5-fold in comparison to previous single inhibitor treatment. Arrows indicated locations of shunts in the images and direction of blood flow. Scale bars = 100 μm , 20X images. Quantifications of zebrafish shunt recovery with combined treatments shown on the right. Numbers on the graph indicate the total numbers of zebrafish embryos analyzed for each treatment condition. Box and whiskers: min to max. One-way ANOVA with Tukey's post hoc test. n=14-16 independent experiments. ns = not significant, ** p-value<0.01.

With our combined treatment experiment using lower doses of SL327 (0.2 μ M), 2-DG (40 μ M) and BAY876 (0.2 μ M), we observed that SL327+BAY876 led to a greater proportion of embryos with arteriovenous shunt recovery (total number of recoveries across all experiments = 60.0% [78/130]) compared to SL327 treatment alone (41.8% [56/134]), and this improvement was statistically significant (revised Figure 7B). SL327+2-DG treatment yielded total recoveries across all experiments of 50.8% (62/122), however this did not achieve statistical significance compared to SL327 treatment alone, although there was a trend towards improvement. It is worth noting that the reduced concentration of SL327 used in this combination study resulted in reduced proportions of shunt recovery compared to the high dose SL327 treatment (1 μ M) in Figure 7A. This new zebrafish data provides additional support that combining MEK and glycolysis inhibition can improve shunt recovery. Importantly, the drugs at these concentrations were well tolerated.

To further validate our findings, we also performed a combined treatment experiment using our KRAS^{G12V} EC line in a novel 3D vascular network model, which included ECs and supporting mural cells. Quantification showed that combined treatment led to significant reductions in average vessel length and increased numbers of end points (less connections), indicating overall decreased angiogenic capacity with combined treatment (Reviewer Figure 1). The combined treatment of SL327+BAY876 showed a greater effect than single inhibitor treatments. 2-DG, being a potent glycolysis inhibitor *in vitro*, suppressed angiogenesis by itself and disrupted network formation (data not shown), making it difficult to tease apart potential synergies. We are writing a manuscript on drug screening in 3D vascular models, and plan to include this data in this follow-up manuscript, so have only included the data as a reviewer figure.

Figure for reviewers removed

Overall, while further validation studies are needed in animal models with established AVMs (such as mouse models), our data supports the concept that combined MEK and glycolysis inhibition can provide synergistic therapeutic benefits. It is worth noting that metabolic inhibitors have not achieved widespread success in oncology, partly due to their inability to eradicate cancer cells. However, cell death and complete elimination are not necessarily desirable or needed in the therapeutic management of AVMs. Hence, repurposing glycolysis inhibitors for the treatment of AVMs represents a promising therapeutic opportunity, which should be further pursued in future studies.

Minor:

“1. To specifically highlight the metabolic pathway implicated in AVM pathology, the authors may consider including the term glycolysis in the title.”

Thank you for this suggestion. We have adjusted the title of the paper to “KRAS-Dependent Glycolytic Reprogramming of Endothelial Cells in Sporadic Arteriovenous Malformations”.

“2. The study would benefit from a discussion on whether the dependence on glycolysis represents a unique vulnerability in KRASG12V-mutant AVM ECs or a broader feature shared across other AVM genotypes and even vascular anomaly entities. Elucidating whether this metabolic dependency may be a common theme of EC pathology could have broader implications for the field of vascular anomalies and, specifically, precision medicine in VAs.”

We thank the reviewer for this suggestion, as this has importance for the applicability of our findings to multiple types of vascular anomalies. To begin to address this comment, we have obtained the mutation status of the patient samples tested in this study after all experiments/analyses were finished (the status was blinded at the time of the experiment). Patients #2 and #3 carry KRAS-G12V mutations in the CD31+ EC fraction, while patients #1 and #4 are KRAS-WT samples and presumably have mutations in other genes. Interestingly, based on the individual patient sample analysis shown in Expanded View Figure EV5B, it appears that the KRAS-G12V samples responded more robustly to trametinib treatment alone than KRAS-WT samples. Strikingly, however, combined treatment of trametinib and glycolysis inhibitors reduced angiogenic sprouting in all samples regardless of KRAS mutation status (revised Figure 7C; revised Figure EV5B). This may suggest that a combinatorial strategy can offer therapeutic benefits irrespective of the causative mutation. While this is discussed in the paper, we are careful to not over-interpret this data given the low number of samples (n=2 per group). Due to the limited number of bAVM surgeries during the revision period and the requirement for fresh tissue samples for the

isolation/culture of cells, we were not able to obtain additional samples with distinct mutations to expand this analysis. We have added this as a limitation of the study in the discussion section. Whether combinations of MEK and glycolysis inhibition will be helpful in other vascular anomalies should be tested in future studies. Importantly, many of these vascular anomalies appear to depend on precocious angiogenesis⁷⁻¹⁰, so they may very well be helpful as MEK/glycolysis inhibition normalizes angiogenesis. We have added some speculation regarding this to the Discussion.

Referee #2 (Remarks for Author):

“The study reveals KRAS-driven metabolic reprogramming in endothelial cells (ECs) of brain arteriovenous malformations (bAVMs), identifying HK2-mediated glycolysis as a key pathway-representing a novel therapeutic vulnerability not previously emphasized in AVM biology. The authors used a comprehensive experimental approach, combining in vitro models (immortalized and primary ECs), in vivo validation (zebrafish), omics analyses (RNA-seq and proteomics), and clinical samples (human AVMs). The manuscript is well written, and the authors thoughtfully differentiate between MEK-dependent (HK2 induction) and MEK-independent (GLUT1 membrane localization) components of metabolic reprogramming, providing nuanced insight into KRAS signaling.

They also explore therapeutic implications, demonstrating the potential for dual inhibition of glycolysis and the MEK pathway to reverse pathological angiogenesis in AVMs. The reported synergy between BAY876 (a GLUT1 inhibitor) and trametinib (a MEK inhibitor) is particularly noteworthy. The inclusion of patient-derived ECs and tissues further enhances the clinical relevance of the findings.”

We thank the reviewer for their positive assessment of the quality and impact of our study.

However, I have a specific concern regarding the main conclusion. The PI3K-AKT pathway is a well-established regulator of glycolysis, enhancing it through multiple mechanisms-including upregulation and membrane translocation of GLUT1, which facilitates glucose uptake. It is also well documented that AKT activates hexokinases, particularly HK2, which phosphorylates glucose, committing it to the glycolytic pathway.

Although the authors report that MEK inhibitors reduce HK2 and GLUT1 expression at both the mRNA and protein levels, these results should be interpreted with caution. At the doses used, MEK inhibitors may have off-target effects, potentially influencing the PI3K-AKT pathway. It is widely recognized that oncogenic KRAS mutations activate PI3K-AKT signaling-a fact that has led to targeted therapeutic strategies such as those developed by BBO, with promising preclinical data recently published for KRAS-mutant cancers (PMID: 40504949).

Thus, the finding that KRAS mutations do not activate AKT in vitro is unexpected and may not accurately reflect in vivo biology or patient-derived cellular responses.

To support their claims, the authors must confirm all these findings using an endothelial-specific mouse model harboring the KRAS G12V mutation, as well as primary ECs from patients with this mutation.”

We thank the reviewer for these thoughtful suggestions. The involvement of AKT in the phenotypes is addressed in response to the comment below. Regarding the comment about using primary ECs from patients with this mutation, we have confirmed the mutation status of the patient samples used in this study, which has provided us with insight into the potential applicability of our findings to a broad range of patients. Patients #2 and #3 carry the KRAS-G12V mutation and patients #1 and #4 are KRAS-WT samples (revised Figure EV5B). All mutations were detected in the CD31+ EC fraction. As mentioned in the response to the Editor, while we have not extensively utilized our mouse models for this project, we have provided an additional confirmation of HK2 induction in the brain blood vessels of EC-specific mutant KRAS^{G12D} mice, which has now been included in revised Figure 5D. We were not able to include further data from our mouse model or an EC-specific KRAS^{G12V} mouse model given time constraint, but this will be a focus of future studies.

Revised Figure 5D. HK2 Immunofluorescence Staining of Mouse Brain Tissue. *Cdh5-Cre^{ER}; Kras^{WT/WT}; R26^{LSL-tdTomato}* and *Cdh5-Cre^{ER}; Kras^{LSL-G12D/WT}; R26^{LSL-tdTomato}* mice brain sections (300µm thickness) were stained with anti-HK2 (cyan) after two months of induced KRAS expression in the endothelium. Representative images shown on the left. Magenta indicates cells in which the tdTomato reporter is expressed, indicating Cre-mediated recombination. Claudin 5 (yellow) was stained to indicate the brain endothelium and mice were perfused with lectin (grey). Scale bars = 50µm. Quantification of HK2 fluorescence intensity (A. U.) shown on the right. Vessels were divided into large vessels (diameter >15µm) and small vessels (diameter <15µm) for analysis. Mean ±SD. Unpaired multiple t-tests. n=3 animals per group. * p-value<0.05.

“Indeed, while the data presented are compelling, I believe the study underestimates the role of the AKT pathway in KRAS-mediated glycolysis and may have taken a direction that overlooks a key signaling axis associated with the Warburg effect. Further investigation into PI3K-AKT involvement is warranted to provide a more complete picture of KRAS-driven metabolic reprogramming in AVMs.”

We thank the reviewer for this suggestion. We agree with the reviewer that it is interesting and unexpected to find no alteration in the activity of the PI3K/AKT signaling axis in mutant KRAS ECs, given what is known from the oncology literature about KRAS signaling. It is important to highlight that this pathway activation pattern is not only observed in our immortalized cell line model, but also in patient-derived ECs and in tissue sections from patient bAVMs. Based on our initial investigation in bAVM patient

samples, pERK level was preferentially elevated in patient-derived ECs, but no change in pAKT/total AKT ratio was observed¹¹. In this regard, our immortalized cell line model recapitulates the same signaling pathway activation as clinical samples. We would also like to highlight that while MEK inhibition with SL327 was able to reverse arteriovenous shunts in our KRAS-dependent zebrafish AVM model, inhibiting AKT with LY294002, was not able to do so¹.

In response to the suggestion from the reviewer, we have further investigated the role of PI3K/AKT pathway in our observed metabolic changes. We utilized three PI3K inhibitors with various potencies – Copanlisib, Buparlisib and LY294002. These inhibitor studies did not reveal a direct relationship between inhibition of AKT signaling and HK2 expression. While LY294002 at 10 μ M reduced HK2 protein level, Copanlisib and Buparlisib did not affect HK2 levels to the same extent, despite both potently inhibiting pAKT (Reviewer Figure 2A). Please note, inhibitors were tested at multiple concentrations and 10 μ M of Copanlisib and Buparlisib led to cell death after overnight treatment (data not shown). Moreover, we observed that trametinib, being a potent MEK inhibitor, was able to elicit compensatory activation of the PI3K/AKT signaling over time, similar to what has been observed in cancer cells¹². After 72 hours of trametinib treatment, HK2 level was significantly reduced, despite pAKT signaling being elevated (Reviewer Figure 2B,C). This further indicates that HK2 expression is not directly linked to the activation of the PI3K/AKT pathway. Using the same inhibitors, we also performed a luminescent glucose uptake assay to assess the roles of PI3K and MAPK pathways in the regulation of glucose transport. We observed that both PI3K and MAPK inhibition showed a similar degree of reduction in EC glucose uptake, suggesting the involvement of both pathways in this process (Reviewer Figure 2D). Neither PI3K nor MAPK inhibition was able to abolish glucose uptake, as seen with the glucose transporter inhibitor, BAY876. However, since mutant KRAS does not increase AKT signaling in ECs, our observations suggest that AKT may be involved in basal glucose uptake rather than in the KRAS-dependent increases in glucose uptake, *per se*.

In an ongoing study, we have also conducted a high content phenotypic screen by treating mutant KRAS ECs with a library of 720 kinase inhibitors. Briefly, in the kinase library, there were 41 compounds targeting various components of the MAPK pathway and 68 compounds targeting the PI3K pathway. Among these, 22 out of 41 (~51%) MAPK inhibitors were able to rescue pathological phenotypes in mutant KRAS ECs, while only 4 out of 68 (~6%) PI3K inhibitors showed an effect (data not shown). The majority of effective MAPK inhibitors targeted either MEK1/2 or ERK1/2, while the ineffective MAPK inhibitors were targeting other MAPK signaling arms, such as ERK5, ASK1, etc. Furthermore, we shortlisted 2 out of the 4 PI3K inhibitors in a secondary screen using 3D vascular models (as shown in Reviewer Figure 1A). The two PI3K inhibitors that we tested were unable to affect angiogenic phenotypes, despite having high potencies for their targets, while MAPK inhibitors were able to significantly impact vascular network formation, demonstrating functional involvement of the MAPK pathway in EC angiogenesis (Reviewer Figure 2E). Overall, our published and unpublished studies have highlighted the importance of the MAPK/MEK/ERK signaling axis in driving pathological phenotypes in mutant KRAS ECs. We have included this data on PI3K inhibition for the reviewers but have not included this in the main manuscript as we think it may distract from the main message and some of this data will be incorporated into a future publication.

Collectively, our data suggests that the PI3K/AKT pathway, as a major signaling axis in the regulation of cellular homeostasis, does play a role in the regulation of cellular metabolism and inhibition of this pathway does suppress metabolic phenotypes. However, since its activity level is not altered under pathological conditions in the context of KRAS-driven AVMs, mutant KRAS does not seem to upregulate the PI3K pathway to drive the abnormal phenotypes observed in mutant ECs. In contrast, the MAPK/ERK signaling axis plays a prominent role in bAVMs.

A

Figure for reviewers removed

Figure for reviewers removed

D

Figure for reviewers removed

Reviewer Figure 2. PI3K Inhibitors in the Regulation of HK2 Expression, Glucose Uptake and Angiogenesis in Mutant KRAS^{G12V} ECs. (A) Western blots of IM-HUVEC KRAS^{G12V} cells (5µg/mL Dox) treated with three different PI3K inhibitors for 2 days – Copanlisib (100nM), Buparlisib (1µM) and LY294002 (10µM). Trametinib (MEKi, 100nM) was also included as a comparison group. n=1. (B) Representative time course western blots of IM-HUVEC KRAS^{G12V} cells (-/+5µg/mL Dox) treated with 0.1µM Trametinib from 3, 6, 24 and 72hrs. n=5. (C) Quantifications of the western blots shown in panel B. Values were normalized to the ‘no Dox, no treatment’ condition. Mean ±SD. One-way ANOVA with Tukey’s post hoc tests. n=5. (D) Luminescent 2-DG uptake assay with 3 days of mutant KRAS induction, including 1 day of inhibitor treatment, prior to the uptake experiment. Inhibitor treatments: Copanlisib (100nM), Buparlisib (1µM) and LY294002 (10µM), Trametinib (100nM) and BAY876 (5µM). Mean±SD. One-way ANOVA with Tukey’s post hoc test. n=4. (E) Representative images of the lumenized vascular network formed in the fibrin droplet assay, which were treated with 400nM of either MEK or PI3K inhibitors. IM-HUVEC cells with WT and mutant KRAS expression were mixed at 80:20 ratio, respectively. Top panel shows merged channel images of nuclei and VE-cadherin, while bottom panel shows VE-cadherin channel only of the same images. Scale bar = 200µm. n=5. In all figures, * p-value<0.05, ** p-value<0.01, *** p-value<0.001.

Referee #3 (Remarks for Author):

“Sporadic Arteriovenous malformations are difficult to treat, with no approved pharmacological treatment options. The authors propose that glycolytic inhibition can effectively reverse pathological EC behaviors driven by mutant KRAS and may synergize with MEK inhibition. This may offer new therapeutic opportunities to minimize deleterious effects on healthy endothelial cells. This study extends earlier work by the researchers, where they have shown that KrasG12V is found in some AVMs and is associated with elevated MEK/ERK activity. Mechanisms underlying the effect of KrasG12V on endothelial cells are explored in the current study, using immortalized cell line, patient-derived endothelial cells and a zebrafish model. Extensive functional, histological, transcriptomic and proteomic assessments have been conducted.

Overall, the study was well-devised and advances our understanding of the role of Kras in endothelial cell metabolic programming. The authors provide numerous complementary types of evidence to build support for their conclusions. The inclusion of patient tissue analyses, and investigation of cells derived from patient tissue, is a strength of the study. The authors' main findings are that the functional effects of the KrasG12V mutation in promoting increased glycolysis, basement membrane degradation, migration, junctional organization and permeability can be improved either by MEK inhibition or by inhibition of glycolysis. However, the extent to which these treatments would benefit pre-established AVMs without harming normal vasculature will require more investigation.”

We thank the reviewer for their positive assessment of our manuscript. We agree with the reviewer that it is not known whether inhibition of MEK and glycolysis will benefit pre-established complex AVMs without impacting the normal vasculature. We show initial data in the zebrafish model that shunts can be reversed without impacting normal vessels, but these represent early lesions rather than complex lesions. We mention this as one of the limitations of our study and agree that this should be investigated further in future studies.

There are several issues that the authors could address to strengthen the study.

“1. In most experiments (ie. Fig. 1A, B, E, F, G, H, I), it would be useful to have clearer information about

the duration of Dox induction prior to the assessment. Does the phenotype become more severe the longer KRAS mutant is expressed? For example, the change in VE cadherin is minimal (approx. 75% of no DOX based on the graph) and change in dextran leakage is also minimal. Was a time course done to ensure that maximal effect was detected?"

We thank the reviewer for this question regarding the kinetics of our inducible model. We previously demonstrated that KRAS^{G12V} expression in ECs led to increased MAPK/ERK activation as early as six hours following doxycycline (dox) induction¹³. In response to the reviewer's suggestion, we performed more granular evaluations of dox induction time course from 15 minutes to 5 days (revised Figure EV1E,F). The data reveals that pERK activation is increased to its maximum level as early as six hours following dox induction and the signaling activation is maintained over time with no further elevation, likely reaching a balance with negative feedback mechanisms. Interestingly, mutant KRAS expression in our cell line model does not reach its maximum level until 2 days following dox induction. This suggests that a high expression of the mutant protein in a cell is not required to elicit a full signaling response, even in the presence of wildtype KRAS proteins. This is interesting in light of the low allelic frequency/dosage of mutant KRAS in AVM patients. Importantly, there was no activation of pAKT pathway with long-term KRAS expression. We did observe an initial stimulation of pAKT (and pERK to a lesser degree) at 15 minutes. However, this effect dropped quickly and is likely attributed to the exposure and stimulation by doxycycline¹⁴. All experiments performed in Figure 1 were performed in cells subjected to 1-5 days of dox induction (note: detailed dox induction time is included in the Supplementary Methods for each assay), at which point, MAPK/ERK signaling reached its maximum level. Dox induction time is variable based on the experimental timelines of different assays to ensure that our findings are generalizable and not limited to a particular time point. It is also important to note that all characterization data was done with a KRAS^{WT} cell line that also received dox stimulation as a control comparison group.

Revised Figure EV1E,F. Dox-Induction Time Course in KRAS^{G12V} Cell Line. Representative western blots showing pERK, pAKT and KRAS levels in the stable IM-HUVEC KRAS^{G12V} cell line over time from 15 minutes to 5 days. Quantifications of western blots are shown at the bottom. All groups were normalized to no Dox condition, except for the mScarlet/endogenous KRAS comparison. Mean \pm SD. One-way ANOVA with Tukey's post hoc tests. n=3 (for KRAS), n=4 (for pERK and pAKT). * p-value<0.05.

Regarding the VE-cadherin data in particular, the experiment in Figure 1A was fixed and stained after 4 days of dox induction. During initial optimization and data generated for other experiments, we had stained for VE-cadherin after 2-4 days of dox induction with variable total culturing time and we did not observe any difference in phenotype based on different dox induction and culturing times (data not shown). The biggest difference we observed with VE-cadherin staining was between dox induction before and after the formation of a confluent monolayer, as mentioned in Appendix Figure S1A, although we did not further examine the mechanisms responsible for this phenotype for the current paper. While the quantification of VE-Cadherin intensity by immunofluorescence shows only a modest (~25%) change, the alteration of localization to junctions is much more dramatic, with much less VE-Cadherin present at junctions (Figure 1A). Since total levels of VE-Cadherin by western blot do not change (Figure 1C), it is likely that the quantification of immunofluorescence is underestimating the loss of adherens junctions. While the increase in FITC leakage may be modest (Figure 1B), it is highly reproducible and is likely biologically meaningful. We also see that VE-Cadherin localization to junctions is compromised in human bAVM patient samples (Figure 5F). In our mouse model, while mutant KRAS expression elevates the risks of hemorrhage, we do not observe spontaneous vessel rupture and hemorrhage at a high frequency. As such, we are not expecting the change in VE-cadherin to be causing severe vessel instability, but it does enhance leak, which may have long-term consequences.

“2. The significance/relevance of the dot blots shown in Fig. 1E is not well explained, within the context of the rest of the study. This analysis appears to be a tangent that distracts from the main emphasis of the study. [Furthermore, were these outcomes validated by multiple biological replicates?]”

We thank the reviewer for the opportunity to clarify this data. The dot blot shown in Figure 1E (measuring secreted factors in cell culture media) was repeated in three independent experiments with different passages/batches of cells. The associated quantification in Appendix Figure S1C shows each data point for the three independent experiments and associated statistical analyses. The cell lysate dot blot in Appendix Figure S1D was only done once as an experimental control to show that the pattern of expression of these secreted factors in media and lysate samples is distinct.

Figure 1 of the manuscript provides an overview of various aspects of EC biology, and we did not delve in-depth into the molecular mechanisms of all phenotypes. As we generated this mutant KRAS endothelial model ourselves and it has not been previously fully described, we performed thorough characterizations of this cell line in order to understand: 1) whether this cell line still behaves like endothelial cells after antibiotic-selection and extensive expansion (e.g. VE-cadherin junction staining, sprouting and angiogenic factor secretion, etc); 2) whether this cell line is able to recapitulate previously-known alterations in mutant KRAS ECs (e.g. VE-cadherin disruptions, increased angiogenic capacity, ECM degradation and elevated secretion of ANGPT2, etc); and 3) whether novel phenotypes in mutant KRAS

ECs could be discovered (out of which, we further investigated the metabolic phenotype in this paper). We feel that the data in Figure 1 provides context for readers to understand the model that we are working with and for better interpretation of experimental results. For example, CXCL8/IL8 is pro-angiogenic in tumor biology¹⁵ and we saw an increase in the secretion of CXCL8 in ECs as well, potentially implicating it in the pathobiology of AVMs. Similarly, PlGF was upregulated in our cellular model system, and is also known to be pathologically upregulated in cancer¹⁶. We therefore believe that this data is helpful in Figure 1E. However, we can move this data to an Appendix figure if desired.

“3. The authors state that Kras mutant EC have a competitive advantage over WT EC independent of proliferation or apoptosis, based on data in Fig. S3. I find the lack of effect on proliferation to be surprising. Although I recognize that this outcome matches data published earlier from the group (Circ Res 2020), the transcriptomic data in the current study suggest that proliferation-associated genes are in fact enriched in the mutant EC, and affected by the drug treatments. Furthermore, increased glycolytic flux is well-associated with cell proliferation. Some additional in vitro proliferation assays would clarify this finding.”

We thank the reviewer for this observation. It is indeed quite interesting that we do not observe an increased rate of cellular proliferation in mutant KRAS ECs. We have included cell doubling data in the manuscript as a measure of cell expansion (Figure 6C). Despite the lack of obvious alterations in cell doubling, we speculate that cell cycle regulation in mutant KRAS ECs may be altered, and this could have phenotypic consequences on EC behaviour. For example, Dr. Karen Hirschi’s team has demonstrated in their recent publications that ECs in early G1 versus late G1 phase respond differently to external stimulations, which then influences a cell’s propensity for arterial-venous specification^{17,18}. Since we do observe enrichment in cell cycle terms in our transcriptomic datasets and glycolytic metabolism is closely linked to cell cycle regulation, we will explore the contribution of cell cycle to cellular phenotypes, including cell competition, in future studies. To provide further insight into proliferation, bioinformatic analysis of our team’s recent scRNA-seq study involving ECs isolated from human pathologies revealed that unlike fetal tissue and tumor ECs, AVM ECs do not exhibit a global upregulation of proliferation signatures (Reviewer Figure 3). In particular, Ki67 expression is not different between temporal lobe (control) and AVM ECs. This suggests that mutant KRAS expression in ECs does not induce a strong proliferative signature.

Reviewer Figure 3. Human Single-Cell RNA-Sequencing of Endothelial Cells Showing Proliferation Markers Across Various Brain Pathologies. (A) Heatmaps (z-score) generated from published single-cell RNA-sequencing dataset by

Wälchli and Ghobrial et al (2024). mRNA expressions of proliferation markers were compared in ECs isolated from human adult temporal lobe (TL), fetal brain (fCNS), fetal peripheral tissue (fPeriph) and various brain pathologies, including arteriovenous malformations (AVM), meningioma (MEN), low-grade glioma (LGG), glioblastoma (GBM) and lung cancer metastasized to the brain (MET). **(B)** The same dataset as panel A, but without fetal tissue samples. Black box highlights the *MKI67* gene.

“4. Figure 2 - what is the authors' explanation for the effect of SL327 treatment on decreasing glucose uptake (panel D) but not affecting lactate secretion (panel H)?”

We thank the reviewer for this question. It is intriguing that lactate secretion was not affected by SL327 treatment, despite glucose uptake being reduced. In addition to the data included in Figure 2H, we also assessed a 2hr lactate secretion time point and the result was the same (data not shown). We do not yet know the molecular mechanisms involved in this observation. According to our RNA-sequencing results, *LDHA* and *LDHB* levels were significantly decreased (which may impact conversion of pyruvate to lactate), while the *ALDH1A1* gene was significantly upregulated with SL327 treatment (Reviewer Figure 4). We did not detect any significant changes in monocarboxylate transporter (MCT) genes (which are involved in lactate transport) upon SL327 treatment in our RNA-sequencing data. However, this does not rule out potential protein level or subcellular localization changes. In the oncology literature, *ALDH1A1* plays important roles in promoting glycolytic activation and consequently, increasing lactate levels^{19,20}. It is possible that the increase in *ALDH1A1* enhances lactate secretion, despite decreased glucose uptake. However, further comprehensive studies using radiotracers would be needed to investigate and validate the potential molecular mechanism(s) underlying alterations in lactate secretion upon SL327 treatment. We also cannot rule out the possibility that the shuttling of lactate to other metabolic pathways is being impacted by SL327 inhibition. However, these experiments are beyond the scope of the current manuscript. We have not included this information in the manuscript as we feel it may distract from the main findings of our work.

Reviewer Figure 4. Bulk RNA-Sequencing of KRAS^{G12V} Endothelial Cell Line Showing the Expression of the *ALDH1A1* Gene. (A) Dot blot (z-score normalization of gene counts) of the *ALDH1A1* gene in the bulk RNA-sequencing dataset with various inhibitor treatments, including 2mM 2-DG, 5μM BAY876 and 10μM SL327.

“5. In the results text, the authors state that 2-DG treatment appeared to improve monolayer and

junctional organization. And that BAY876 treatment did not restore VE-cadherin junctions. And that HK2 knockdown improved monolayer and junctional organization in mutant KRAS ECs. What data are these statements based on, considering that VE-cadherin staining was greatly reduced (Fig. S16 panel A, C) or seemingly unaltered (Fig. S16 panel B). Considering the ambiguity of the immunostaining, it seems that these statements are insufficiently justified. Was permeability affected?”

We thank the reviewer for the opportunity to clarify this data. We agree with the reviewer that the VE-cadherin phenotype in mutant KRAS ECs was not completely rescued by 2-DG, BAY876 treatments or HK2 knock down. We note that there was a qualitative improvement in the organization of adherens junction organization, even though VE-Cadherin levels remain lowered. Hence, we have included this as a descriptive supplemental figure to provide readers with this information. We have tempered the language so that we are not over-interpreting this data, especially since we have not assessed permeability. Given that VE-cadherin’s molecular weight was shifted with 2-DG treatment on western blots, it seems to suggest that 2-DG impacts either the folding and/or post-translational modifications of this protein (as 2-DG is known to impact protein glycosylation²¹). However, HK2 knock down and BAY876 treatment do not alter the molecular weight of VE-cadherin, suggesting that this alteration is specific to 2-DG, rather than being associated with the suppression of glycolysis.

Although glycolysis inhibitors do not seem to fully rescue the VE-cadherin phenotype in this study, MEK inhibitors do robustly restore VE-cadherin junctions in mutant KRAS ECs, as shown in our previous studies^{11,13}. VE-cadherin disruption is a MEK-dependent phenotype that can be restored with various MEK/ERK inhibitors (based on our unpublished kinase screening experiment). We now include new data that shows that a combination of MEK and glycolysis inhibitors improves the organization of adherens junctions as well as rescuing VE-Cadherin levels at cellular junctions based on immunofluorescence staining (revised Figure EV5C). This further highlights the utility of a combinatorial treatment approach.

Revised Figure EV5C. VE-cadherin Immunofluorescence Staining of IM-HUVEC KRAS^{G12V} Cells Treated with Combined MEK and Glycolysis Inhibitors. Representative IF images showing VE-cadherin in the stable IM-HUVEC KRAS^{G12V} cell line with 2mM 2-DG treatment alone or 2-DG treatment combined with either 10 μ M SL327 or 100nM Trametinib for 2 days. White asterisks indicate examples of obvious gaps between cells. Scale bars = 100 μ m.

“6. It is extremely difficult to visualize the shunt structures (or their recovery) at the arrow sites in Figure

7A. Ideally, the magnification or the resolution of these images could be improved, which would help the reader to appreciate the effects being described by the authors.”

We thank the reviewer for this suggestion. In response to this comment, we have updated the images and enlarged the images to provide better clarity. It is also important to point out that all zebrafish shunt assessments were conducted by experienced zebrafish technicians who were blinded to treatment conditions. When they were quantifying shunt reversal, they were screening live zebrafish embryos and visualizing their blood flow in real-time. This also makes the identification of shunts easier when blood flow is visualized in motion. To aid in the visualization, we have now enlarged the *gata1:dsRed* images without the complicated EGFP-KRAS overlay and the overall brightfield images of the embryos (revised Figure 7A). The overlay and overview figures have been moved to the Appendix Figure S13 for readers who are interested to know the overall morphology of these zebrafish embryos following treatment. We also added better annotations to indicate where the shunts are located (revised Figure 7A). We have also included images of combination treatments with MEK and glycolysis inhibitors (revised Figure 7B), which are shown in the figure above in response to Referee #1.

Revised Figure 7A. MEK and Glycolysis Treatment Improves Arteriovenous Shunt Recovery in Zebrafish. Tg(*gata1:dsRed*) zebrafish injected with GFP-tagged KRAS^{G12V} constructs and imaged at 2 days-post-fertilization

(dpf). Inhibitor treatments were initiated at 2 dpf (dosed in water) after confirming the presence of shunts. Zebrafish were imaged for 3 more days of inhibitor treatment. Arrows indicate directions of blood flow. Scale bars = 100 μ m, 20X images.

Revised Appendix Figure S13. Inhibition of Glycolysis Restores Arteriovenous Shunt Phenotypes in Zebrafish. Tg(gata1:dsRed) zebrafish injected with GFP-tagged KRAS^{G12V} constructs and imaged at 2 days-post-fertilization (dpf). Inhibitor treatments were initiated at 2 dpf (dosed in water) after confirming the presence of shunts. Zebrafish were imaged after 24-72 hours of inhibitor treatment (representative images shown). Arrows indicate locations of shunts in the images. Scale bars = 500 μ m for 4X images and 100 μ m for 20X images.

“7. Importantly, what does the recovery of a shunt represent at the cellular level? An altered cell size or sprouting? What is the impact of 2-DG on overall embryo development? I would think that cell proliferation and migration would be affected globally by this treatment (and by MEK inhibition), limiting the capacity of any cell to undergo remodeling. More detail regarding the overall impact of the drug treatments on the vascular growth would be useful.”

The reviewer makes an important point. Regarding the zebrafish experiments, inhibitor concentrations were titrated to a dose that does not impact overall development, and vascular development in particular (note: this is also why we provided the 4X full body images of the zebrafish embryos in Appendix Figure S13, which showed normal growth and development over time). Therefore, we do not expect that the inhibitors that we used are impairing cell proliferation and migration globally but instead

are impacting pathological changes in the AVM. MEK and glycolysis inhibitors both impact angiogenic sprouting. As such, one possible molecular explanation for the recovery of shunt is the normalization of vessel sprouting and vascular patterning. As we are using developing zebrafish embryos in this model system, the vasculature is actively growing and remodeling over time, and modulation of excessive angiogenic sprouting of mutant KRAS ECs could contribute to proper vasculature development and prevent abnormal vascular patterning. As can be seen in the zebrafish images in Appendix Figure S13, fluorescently-tagged mutant KRAS ECs were present at the shunt region before and after inhibitor treatments. Thus, reversal of shunts is not attributed to the loss of mutant KRAS ECs at that location, but rather a modulation of cellular behaviours. It also remains possible that these inhibitors increase flow responsiveness of the cells, for example, through improvement in VE-Cadherin localization. It is also worth noting that in adult AVM patients, a growing body of literature seems to suggest that active and ongoing angiogenesis is an important characteristic of established adult AVMs²². Thus, modulation of angiogenesis is not only relevant in the developing embryo scenario, but also applicable to established AVMs. Additional time-lapse imaging studies could be performed in future studies to better understand the cellular changes that accompany shunt reversal.

Minor points:

“1. Sprouting images in Fig. 6A are not very clear in terms of displaying sprouting. I recommend zooming in to make the sprouting easier to see (similar to Fig. 6D).”

We thank the reviewer for this suggestion. We have added zoom-ins as suggested.

“2. There is not a lot of evidence presented in the current paper to support the authors' conclusion that KRAS activity leads to increased extracellular matrix degradation- simply fibronectin and HABP immunostaining. In contrast, studies have shown that increasing glycolysis in tumor cells promotes matrix accumulation (<https://doi.org/10.1016/j.canlet.2024.217156>). Could the authors strengthen their observations through assessment of MMPs or analyses of degradation fragments in the culture media?”

We thank the reviewer for this suggestion. We did not investigate the mechanism of ECM degradation in this study, as this phenotype had been previously reported by another group in KRAS^{G12V} ECs^{23,24}. They demonstrated that MMP1 is implicated in matrix degradation by mutant KRAS ECs as the vascular networks expand and remodel. We also observe that MMP1 is upregulated in our scRNA-sequencing dataset of human ECs from various pathological and fetal tissues (Reviewer Figure 5). However, as matrix proteases encompass a large family of proteins, MMP1 may not be the only player involved in this process. Mural cells and other cell types surrounding the endothelium play major roles in matrix deposition. As such, animal models, or at least organoid models, are better suited to tackle this research question, as 2D cell culture likely cannot fully recapitulate *in vivo* disease phenotypes. We have been careful to not state that matrix is degraded in the manuscript, as we have not tested this. Rather, we state that the levels of fibronectin and HABP were decreased. We have modified Figure 8, removing the word ‘degradation’ when referring to the extracellular matrix and instead refer to ‘remodeling’.

Reviewer Figure 5. Human Single-Cell RNA-Sequencing of Endothelial Cells Across Various Brain Pathologies Showing Matrix Protease Gene Expression. (A) Heatmaps (z-score) generated from published single-cell RNA-sequencing dataset by Wälchli and Ghobrial et al (2024). mRNA expressions of selected matrix protease genes were compared in ECs isolated from human adult temporal lobe (TL), fetal brain (fCNS), fetal peripheral tissue (fPeriph) and various brain pathologies, including arteriovenous malformations (AVM), meningioma (MEN), low-grade glioma (LGG), glioblastoma (GBM) and lung cancer metastasized to the brain (MET). Black box highlights the *MMP1* gene.

Reviewer Response References:

1. Fish JE, Flores Suarez CP, Boudreau E, Herman AM, Gutierrez MC, Gustafson D, DiStefano PV, Cui M, Chen Z, De Ruiz KB, et al. Somatic Gain of KRAS Function in the Endothelium Is Sufficient to Cause Vascular Malformations That Require MEK but Not PI3K Signaling. *Circ Res.* 2020;127:727-743. doi: 10.1161/CIRCRESAHA.119.316500
2. Shin M, Male I, Beane TJ, Villefranc JA, Kok FO, Zhu LJ, Lawson ND. Vegfc acts through ERK to induce sprouting and differentiation of trunk lymphatic progenitors. *Development.* 2016;143:3785-3795. doi: 10.1242/dev.137901
3. Legendijk AK, Gomez GA, Baek S, Hesselton D, Hughes WE, Paterson S, Conway DE, Belting HG, Affolter M, Smith KA, et al. Live imaging molecular changes in junctional tension upon VE-cadherin in zebrafish. *Nat Commun.* 2017;8:1402. doi: 10.1038/s41467-017-01325-6
4. Casie Chetty S, Sumanas S. Ets1 functions partially redundantly with Etv2 to promote embryonic vasculogenesis and angiogenesis in zebrafish. *Dev Biol.* 2020;465:11-22. doi: 10.1016/j.ydbio.2020.06.007
5. Sudduth CL, Blum N, Smits PJ, Cheng YS, Vivero MP, Harris MP, Lawson ND, Greene AK. MAP2K1 Mutation in Zebrafish Endothelial Cells Causes Arteriovenous Shunts Preventable by MEK Inhibition. *J Vasc Anom (Phila).* 2023;4. doi: 10.1097/jova.000000000000063
6. Bassi I, Jabali A, Levin L, Lambiase G, Moshe N, Farag N, Tevet Y, Perlmoter G, Egozi S, Leichner GS, et al. A high-throughput zebrafish screen identifies novel candidate treatments for kaposiform lymphangiomatosis (KLA). *J Exp Med.* 2025;222. doi: 10.1084/jem.20240513
7. Ardelean DS, Letarte M. Anti-angiogenic therapeutic strategies in hereditary hemorrhagic telangiectasia. *Front Genet.* 2015;6:35. doi: 10.3389/fgene.2015.00035
8. Girard R, Zeineddine HA, Koskimaki J, Fam MD, Cao Y, Shi C, Moore T, Lightle R, Stadnik A, Chaudagar K, et al. Plasma Biomarkers of Inflammation and Angiogenesis Predict Cerebral Cavernous Malformation Symptomatic Hemorrhage or Lesional Growth. *Circ Res.* 2018;122:1716-1721. doi: 10.1161/CIRCRESAHA.118.312680

9. Martinez-Corral I, Zhang Y, Petkova M, Ortsater H, Sjoberg S, Castillo SD, Brouillard P, Libbrecht L, Saur D, Graupera M, et al. Blockade of VEGF-C signaling inhibits lymphatic malformations driven by oncogenic PIK3CA mutation. *Nat Commun*. 2020;11:2869. doi: 10.1038/s41467-020-16496-y
10. Kobialka P, Sabata H, Vilalta O, Gouveia L, Angulo-Urarte A, Muixi L, Zanoncello J, Munoz-Aznar O, Olaciregui NG, Fanlo L, et al. The onset of PI3K-related vascular malformations occurs during angiogenesis and is prevented by the AKT inhibitor miransertib. *EMBO Mol Med*. 2022;14:e15619. doi: 10.15252/emmm.202115619
11. Nikolaev SI, Vetiska S, Bonilla X, Boudreau E, Jauhiainen S, Rezai Jahromi B, Khyzha N, DiStefano PV, Suutarinen S, Kiehl TR, et al. Somatic Activating KRAS Mutations in Arteriovenous Malformations of the Brain. *N Engl J Med*. 2018;378:250-261. doi: 10.1056/NEJMoa1709449
12. Sato H, Yamamoto H, Sakaguchi M, Shien K, Tomida S, Shien T, Ikeda H, Hatono M, Torigoe H, Namba K, et al. Combined inhibition of MEK and PI3K pathways overcomes acquired resistance to EGFR-TKIs in non-small cell lung cancer. *Cancer Sci*. 2018;109:3183-3196. doi: 10.1111/cas.13763
13. Soon K, Li M, Wu R, Zhou A, Khosraviani N, Turner WD, Wythe JD, Fish JE, Nunes SS. A human model of arteriovenous malformation (AVM)-on-a-chip reproduces key disease hallmarks and enables drug testing in perfused human vessel networks. *Biomaterials*. 2022;288:121729. doi: 10.1016/j.biomaterials.2022.121729
14. Chang MY, Rhee YH, Yi SH, Lee SJ, Kim RK, Kim H, Park CH, Lee SH. Doxycycline enhances survival and self-renewal of human pluripotent stem cells. *Stem Cell Reports*. 2014;3:353-364. doi: 10.1016/j.stemcr.2014.06.013
15. Xiong X, Liao X, Qiu S, Xu H, Zhang S, Wang S, Ai J, Yang L. CXCL8 in Tumor Biology and Its Implications for Clinical Translation. *Front Mol Biosci*. 2022;9:723846. doi: 10.3389/fmolb.2022.723846
16. Carmeliet P, Moons L, Luttun A, Vincenti V, Compernelle V, De Mol M, Wu Y, Bono F, Devy L, Beck H, et al. Synergism between vascular endothelial growth factor and placental growth factor contributes to angiogenesis and plasma extravasation in pathological conditions. *Nat Med*. 2001;7:575-583. doi: 10.1038/87904
17. Chavkin NW, Genet G, Poulet M, Jeffery ED, Marziano C, Genet N, Vasavada H, Nelson EA, Acharya BR, Kour A, et al. Endothelial cell cycle state determines propensity for arterial-venous fate. *Nat Commun*. 2022;13:5891. doi: 10.1038/s41467-022-33324-7
18. Genet G, Genet N, Paila U, Cain SR, Cwiek A, Chavkin NW, Serbulea V, Figueras A, Cerda P, McDonnell SP, et al. Induced Endothelial Cell Cycle Arrest Prevents Arteriovenous Malformations in Hereditary Hemorrhagic Telangiectasia. *Circulation*. 2024;149:944-962. doi: 10.1161/CIRCULATIONAHA.122.062952
19. Mori Y, Yamawaki K, Ishiguro T, Yoshihara K, Ueda H, Sato A, Ohata H, Yoshida Y, Minamino T, Okamoto K, et al. ALDH-Dependent Glycolytic Activation Mediates Stemness and Paclitaxel Resistance in Patient-Derived Spheroid Models of Uterine Endometrial Cancer. *Stem Cell Reports*. 2019;13:730-746. doi: 10.1016/j.stemcr.2019.08.015
20. Ueda H, Ishiguro T, Mori Y, Yamawaki K, Okamoto K, Enomoto T, Yoshihara K. Glycolysis-mTORC1 crosstalk drives proliferation of patient-derived endometrial cancer spheroid cells with ALDH activity. *Cell Death Discov*. 2024;10:435. doi: 10.1038/s41420-024-02204-y
21. Kurtoglu M, Gao N, Shang J, Maher JC, Lehrman MA, Wangpaichitr M, Savaraj N, Lane AN, Lampidis TJ. Under normoxia, 2-deoxy-D-glucose elicits cell death in select tumor types not by inhibition of glycolysis but by interfering with N-linked glycosylation. *Mol Cancer Ther*. 2007;6:3049-3058. doi: 10.1158/1535-7163.MCT-07-0310
22. Lobeek D, Bouwman FCM, Aarntzen E, Molkenboer-Kuenen JDM, Flucke UE, Nguyen HL, Vikkula M, Boon LM, Klein W, Laverman P, et al. A Clinical Feasibility Study to Image Angiogenesis in

- Patients with Arteriovenous Malformations Using (68)Ga-RGD PET/CT. *J Nucl Med.* 2020;61:270-275. doi: 10.2967/jnumed.119.231167
23. Sun Z, Lin PK, Yrigoin K, Kemp SS, Davis GE. Increased Matrix Metalloproteinase-1 Activation Enhances Disruption and Regression of k-RasV12-Expressing Arteriovenous Malformation-Like Vessels. *Am J Pathol.* 2023;193:1319-1334. doi: 10.1016/j.ajpath.2023.05.015
 24. Lin PK, Sun Z, Davis GE. Defining the Functional Influence of Endothelial Cell-Expressed Oncogenic Activating Mutations on Vascular Morphogenesis and Capillary Assembly. *Am J Pathol.* 2024;194:574-598. doi: 10.1016/j.ajpath.2023.08.017

21st Nov 2025

Dear Dr. Fish,

Thank you for submitting your revised study. As you will see below, the referees are satisfied with the revisions, and I will therefore be able to accept your manuscript once the following editorial concerns are addressed:

1/ Please provide a response to referee #2 in your point-by-point rebuttal letter.

2/ Manuscript text:

- Please accept previous changes and only keep in track changes mode any new modification in the text.
- Please provide up to 5 keywords.
- Please correct the order and headings of the manuscript sections: Abstract / Keywords / The Paper Explained / Introduction / Results / Discussion / Methods / Data Availability / Acknowledgements / Disclosure and Competing Interests Statement / References / Figure Legends / Tables / Expanded View Figure Legends.
- Remove the list of suppl. materials from the manuscript text.
- Remove the list of abbreviations from the manuscript and instead incorporate the abbreviations into the text.
- Remove "Data not shown" (p.10). All data referred to in the paper should be displayed in the main or Expanded View figures.
- Methods:

o Please incorporate the Online Supplementary Materials in the main manuscript file.

o Zebrafish: State details of authority granting ethics approval, provide reference number for approval.

o Patients: please confirm that the experiments conformed to the principles set out in the WMA Declaration of Helsinki and the Department of Health and Human Services Belmont Report. If collected and within the bounds of privacy constraints report on age, sex and gender or ethnicity for all study participants.

o Mice: please indicate the gender of the mice used in the experiments.

o Antibodies: please provide dilutions/concentrations.

- Data Availability: remove "Source Data are provided for the main figures". Please provide the specific URLs for PXD062375 and GSE292538 datasets, and note that the datasets must be public before acceptance of the manuscript.

- Please change the heading "Infographics" to "Graphics" and use the following format:

Graphics:

(some of the... OR Figure #... OR synopsis) Graphics were created with BioRender.com. This section should be placed after Methods.

- Please merge 'Funding' with 'Acknowledgements'.

- Author contributions: Please remove the Authors Contributions from the manuscript and use the free text boxes beneath each contributing author's name in our system to add specific details on the author's contribution. More information is available in our guide to authors.

- Please rename "Competing interests" to "Disclosure and competing interests statement".

- Please reformat the references to alphabetical, with 10 author names listed before et al.

3/ Figures:

- Please remove the figures from the manuscript text. Main figures and EV figures should be uploaded as separate, high resolution figure files. Please remove "Expanded View" from the EV figure files.

- Appendix: Please add a table of contents with page numbers to the first page and correct the nomenclature to "Appendix Figure S1" etc.

- Please check the composition of your Appendix Figure S2C, as left and right pictures show similarities.

- Please address the queries from our data editors in the figure legends:

1. Please define the annotated p values ****/***/**/* as well as provide the exact p-values for the same in the figures or legends of figure 1A-C; D, F, G, H, I; 2A, B, C, D, G, H, I; 5A, B, C, D; 6A-E; 7A-C; EV1 B, C, F; EV2 I, J, M; EV4 A, EV5 B as appropriate.

2. Please indicate the statistical test used for data analysis in the legends of figures 3B, C, D; 4B, D, E; EV3 B-G

3. Please note that information related to n is missing in the legends of figures 3B, D

4/ Synopsis:

- Please remove the synopsis from the manuscript text and upload it as a separate file.

- Thank you for providing a nice visual abstract. I have resized it and cropped a small portion to serve as a thumbnail on our website (attached). Please let us know if you agree with this selection, or provide an alternative image at the right dimensions (115 x 70 px).

5/ As part of the EMBO Publications transparent editorial process initiative (see our Editorial at <http://embomolmed.embopress.org/content/2/9/329>), EMBO Molecular Medicine will publish online a Review Process File (RPF) to accompany accepted manuscripts.

This file will be published in conjunction with your paper and will include the anonymous referee reports, your point-by-point response and all pertinent correspondence relating to the manuscript. Let us know whether you agree with the publication of the RPF and as here, if you want to remove or not any figures from it prior to publication. Please note that the Authors checklist will be published at the end of the RPF.

I look forward to receiving your revised manuscript.

Yours sincerely,

Lise Roth

***** Reviewer's comments *****

Referee #1 (Comments on Novelty/Model System for Author):

This work underscores the increasing importance of incorporating metabolic programs into therapeutic considerations for vascular anomalies, translated from oncology. I am very pleased with the revisions and confident that the manuscript will be well-received by the EMBO Molecular Medicine readership, with meaningful future clinical impact.

Referee #1 (Remarks for Author):

I am very pleased with the revisions made to the manuscript. The implementation of the combination therapy approach, utilizing a glycolysis inhibitor alongside a MEK inhibitor at doses five times lower than that of the respective monotherapies in the zebrafish model, is particularly noteworthy. This approach holds substantial clinical significance and emphasizes the critical role of integrating metabolic pathways into therapeutic strategies - a concept increasingly affirmed in oncology. Moreover, the establishment of the 3D vascular model is commendable, offering a robust tool for in vitro drug screening. I eagerly anticipate further insightful contributions from this research team.

Referee #2 (Remarks for Author):

The authors have addressed my comments, but I remain confused about the molecular mechanism. MEK/ERK pathway has never been demonstrated to recruit HK2, and AKT should be activated in the context of a KRAS mutation. I am aware of their previous publication, but I was already puzzled by the in vitro data.

Referee #3 (Comments on Novelty/Model System for Author):

I think the authors have put together a convincing series of experiments illustrating the effects of glycolytic inhibition on KRAS-induced AVMs, and these findings have some translational potential. I am satisfied with the revisions.

Referee #3 (Remarks for Author):

The authors have provided a thoughtful and thorough response to reviewers' comments. Their data are comprehensive and convincing. All previous concerns have been addressed to my satisfaction.

We thank the editor for their positive response regarding our resubmission and for the suggestions for improvement of our manuscript. We have incorporated the editorial suggestions and have responded to the remaining comment from referee #2. The comments are included in red text below.

1/ Please provide a response to referee #2 in your point-by-point rebuttal letter.

Referee #2 (Remarks for Author):

The authors have addressed my comments, but I remain confused about the molecular mechanism. MEK/ERK pathway has never been demonstrated to recruit HK2, and AKT should be activated in the context of a KRAS mutation. I am aware of their previous publication, but I was already puzzled by the in vitro data.

We thank the reviewer for their comment and the opportunity to provide further clarification. We agree with the reviewer that there is a lack of knowledge in the field regarding the precise molecular mechanisms regulating the expression of HK2. While further mechanistic studies are needed, we have demonstrated the relationship between the MEK/ERK signaling pathway and HK2 expression in endothelial cells through the use of MEK inhibitors. MEK inhibitors reduce both the RNA and protein levels of HK2, as evident in our RNA-sequencing and proteomics studies. This is suggestive of regulation of HK2 at the transcriptional level, rather than through protein-protein interaction with KRAS or MEK/ERK components. Similar findings were observed in an oncology study where MEK1/2 inhibition reduced HK2 expression (Falck Miniotis et al, 2013). This study also suggested that the MEK/ERK-associated reduction in HK2 levels could be driven by the modulation of c-MYC, a transcription factor that regulates HK2 expression. Indeed, we have an ongoing study with preliminary evidence showing that MYC transcriptionally regulates HK2 expression in mutant KRAS endothelial cells. MYC expression is known to be regulated, at least in part, through the MAPK/ERK pathway (reviewed in Stine et al, 2015). Determining the mechanistic basis of HK2 regulation by MEK is outside the scope of the current manuscript but will be the focus of a follow-up study.

Regarding the activation of AKT in mutant KRAS cells, we also find the differences in molecular signaling in mutant KRAS endothelial cells and cancer cells to be of great interest. Future studies, such as BioID experiments, will be needed to understand the interacting partners of mutant KRAS in endothelial compared to cancer cells. It is worth pointing out that, even in the context of cancer cells, not all cancers feature both hyper-activated MAPK and PI3K signaling. For example, in a study by Hobbs et al (2020), they showed that KRAS^{G12R} is defective for interaction with p110 α PI3K (PI3K α) (Hobbs et al, 2020). Similar studies should be performed to understand the role of KRAS signaling in diverse cell types, such as endothelial cells.

References:

Falck Minitis M, Arunan V, Eykyn TR, Marais R, Workman P, Leach MO, Belouèche-Babari M (2013) MEK1/2 inhibition decreases lactate in BRAF-driven human cancer cells. *Cancer Res.* **73**(13):4039-49.

Hobbs GA, Baker NM, Miermont AM, Thurman RD, Pierobon M, Tran TH, Anderson AO, Waters AM, Diehl JN, Papke B et al (2020) Atypical KRASG12R Mutant Is Impaired in PI3K Signaling and Macropinocytosis in Pancreatic Cancer. *Cancer Discov.* **10**(1):104-123.

Stine ZE, Walton ZE, Altman BJ, Hsieh AL, Dang CV (2015) MYC, Metabolism, and Cancer. *Cancer Discov.* **5**(10):1024-39.

27th Jan 2026

Dear Dr. Fish, Dear Jason,

Thank you for submitting your revised files. I am pleased to inform you that your manuscript is accepted for publication and is now being sent to our publisher to be included in the next available issue of EMBO Molecular Medicine!

You may qualify for financial assistance for your publication charges - either via a Springer Nature fully open access agreement or an EMBO initiative. Check your eligibility: <https://link.springer.com/journal/44321/how-to-publish-with-us>

With kind regards,

Lise

>>> Please note that it is EMBO Molecular Medicine policy for the transcript of the editorial process (containing referee reports and your response letter) to be published as an online supplement to each paper. If you do NOT want this, you will need to inform the Editorial Office via email immediately. More information is available here: <https://link.springer.com/partners/embo-press/editorial-policies#Peer%20review>